# Plasma lipopolysaccharide levels predict mortality in acutely ill children in Low- and Middle-Income Countries

Childhood mortality remains high in low-resource settings, where environmental enteric dysfunction (EED) is prevalent. Peripheral blood bacterial lipopolysaccharides (LPS) are potential biomarkers of intestinal microbial translocation and inflammation; however, the effects of LPS translocation on mortality in this context remains unexplored. We investigate the association between plasma LPS and mortality among 638 acutely ill hospitalised children and compare them to 251 well community peers in a nested case-cohort (NCC) conducted between November 2016 and January 2019 across 9 sites in 6 countries in sub-Saharan Africa and South Asia. Higher levels of plasma LPS and inflammatory biomarkers (fecal calprotectin, plasma myeloperoxidase, and CD14) are associated with elevated 90-day mortality, and those associations are independent of wasting status. Non-survivors with high plasma LPS exhibit elevated gram-negative enteric microbiota, increased fecal biomarkers of EED, systemic inflammatory proteins, and differentially expressed proteins linked to the Insulin-like growth factor (IGF) nutritional axis, Interleukin-1 and collagen regeneration. Cellular interaction network models deconvoluted from a single-cell transcriptomic dataset enable an exploratory investigation of systemic immune responses and epithelial-immune cells crosstalk active in pathways leading to mortality. This knowledge can guide the identification of potential therapeutic signaling pathways in settings with high EED and malnutrition.

Childhood mortality remains a major global health challenge, with children from sub-Saharan Africa (55%) and South Asia (27%) accounting for the largest shares of deaths[1]. Environmental enteric dysfunction (EED), previously termed environmental enteropathy, is typically observed in resource-limited settings and originates from exposure to fecal-oral enteropathogens in areas with inadequate hygiene and sanitation practices. EED represents a condition characterized by chronic small intestinal inflammation, villous blunting, and crypt hyperplasia, which may result in wasting and stunting[2–6]. Both wasting and stunting are associated with an increased risk of mortality without a clear threshold effect[7]. The MAL-ED (Etiology, Risk Factors and Interactions of Enteric Infections and Malnutrition and the

Consequences for Child Health and Development) Birth Cohort study identified an increased burden of enteropathogens, growth impairment, and asymptomatic infections associated with EED biomarkers[8]. Increased expression of genes associated with dysregulated antimicrobial immunity and epithelial barrier function has been observed in children with EED[6,9]. The responses to oral vaccines are also diminished among these children[10]. Therapeutic interventions with antibiotics, micronutrients, sanitation and hygiene have shown limited success in reversing growth impediments or intestinal dysbiosis in children with EED[5]. Recent interventions in EED, such as teduglutide (glucagon-like peptide 2 analog; growth and repair), budesonide (enteric inflammation) and colostrum (epithelial permeability), show

✉e-mail: holm.uhlig@well.ox.ac.uk

partial promise in epithelial regeneration and restoration of barrier function and reducing enteric and systemic inflammation among children discharged from complicated malnutrition[11]. While the current paradigm suggests persistent inflammation and defective epithelial barriers lead to compromised nutrient absorption, hormonal pathways, and host protective immunity, the therapeutic options are limited[12].

Lipopolysaccharide (LPS) is a component in the outer membrane of most gram-negative bacteria. LPS binds to toll-like receptor 4 (TLR4) expressed on a number of immune and non-immune cells, such as myeloid and epithelial cells. LPS engages with TLR4 and accessory proteins (Myeloid Differentiation protein 2 (MD-2) and LPS-binding protein (LBP)) to induce an inflammatory response with tumor necrosis factor (TNF) and interleukin (IL)-1β release[13]. LPS-TLR4 interaction plays an indispensable role in gut homeostasis. TLR4 is expressed by intestinal epithelial cells and controls proliferation, apoptosis, mucin secretion, goblet cell differentiation, and Paneth cell degranulation[14]. In Zambian children, stunting was associated with elevated plasma LPS levels[2]. Plasma LPS levels are correlated with differential gene expression profiles in intestinal biopsies[15]. Biallelic TLR4 and MD2 loss-of-function variants can cause inborn errors of immunity with Crohn's disease-like intestinal inflammation[16,17]. Peripheral LPS level has been employed as a biomarker for microbial translocation, including in EED[18–20], but its association with mortality in the low-resource, EED-prevalent settings remains unexplored. Also, chronic inflammation has been considered one of the hallmarks of EED, but the mechanistic effect of this chronic inflammation in acutely ill children remains unknown.

In this study, we explored the association between plasma LPS levels and mortality in the Childhood Acute Illness and Nutrition (CHAIN) Network nested case cohort (NCC) from 9 sites across 6 countries in sub-Saharan Africa and South Asia. We report that increased levels of plasma LPS are related to short-term mortality in subjects. We perform a pathway analysis using plasma proteomics profiles to identify mechanisms. Our analysis identifies LPS as a biomarker of mortality with clear therapeutic implications.

## Results

### Baseline characteristics of the study children
To assess the association of plasma LPS with mortality, we investigated 638 hospitalized CHAIN NCC children and 251 community children[21] from nine sites in six countries across sub-Saharan Africa and South Asia (Fig. 1). 495 of the 638 hospitalized children also had LPS assessment at discharge. Baseline characteristics of the admission cohort and community children are summarized in Table 1. Age, sex distribution, HIV and breastfeeding statuses were similar (Wilcoxon Rank-Sum test or Chi-squared, $p \geq 0.05$) between the two groups. Unsurprisingly, comorbidities such as gastroenteritis, malaria, sepsis, and lower respiratory tract infection (LRTI) were more prevalent (Chi-squared test, $p < 0.001$) in the admission cohort and were associated with elevated fecal quantification (Wilcoxon Rank-Sum test, $p < 0.05$) of specific bacterial, viral and parasitic enteropathogens. Biomarkers of intestinal barrier function suggested more impaired integrity in the admission cohort, likely driven by acute infection and increased enteropathogen load.

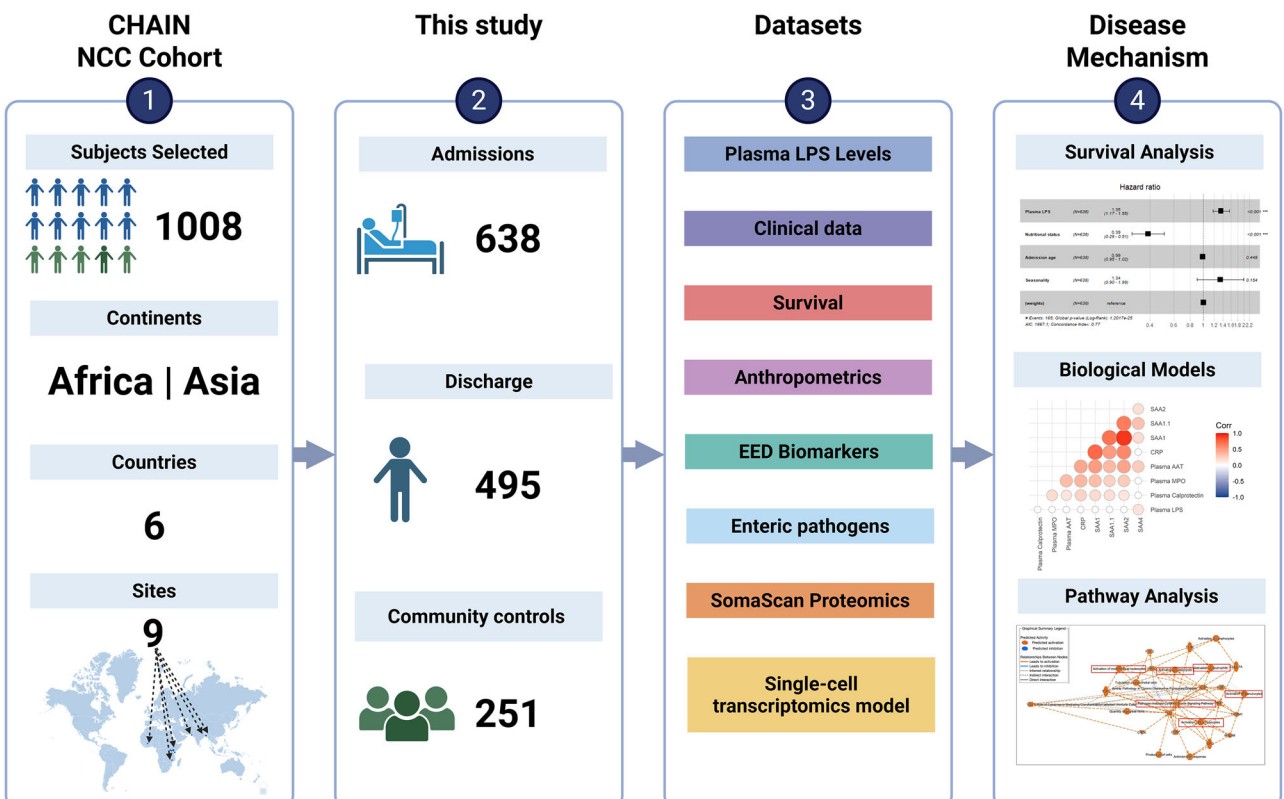

**Fig. 1 | CHAIN LPS study workflow.** The study analyzed data from 638 children from the CHAIN nested case-cohort (CHAIN NCC, details[21]) who were admitted to the hospital (admissions) and 251 well children from the same neighborhoods as the admissions cohort (community). Overall, these children were from 9 sites across 6 countries in Sub-Saharan Africa and South Asia. This study includes data from LPS and SomaScan proteomics in plasma, markers for enteric dysfunction measured in stool, and TaqMan Array Card (TAC) enteropathogen data from fecal swabs. Plasma LPS data were also available for 495 cohort children at hospital discharge. Analyses included evaluation of plasma LPS levels, medical conditions, hospitalization, survival, plasma proteomics and deconvolution of a single-cell transcriptomic dataset. Abbreviations: LPS lipopolysaccharide, NCC nested case-cohort. Created in BioRender. Allen, C. (2025) https://BioRender.com/q3bom29.

**Table 1 | Demographic, Enteropathogen, fecal biomarkers and plasma barrier and systemic inflammation profiles of hospitalized vs. community children**

| Characteristics | Admission Weighted N = 1894[a] Unweighted n = 638 | Community children Weighted N = 251[a] Unweighted n = 251 | p-Value[b] |
|---|---|---|---|
| Plasma LPS (E.U./ml) (Mean (SD)) | 4.40 (5.25) | 4.47 (5.34) | >0.9 |
| **Demographics** | | | |
| Age group | | | 0.2 |
| <6 months | 361 / 1,894 (19%) | 41 / 251 (16%) | |
| 6-11 months | 690 / 1,894 (36%) | 81 / 251 (32%) | |
| 12 months & above | 844 / 1,894 (45%) | 129 / 251 (51%) | |
| Admission age (Mean (SD)) | 11.47 (5.57) | 12.48 (6.11) | 0.049 |
| Sex | | | 0.047 |
| Female | 742 / 1,894 (39%) | 118 / 251 (47%) | |
| Male | 1,152 / 1,894 (61%) | 133 / 251 (53%) | |
| Continent | | | 0.032 |
| Africa | 1,085 / 1,894 (57%) | 165 / 251 (66%) | |
| Asia | 809 / 1,894 (43%) | 86 / 251 (34%) | |
| **Comorbidity frequencies** | | | |
| Malaria History | 337 / 1,894 (18%) | 11 / 251 (4.4%) | <0.001 |
| Sepsis | 295 / 1,894 (16%) | 0 / 251 (0%) | <0.001 |
| Gastroenteritis | 1,142 / 1,872 (61%) | 0 / 251 (0%) | <0.001 |
| URTI | 98 / 1,894 (5.2%) | 0 / 251 (0%) | 0.060 |
| LRTI | 774 / 1,894 (41%) | 0 / 251 (0%) | <0.001 |
| HIV status (Positive) | 50 / 1,894 (2.6%) | 3 / 251 (1.2%) | 0.2 |
| TB | 3 / 1,701 (0.2%) | 0 / 221 (0%) | 0.6 |
| Breastfeeding status | 1,428 / 1,894 (75%) | 202 / 251 (80%) | 0.12 |
| **Presence of at least 1 Enteropathogen** | | | |
| Viruses | 776 / 1,702 (46%) | 53 / 222 (24%) | <0.001 |
| Parasites | 517 / 1,770 (29%) | 56 / 231 (24%) | 0.2 |
| Gram-negative Bacteria | 1,579 / 1,820 (87%) | 206 / 241 (85%) | 0.6 |
| **Enteropathogen Quantification (Mean (SD))** [c] | | | |
| Enteroaggregative E.coli | 28.85 (6.31) | 30.31 (5.32) | 0.007 |
| Enterotoxigenic E.coli | 32.79 (4.69) | 33.86 (2.91) | 0.008 |
| Shigella toxin-producing Enterotoxigenic E.coli | 33.51 (4.06) | 34.63 (1.71) | <0.001 |
| V.cholerae | 34.79 (1.53) | 35.00 (0.02) | 0.006 |
| CTX-M-producing bacteria | 27.24 (6.79) | 30.62 (4.99) | <0.001 |
| mphA Azithromysin Resistance | 23.08 (6.47) | 27.36 (5.69) | <0.001 |
| 16S rRNA | 13.90 (3.28) | 15.25 (3.71) | <0.001 |
| Cryptosporidium | 33.90 (3.46) | 34.35 (2.58) | 0.015 |
| Adenovirus 40/41 | 34.22 (3.19) | 34.71 (1.59) | 0.005 |
| Rotavirus | 33.01 (4.26) | 34.92 (0.82) | <0.001 |
| Sapovirus | 34.64 (1.65) | 34.89 (0.77) | 0.029 |
| **Fecal Biomarkers of EED (Mean (SD))** | | | |
| Fecal MPO ηg/ml | 4,105.51 (4,584.89) | 3,708.52 (3,552.86) | 0.7 |
| Fecal Calprotectin ( > 12 months) (µg/ml) | 698.85 (1,086.33) | 502.96 (1,192.81) | 0.13 |
| Fecal AAT (non-acute diarrhea) (µg/ml) | 448.57 (557.01) | 341.36 (372.63) | 0.30 |
| **Barrier Function Biomarker (Mean (SD))**[d] | | | |
| Plasma Zonulin | 46,479.64 (29,948.42) | 33,384.53 (19,464.19) | <0.001 |
| Plasma Diamine Oxidase | 4,776.11 (4,226.63) | 6,809.30 (3,594.82) | <0.001 |
| Plasma CDH1.1 | 1,468.48 (2,669.83) | 1,602.89 (2,216.58) | 0.005 |
| Plasma OCLN | 960.71 (291.05) | 908.77 (166.60) | 0.005 |
| Plasma ZO-1 | 719.87 (1,718.78) | 788.77 (1,570.68) | <0.001 |
| **Systemic Immune Activation and Inflammation Biomarkers (Mean (SD))** [d] | | | |
| Plasma Heparin-Binding Protein | 24,326.27 (11,449.57) | 19,629.43 (8,129.72) | <0.001 |
| Plasma TREM1 | 2,168.66 (1,457.04) | 1,691.83 (515.93) | <0.001 |

**Table 1 (continued) | Demographic, Enteropathogen, fecal biomarkers and plasma barrier and systemic inflammation profiles of hospitalized vs. community children**

| Characteristics | Admission Weighted N = 1894[a] Unweighted n = 638 | Community children Weighted N = 251[a] Unweighted n = 251 | p-Value[b] |
|---|---|---|---|
| *Systemic Inflammatory Biomarkers (Mean (SD))* [d] | | | |
| Plasma CRP | 69,067.92 (35,630.07) | 32,055.93 (28,576.09) | <0.001 |
| Plasma SAA1 | 72,275.72 (72,226.17) | 17,974.93 (35,545.88) | <0.001 |
| Plasma MPO | 27,618.30 (14,306.43) | 16,296.82 (5,131.00) | <0.001 |
| Plasma CAL | 1,773.65 (1,065.01) | 1,749.45 (3,856.34) | <0.001 |
| Plasma AAT | 50,645.94 (15,708.32) | 33,395.31 (9,471.53) | <0.001 |

[a]N = inverse proportionally weighted total within CHAIN cohort (n = 3101); [b]Design-based Wilcoxon test, Pearson's X^2: Rao & Scott adjustment; [c]Threshold Cycle (Ct) from quantitative PCR (qPCR) assay; *E.U./ml* Endotoxin Unit per ml; [d]*RFU* Relative Fluorescence Unit; *SD* Standard Deviation.
Demographics (age, sex, continent), fecal biomarkers (MPO, CAL, AAT), and plasma LPS concentrations are comparable between Admission and Community children (p ≥ 0.05). In contrast, the presence of comorbidities (malaria, sepsis, gastroenteritis, LRTI), barrier protein expression (zonulin, diamine oxidase) and plasma systemic inflammatory (CRP, SAA1, MPO, CAL, AAT) differ significantly between the two groups (p < 0.001). Both unweighted and inverse proportionally weighted totals (N) are presented. Only enteropathogens with significant difference in expression between the groups are presented. See Supplementary Table 11 for all pathogen levels.
*LPS* Lipopolysaccharide, *URTI* Upper Respiratory Tract Infection, *LRTI* Lower Respiratory Tract Infection, *TB* Tuberculosis, *EED* Environmental Enteric Dysfunction, *MPO* Myeloperoxidase, *AAT* Alpha-1-antitrypsin, *CAL* Calprotectin, *CDH1.1* Cadherin-1, *OCLN* Occludin, *OCLN* Tight Junction Protein ZO-1, *TREM1* Triggering Receptor Expressed on Myeloid Cells 1, *CRP* C-reactive Protein, *SAA1* Serum Amyloid A1.

## High prevalence of EED in the CHAIN cohort, but no associations with plasma LPS

We initially characterized enteric dysfunction by investigating the levels of fecal biomarkers for enteric inflammation; myeloperoxidase (MPO) and calprotectin (CAL) and permeability; alpha-1-antitrypsin (AAT) and tested their association with plasma LPS. We observed that the levels of MPO, CAL and AAT did not differ (Wilcoxon Rank-Sum test, p > 0.05) between admitted children and those from community settings (Table 1 and Supplementary Fig. 4G-I). Stunting frequency and height-for-age Z-score (HAZ) was also similar (Chi-squared or Wilcoxon Rank-Sum test; p ≥ 0.05) between the groups.

The levels of these EED biomarkers were elevated in both admission and community children compared to references (MPO: 2000 ng/ml[22]; CAL: 122 µg/ml[23]; AAT: 270 µg/ml[24]) from high-income settings (Supplementary Table 9) but community children's expressions were comparable to children from LMIC (low- and middle-income countries) settings based on review of 5 independent cohorts (Supplementary Table 10). While IPW correlation matrices suggested that plasma LPS was weakly associated (Pearson's correlation, R < 0.3) with fecal biomarkers of EED (Supplementary Fig. 5A) and plasma biomarkers of barrier integrity (Supplementary Fig. 5B) among children at admission, no correlations were observed among community children (Supplementary Fig. 5C, D, respectively). These findings suggest that EED may be prevalent in the community within this setting and support the view that intestinal inflammation alone is not the cause of hospitalization or mortality.

## High plasma LPS concentrations are associated with 90-day mortality

We next compared the levels of plasma LPS within the study cohort at admission and discharge, and also compared levels within community children (Supplementary Table 1). Plasma LPS concentration (weighted mean (SD): 4.40 (5.25) and 4.47 (5.34) E.U./ml, respectively; Wilcoxon Rank-Sum test, p > 0.9) was similar (p ≥ 0.05) between the admission cohort and community children (Table 1). By contrast, inverse proportionally weighted (IPW) median interquartile range (Q1, Q3) plasma LPS levels were elevated at admission versus discharge in unpaired Wilcoxon Rank-Sum test comparisons (2.5 (0.5, 6.0), 2.0 (0.3, 4.2) E.U./ml; Supplementary Table 2). Paired comparisons (Wilcoxon Rank-Sum test, p = 0·14; Supplementary Table 3) did not show this trend, indicating that reduction of levels at discharge may be driven by increased mortality in children with high plasma LPS levels.

The LPS levels of the 165 children who died within 90 days of hospital admission (weighted mean (SD): 7(6.7) E.U./ml) were elevated, compared to the 34 who died after 90 days or the 349 who survived

(weighted mean (SD): 6.2 (7.1) and 4.2 (5.0) E.U./ml, respectively; Kruskal–Wallis test, p < 0.001; Supplementary Table 6). The distribution of anthropometric classification (p < 0.001) and sex (p = 0.003), but not age, was significantly different among deceased children (chi-squared test, Supplementary Table 7). Plasma LPS associations with mortality were reinforced by elastic net regression (Supplementary Table 8), which also identified other mortality-related comorbidities, such as anthropometric factors (MUAC, weight, height), blood glucose, dehydration, capillary refill time, fecal bacterial populations and fecal calprotectin.

We then categorized children at admission into tertiles depending on the levels of their plasma LPS (Supplementary Table 4). Children who were hospitalized with plasma LPS concentration in highest tertile (weighted mean: 10.24 E.U./ml) had significantly elevated mortality, compared to those in the lowest tertile (weighted mean: 0.28 E.U./ml) (% mortality: 10.75% vs. 4.25%, respectively; chi-squared test, p < 0.001; Supplementary Tables 4 and 5).

To further understand the association between plasma LPS concentrations and mortality, we determined an optimal threshold for plasma LPS that would stratify patients into either high or low plasma LPS based on the greatest separation of mortality outcomes (Supplementary Methods 4). Our analysis on density distributions and maximally selected rank statistics showed that the optimal cutoff for plasma LPS was 11.58 EU/ml (Supplementary Fig. 1A, B). Using this cutoff, we generated unweighted Kaplan–Meier survival curves. Children presenting with high plasma LPS levels (Supplementary Fig. 1A) had increased risk of 90-day mortality (p < 0.001; Fig. 2A). This was confirmed by an inverse proportional weighted Cox proportional hazards models (CoxPH) (Hazard ratio: 1.4 (95% CI: 1.21, 1.71); p < 0.001; Fig. 2B) with adjustments for anthropometric classification and sex. Proportional hazards and linearity assumptions were assessed with no evidence of violations (Supplementary Fig. 2A–C). More severe anthropometric classification was significantly associated with mortality, consistent with recent work published by the CHAIN Network and others[25]. However, an independent association between plasma LPS and mortality was also observed among non-severely acutely malnourished (non-SAM) children (MUAC > 11.5 cm) in the hospital admission cohort (p < 0.001; Fig. 2C). This association remained significant in IPW CoxPH models, in which other demographic (age, sex and site), comorbidities (gastroenteritis, malaria, sepsis, HIV, LRTI, URTI) and/or biomarker covariates were mutually adjusted for (hazard ratios: 1·35 (95% CI: 1.11, 1.63, p = 0.002) and 1.43 (95% CI: 1.12, 1.83, BH-adjusted p = 0.014), Fig. 2D, E, respectively).

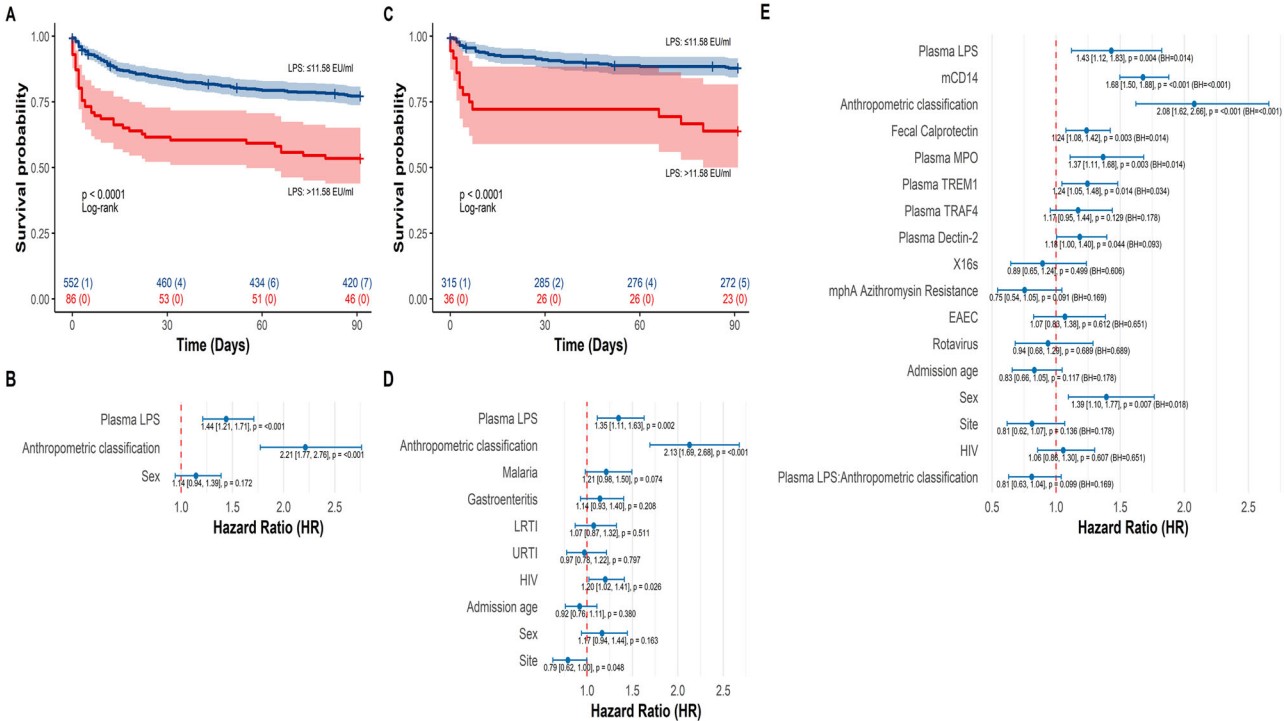

**Fig. 2 | Levels of plasma LPS at admission are associated with mortality.** Elevated plasma LPS is associated with increased mortality risk in hospitalized children ($n = 638$) as assessed by **A** unweighted Kaplan Meier curve and **B**, **D** inverse proportionally weighted Cox proportional hazard models. **C** Unweighted Kaplan Meier curve also shows association between plasma LPS and mortality in non-severely acutely malnourished (non-SAM; MUAC > 11.5 cm) hospitalized children ($n = 351$). **E** Elevated plasma LPS, membrane CD14 (mCD14), fecal calprotectin, plasma MPO, TREM1, more severe anthropometric classification, and female sex were independently associated with increased risk of mortality among admitted children with proteomics data available ($n = 635$), based on a Cox proportional hazards model. Thirty-eight biomarkers were selected: (1) 19 based on biological relevance and differential expression in unadjusted or adjusted comparisons (see Supplementary Tables 14 and 15), and (2) 19 based on strongest correlations with plasma LPS in the $LPS^{hi}$ F subgroup (Fig. 3A). Biomarkers with $p < 0.2$ in univariate Cox models were retained, resulting in 22 proteins (Supplementary Table 20). No highly correlated pairs were identified (all Pearson's $R < 0.9$). The final multivariable model included plasma LPS, four differentially detected enteropathogen markers, six biomarkers significantly associated with mortality from the full multivariate model (Supplementary Table 21) and known mortality-related factors from Fig. 2D

(anthropometric classification, site, and HIV status), as well as age and sex.
**A**, **C** Plasma LPS levels were stratified according to maximally selected rank statistics (Supplementary Fig. 1A, B), and survival curves were compared by Log-rank tests. Colored shadings represent a 95% confidence interval for each stratum. Blue lines represent children who presented with low LPS (≤11.58 E.U./ml), while red lines denote those presenting with high LPS (>11.58 E.U./ml). Each figure also displays a table of the number of at-risk participants and the cumulative number of censored cases. **B**, **D**, **E** Forest plots show point estimates of log hazard ratios with 95% confidence interval bars and raw $p$-values from inverse proportionally weighted Cox proportional hazard models. Analyses were weighted to account for selection biases. Covariates included in the CoxPH models were mutually adjusted. Anthropometric classifications were ordered from normal to severe. The models satisfied hazards and linearity assumptions (Supplemental Fig. 2A–C). **A–E** All $n$ values refer to independent children; no technical replicates were used. All analyses were two-tailed. **E** Benjamini–Hochberg (BH) correction was applied to account for multiple comparisons. LPS Lipopolysaccharide, URTI Upper Respiratory Tract Infection, LRTI Lower Respiratory Tract Infection, MPO Myeloperoxidase, TREM1 Triggering Receptor Expressed on Myeloid Cells 1, TRAF4 TNF Receptor-associated Factor 4, X16s 16S rRNA, EAEC Enteroaggregative E. coli, F Fatal, NF Non-fatal.

## Patients with high plasma LPS and increased mortality have a plasma protein signature linked to immune and regenerative processes

To understand what biological mechanisms underlie the link between LPS and mortality, we assessed plasma biomarkers. Systemic activation and inflammatory biomarkers and LPS signal transduction proteins were significantly elevated at hospital admission (Table 1 and Supplementary Fig. 4D–F). Hospitalized children were categorized into 4 subgroups (*LPS^hi Fatal; F;* died within 90 days post-hospitalization), *LPS^hi Non-fatal (NF;* alive 90 days post-hospitalization), *LPS^lo F, LPS^lo NF)* using a plasma LPS cut point of 11.58 E.U./ml (Supplementary Fig. 1).

IPW correlation analysis identified the top 18 proteins most correlated with plasma LPS (Pearson's correlation, $R ≥ 0.4; p < 0.05$) in the *LPS^hi F* sub-group (Fig. 3A and Supplementary Table 12). These proteins had functions in immunity (e.g., SIRPB1 and CD7), tissue repair and regeneration (e.g., DNAJB2), nutrition (e.g., SMS) and LPS signaling (e.g., TRAF4 and CLEC6A) (Supplementary Tables 22 and 23).

Proteins related to the insulin-like growth factor (IGF) nutritional axis (IGF1, IGF2, IGFBP3), and collagen regeneration (COL10A1,

COL1A1) were among the top differentially abundant proteins in the *LPS^hi F* sub-group compared to community children (Supplementary Tables 16 and 17). The expression of these proteins in *LPS^hi F* sub-group was significantly lower than in the other sub-groups (Fig. 4, Supplementary Tables 16 and 17). IGF differences persisted even after stratification based on anthropometric parameters (Supplementary Fig. 6) and adjustments for sex, age, HIV status, and site using generalized linear models (Supplementary Tables 18 and 19). These findings highlight the link between LPS-associated intestinal and systemic inflammation, as well as nutrition and regeneration, as indicators for mortality-associated inflammation and tissue remodeling.

## High LPS levels in the fatal group are associated with dysregulated CD14−TLR4 immune activation

Having observed that (1) plasma LPS concentration correlated with 90-day mortality (Fig. 2) and that (2) biomarkers of intestinal and systemic inflammation and LPS signaling were differentially abundant among LPS sub-groups (Kruskal–Wallis test, Table 2), we assessed potential confounders of LPS-associated mortality effects. Key comorbidities

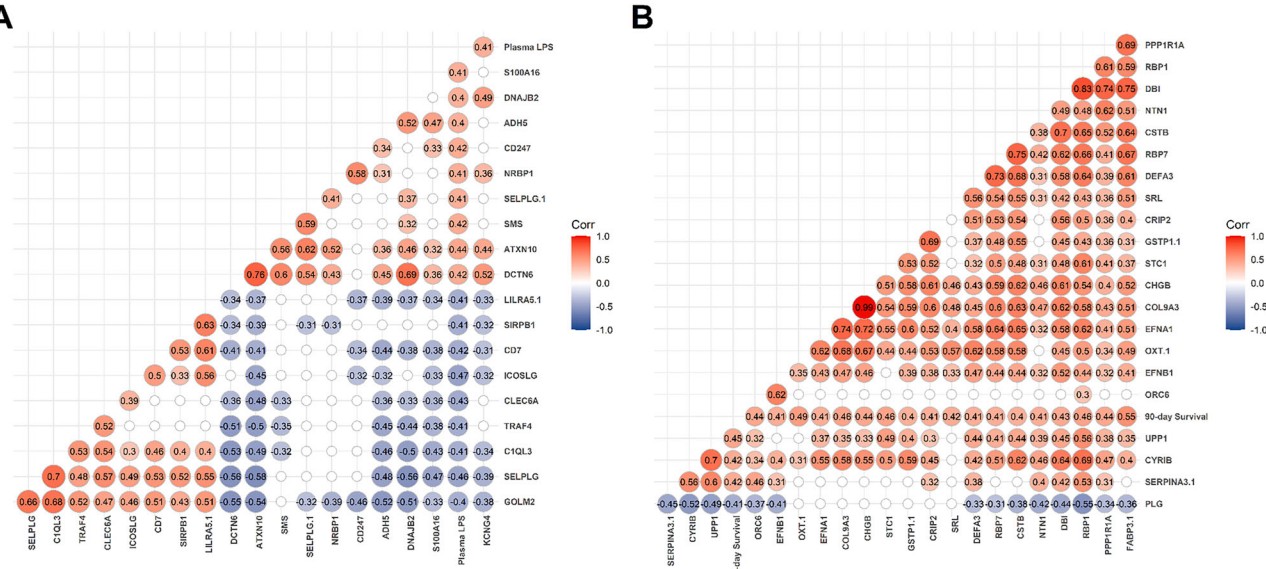

**Fig. 3 | Correlation of proteins and their associations with plasma LPS and 90-day survival in *LPS^hi Fatal* sub-group.** Correlograms depicting the associations of **A** plasma LPS and **B** 90-day survival with the strongest of correlations with SomaScan proteins detected in the *LPS^hi Fatal* sub-group (*n* = 39). Correlation matrices were inverse-proportionally weighted to account for selection biases. Correlograms were hierarchically clustered using the 'ward.D2' method. Each circle represents the Pearson's correlation coefficient (*r*), with color intensity indicating the strength of the correlation. Positive correlations are shown in red, and negative correlations are shown in blue. Only correlations with an absolute value of *r* ≥ 0.4 are displayed, highlighting moderate to high associations (see Supplementary Table 12). Additionally, only coefficients with a two-tailed *p*-value < 0.05 are shown; coefficients with *p*-value ≥ 0.05 are represented as blank, white circles. All *p*-values were adjusted for false discovery rates using the Benjamini–Hochberg (BH) method, based on the full SomaScan dataset of over 7300 proteins. All *n* values refer to independent children; no technical replicates were used.

prevalent in children in low-resource settings, including lower respiratory tract infections, sepsis, malaria or gastroenteritis, were not different among LPS-mortality sub-groups (Chi-squared test, Supplementary Table 13). There were significant differences in HIV status between the LPS-mortality sub-groups.

Interestingly, stool quantitative PCR (qPCR) assays suggested that overall bacterial load (16S rRNA), as well as gram-negative *E. coli* EAEC3 strains and *mphA* azithromycin-resistant bacteria were significantly elevated in the *LPS^hi F* sub-group (Kruskal–Wallis test, Table 2).

Compared to other sub-groups, the *LPS^hi F* children exhibited a significant elevation of intestinal inflammatory (fecal calprotectin), systemic inflammatory (plasma MPO), and LPS signaling-associated (CD14 and TREM1) biomarkers, but reduced TLR4 levels (Kruskal–Wallis test, Table 2). This indicates differential LPS-mediated immune signaling in the *LPS^hi F* sub-group. Notably, in an inverse proportionally weighted CoxPH model, these biomarkers were also independently associated with 90-day mortality, even when plasma LPS (hazard ratio: 1·43 (95% CI: 1.12, 1.83); BH-adjusted *p* = 0·014) and anthropometric classification (hazard ratio: 2.08 (95% CI: 1.62, 2.66); BH-adjusted *p* < 0.001) were adjusted for (Fig. 2E) - fecal calprotectin (hazard ratio: 1.24 (95% CI: 1.08, 1.42); BH-adjusted *p* = 0.014), plasma MPO (hazard ratio: 1.37 (95% CI: 1.11, 1.68); BH-adjusted *p* = 0.014), TREM-1 (hazard ratio: 1·24 (95% CI: 1.05, 1.48); *p* = 0.034) and plasma CD14 (hazard ratio: 1.68 (95% CI: 1.50, 1.88); *p* < 0.001).

**A cellular model of LPS-associated immunopathology**
Plasma proteomics analysis showed that immunologically relevant soluble membrane-derived proteins (like TLR4, IFNGR2) (Fig. 5A) or extracellular proteins (like CCL8 and IFNA7) (Fig. 5B) were down-regulated in *LPS^hi F* compared to all the other sub-groups, indicating an immune dysregulation. To understand the mechanisms of intestinal and systemic inflammation and LPS signaling, we leveraged our SomaScan proteomics data and a publicly available single-cell transcriptomics dataset from EED patients[26]. The expression of genes

encoding differentially abundant proteins in *LPS^hi F* sub-group by cell types in single cell datasets was analyzed (Supplementary Fig. 7). This deconvolution method helped us identify which intestinal cell types are potentially involved in immune signaling in *LPS^hi F* sub-group. The transcript level expression of the differentially expressed proteins in the single cell dataset indicated that these are expressed by various immune (i.e., *TLR4* and *VCAN*) and epithelial cell types (i.e., *DPP4* and *VTN*) in the small intestine (Fig. 5C). This similarly allowed for visualization of cellular interactions mediated by genes for relevant differentially abundant proteins, like *CCL8*, *IFNG*, *VEGF*, *VCAN*, etc., and their binding partners. The possible interactions, like–*CCL8* (expressed by monocytes) with *CCR2*/*CCR5* (plasma cells and T cells), *XCL1* (T cell) with *XCR1* (dendritic cells), *VCAN* (monocytes) with *CD44* (immune and epithelial cells) and *IFNG* (T cells) with *IFNGR2* (immune and epithelial cells), highlight immune–immune and immune–epithelial cell cross-talk (Fig. 6A, B), illustrating immunopathology in LPS-mediated mortality in the *LPS^hi F* sub-group.

To understand the possible cellular and molecular mechanisms involved in LPS-associated mortality, IPA analysis was performed on the detected proteins (log2fold change ≥ 1; Holm-adjusted Wilcoxon *p*-value ≤ 0.05) from each sub-group versus community children. In this exploratory systems approach, comparative pathway analysis of the four sub-groups revealed that cytokine-related pathways like IL-10 and VEGF signaling were upregulated, whereas IL-15 production was downregulated in the *LPS^hi F* cohort, compared to the other sub-groups (Fig. 7A). Thus, we hypothesize that immune-cytokine dysregulation may contribute to increased mortality in the cohort. To decipher the cellular cross-talks in these settings, the important immunological proteins involved in select pathways from IPA analysis, and their binding partners, determined using CellphoneDB, were deconvoluted on single-cell transcriptomics data. The proteins regulating these pathways are expressed by myeloid cells, intestinal epithelial and endothelial cells (Fig. 7B), indicating an immune crosstalk amongst these cell types in the small intestine. Given the pattern of

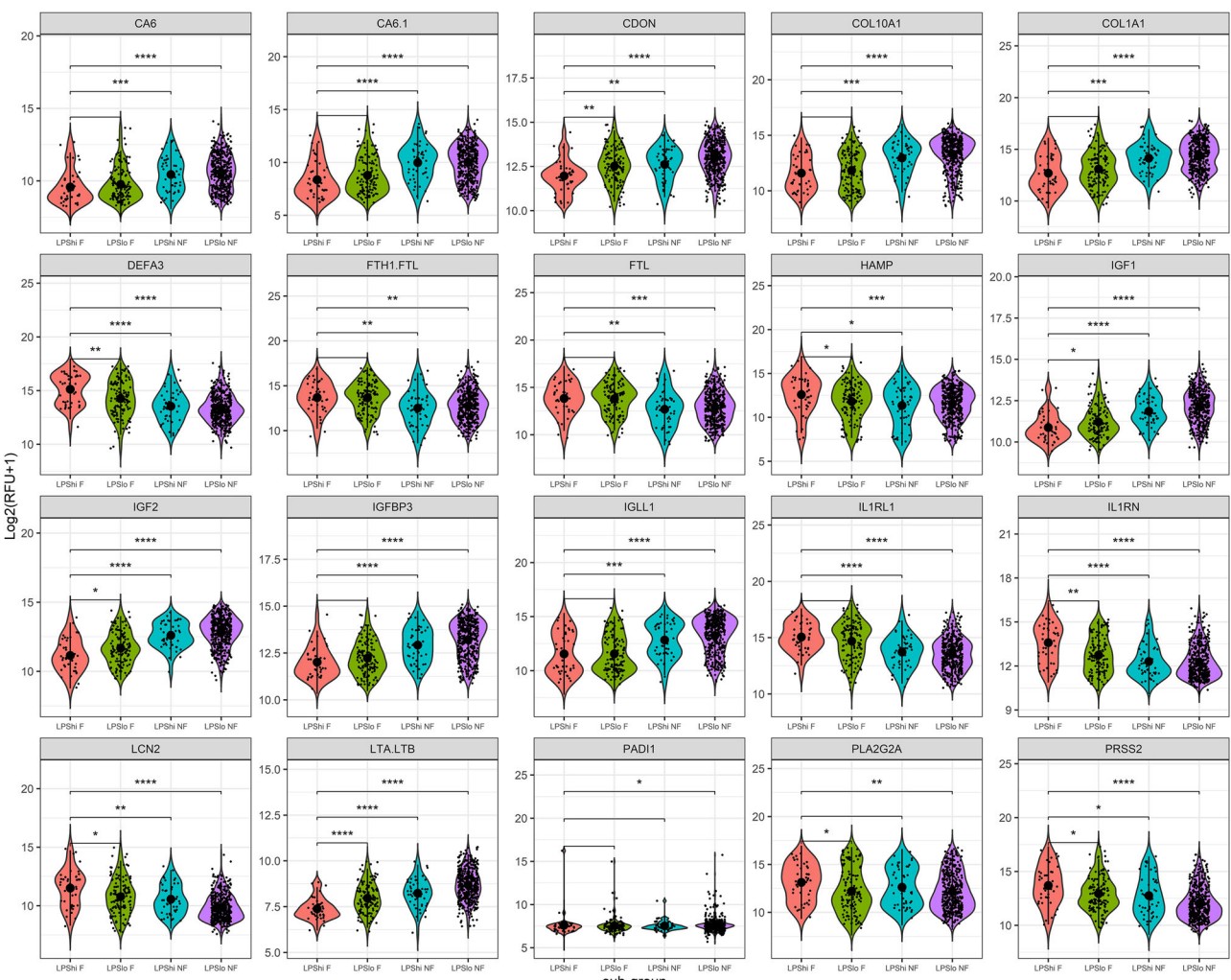

**Fig. 4 | Top differentially expressed proteins in the *LPShi Fatal* sub-group.**
Proteins of the insulin-like growth factor nutritional axis, inflammation and collagen regeneration are the top 10 up or downregulated proteins in the *LPS^hi F* subgroup, as identified by log2 fold change of expression between *LPS^hi F* subgroup ($n = 39$) and community children ($n = 251$). All proteins had significantly different expression (two-tailed $p < 0.05$) as assessed by the Wilcoxon signed-rank test and were Holm-adjusted for multiple comparisons. There was a significantly different expression of these 20 plasma proteins in the *LPS^hi F* compared to other LPS sub-

groups. The *x*-axis denotes the LPS sub-groups, while the *y*-axis represents protein expression in units of Log2 transformed (RFU + 1). Central tendencies are shown as large dots representing mean ± standard deviation (SD), while individual values for each child are indicated by small dots. Models adjusted for sex, age, site, and HIV status confirmed differential expression for the majority of identified proteins (Supplementary Tables 18 and 19). RFU relative fluorescence units. All *n* values refer to independent children; no technical replicates were used. F fatal, NF non-fatal.

increased IL-10 and VEGF signaling, reduced IL-15 production, enhanced PI3K/Akt signaling and reduced regulation by PTEN, we infer that an immune-dysregulation is likely associated with LPS-correlated mortality (Fig. 7C).

## Discussion

We assessed the associations of plasma LPS and mortality in a cohort of acutely ill hospitalized children in a setting where EED, enteric and systemic infections and malnutrition are prevalent. We demonstrated that plasma LPS levels are associated with mortality. Our study underscores the complex nature of mortality in EED-prevalent settings and its contextual links to plasma LPS levels, and it is supported by a substantial sample size of the population at risk, a control group of children from the same community as the hospitalized cohort, and plasma LPS, proteomic and stool biomarker analysis.

We demonstrate that while wasting status was linked to increased mortality, LPS-related mortality is an independent and synergistic factor. Since chronic intestinal inflammation is widespread in this

cohort and intestinal and systemic inflammatory biomarkers were also independently associated with mortality, our findings emphasize the importance of targeting immunological pathways as a potential strategy to reduce mortality. These pathways illustrate the intersection between inflammation, metabolism (IGF nutritional axis), and tissue remodeling (collagen regeneration processes). Our findings connect LPS signaling, severity of anthropometric classification, inflammation and tissue integrity as key factors associated with mortality. This study confirms that wasting affects mortality in EED-enriched settings but also identifies the biological context in which elevated LPS and other inflammatory biomarkers drive mortality risk.

LPS is a reliable indicator of 90-day survival in the context of heterogeneous presentation of comorbidities. While this study did not definitively ascertain the source of elevated LPS, it did offer some clues. Acutely ill hospitalized children had significantly higher quantification of 16S rRNA, *E. coli* strains, *V. cholerae* and CTX-M-producing bacterial qPCR in stool (Table 1) compared to community children, suggesting differential colonization. However, similar plasma LPS

**Table 2 | Key differences in pathogen quantification and inflammatory biomarker expression between LPS-mortality sub-groups**

| Characteristic | LPS$^{hi}$ Fatal Weighted N = 34 [a]Unweighted n = 39 | LPS$^{hi}$ Non-fatal Weighted N = 145 [a]Unweighted n = 46 | LPS$^{lo}$ Fatal Weighted N = 98 [a]Unweighted n = 125 | LPS$^{lo}$ Non-fatal Weighted N = 1609 [a]Unweighted n = 425 | p-Value[b] |
|---|---|---|---|---|---|
| Plasma LPS (E.U./ml) (Mean (SD)) | 16.7 (4.6) | 17.3 (4.6) | 3.6 (3.2) | 3.0 (3.1) | <0.001 |
| Anthropometric classification | | | | | <0.001 |
| Acute A (SAM) | 18/34 (52%) | 48/145 (33%) | 64/98 (66%) | 341/1609 (21%) | |
| Acute B (MAM) | 5/34 (15%) | 29/145 (20%) | 14/98 (14%) | 258/1609 (16%) | |
| Acute C (Normal) | 11/34 (33%) | 68/145 (47%) | 20/98 (20%) | 1009/1609 (63%) | |
| Admission age (Mean (SD)) | 10.2 (5.6) | 10.8 (4.5) | 11.0 (5.6) | 11.6 (5.6) | 0.5 |
| Sex | | | | | 0.008 |
| Female | 19/34 (56%) | 83/145 (57%) | 47/98 (48%) | 591/1609 (37%) | |
| Male | 15/34 (44%) | 62/145 (43%) | 51/98 (52%) | 1017/1609 (63%) | |
| HIV status | | | | | <0.001 |
| Negative | 31/34 (90%) | 139/145 (96%) | 86/98 (88%) | 1579/1609 (98%) | |
| Positive | 3/34 (9.9%) | 5/145 (3.6%) | 11/98 (12%) | 30/1609 (1.8%) | |
| *Enteropathogen quantification[c]* | | | | | |
| Enteroaggregative *E. coli* | 25.8 (5.8) | 30.5 (5.9) | 27.5 (5.9) | 28.8 (6.4) | 0.001 |
| mphA Azithromycin Resistance | 21.2 (5.2) | 26.7 (7.1) | 21.4 (5.9) | 22.9 (6.4) | <0.001 |
| 16S rRNA | 12.2 (1.9) | 14.0 (3.5) | 13.0 (3.0) | 14.0 (3.3) | <0.001 |
| Rotavirus | 31.9 (5.7) | 33.2 (4.7) | 34.5 (2.3) | 32.9 (4.3) | <0.001 |
| *Fecal biomarkers of EED (mean (SD))* | | | | | |
| Fecal CAL | 852.4 (1210.2) | 328.3 (462.0) | 895.7 (1387.5) | 586.3 (951.8) | 0.003 |
| *Barrier function biomarker (mean (SD))[d]* | | | | | |
| Plasma zonulin | 33,736.2 (32,243.8) | 38,441.4 (30,298.5) | 42,406.5 (35,107.7) | 47,717.2 (29,380.6) | 0.021 |
| Plasma diamine oxidase | 3377.3 (3205.3) | 4433.3 (2628.9) | 4808.8 (12,543.9) | 4834.4 (3252.7) | <0.001 |
| Plasma CDH1.1 | 955.5 (409.8) | 1130.3 (474.6) | 1278.8 (1012.4) | 1521.2 (2872.5) | 0.007 |
| Plasma CDH1.2 | 22,900.8 (8445.9) | 27,482.1 (7827.2) | 23,456.4 (8073.5) | 25,908.1 (6163.6) | 0.004 |
| Plasma OCLN | 1053.7 (386.2) | 1012.9 (520.2) | 1029.2 (331.7) | 949.9 (254.8) | 0.008 |
| Plasma ZO-1 | 446.2 (126.4) | 621.5 (615.5) | 518.0 (510.9) | 746.7 (1,846.2) | <0.001 |
| Plasma TNFAIP3 | 1459.9 (451.2) | 1305.2 (306.8) | 1627.1 (1337.0) | 1657.9 (1059.6) | 0.003 |
| *Systemic immune activation and inflammation biomarkers (Mean (SD))[d]* | | | | | |
| Plasma heparin-binding protein | 31,240.7 (10,668.0) | 26,322.3 (13,020.1) | 29,175.7 (19,256.3) | 23,707.3 (10,530.0) | <0.001 |
| Plasma TREM1 | 4411.7 (4014.6) | 2601.7 (1686.2) | 2908.0 (2404.2) | 2037.7 (1177.0) | <0.001 |
| *Systemic inflammation (Mean (SD))[d]* | | | | | |
| Plasma MPO | 35,788.1 (20,177.6) | 24,180.2 (12,891.4) | 34,692.0 (22,355.8) | 27,326.2 (13,461.8) | 0.002 |
| *LPS signal transduction (Mean (SD))[d]* | | | | | |
| Plasma CD14 | 2397.6 (1344.7) | 1607.0 (605.7) | 2364.8 (1620.1) | 1673.0 (617.4) | <0.001 |
| Plasma LBP | 77,694.5 (40,981.5) | 53,099.1 (31,694.3) | 76,151.9 (38,968.7) | 67,794.0 (33,009.2) | 0.001 |
| Plasma TLR4 | 379.6 (176.7) | 438.6 (196.9) | 673.4 (1266.3) | 511.7 (1053.9) | 0.019 |

[a]N = inverse proportionally weighted total within CHAIN cohort (n = 3101); [b]Design-based Kruskal–Wallis test, Pearson's $X^2$: Rao & Scott adjustment; [c]Threshold Cycle (Ct) from quantitative PCR (qPCR) assay; [d]*RFU* relative fluorescence units; *E.U./ml* endotoxin units per ml; *SD* Standard Deviation.

Quantification of enteropathogens (16S rRNA, EAEC, *mphA* azithromycin resistance genes, and rotavirus) ($p \le 0.001$), fecal calprotectin ($p = 0.003$), plasma MPO ($p = 0.002$), biomarkers of barrier function (zonulin, diamine oxidase, CDH1, OCLN, ZO-1, and TNFAIP3) ($p \le 0.05$), immune activation (heparin-binding protein and TREM-1) ($p < 0.001$), and LPS signal transduction (CD14, LBP, and TLR4) ($p < 0.02$) revealed significant differences between LPS–mortality sub-groups in unadjusted comparisons.

Specifically, levels of 16S rRNA, EAEC, *mphA*, plasma diamine oxidase, zonulin, MPO, CD14, heparin-binding protein, TREM-1, and CDH1 differed significantly between the *LPS$^{hi}$ F* group and at least one other LPS–mortality sub-group in both unadjusted and adjusted (Supplementary Tables 14 and 15) analyses. CTX-M–producing bacteria were also detected at significantly higher levels for the *LPS$^{hi}$ F* vs. *LPS$^{lo}$ NF* sub-group in adjusted comparisons. Adjusted analyses controlled for admission age, sex, study site, and HIV status.

*F* Fatal, *NF* Non-fatal, *LPS* Lipopolysaccharide, *CAL* Calprotectin, *CDH1.1/CDH1.2* Cadherin-1, *OCLN* Occludin *ZO-1* Tight Junction Protein ZO-1, *TNFAIP3* Tumor Necrosis Factor Alpha-induced Protein 3, *TREM1* Triggering Receptor Expressed on Myeloid Cells 1 *MPO* Myeloperoxidase, *LBP* LPS Binding Protein, *TLR4* Toll-like Receptor 4.

levels between the cohorts indicate that hospitalization-related enteric bacterial infection is not the sole source of elevated plasma LPS. Further studies are needed to evaluate the impact of diet, dysbiosis, liver impairment, and antibiotic use on plasma LPS concentrations. But a role for leaky gut is supported by expression of fecal biomarkers like calprotectin, plasma barrier function markers like zonulin, ZO-1

(Table 1), as well as inhibition of RAC signaling and cell junction organization pathways (Fig. 7A) in *LPS$^{hi}$ F* group. Thresholds of EED biomarkers in this setting would benefit from systematic reviews and meta-analyses that adjust for key confounders such as age, HIV status, and diarrhea. Previous studies have employed composite EED scores to standardize EED assessment[27,28], and while valuable, these scores are

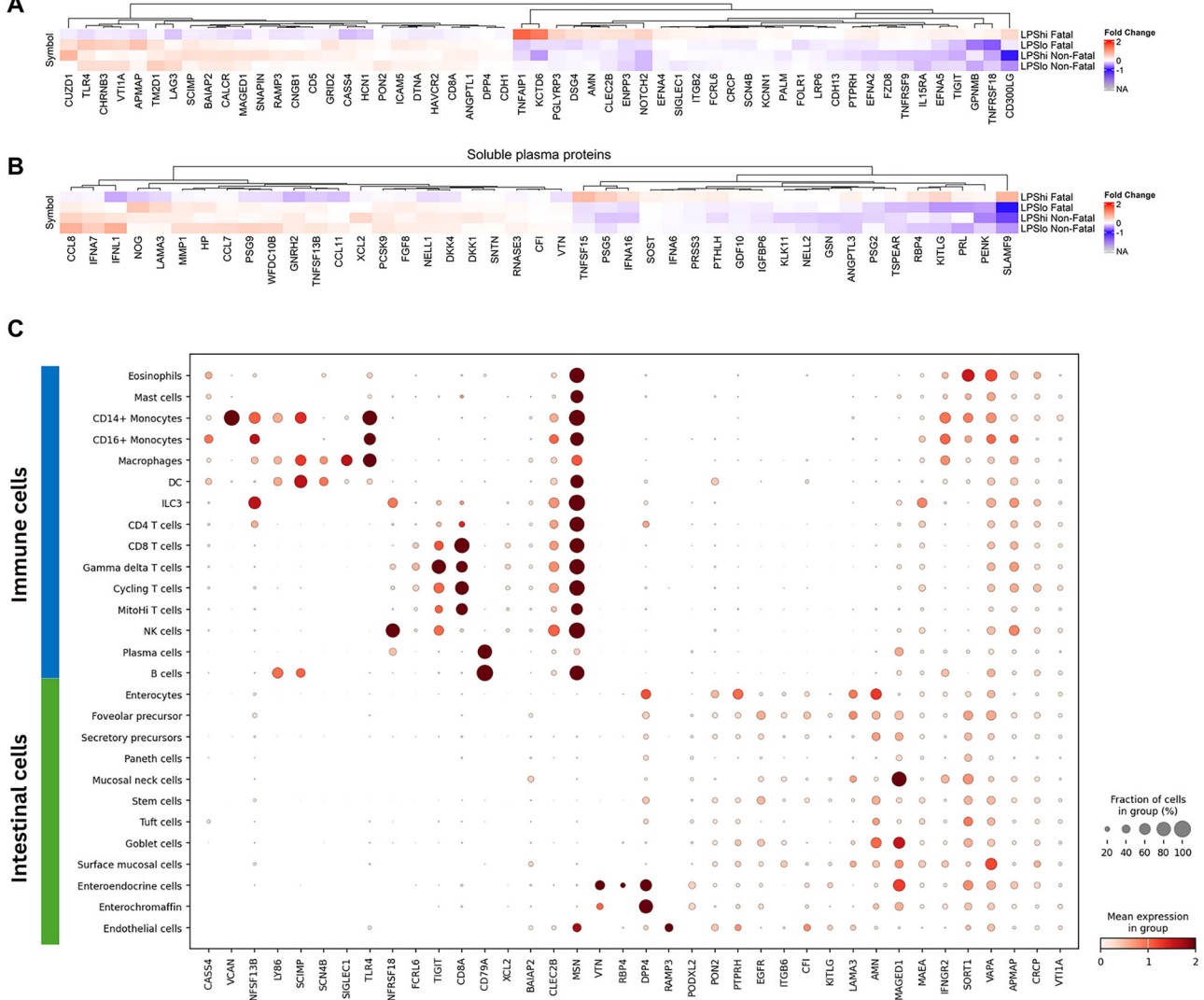

**Fig. 5 | Differentially abundant plasma proteins in *LPS^hi^ Fatal* cohort indicate a cross-talk between immune and intestinal cell types.** Evaluation of proteomic expression of LPS sub-groups (*LPS^hi^ F (n = 39), LPS^lo^ F (n = 125), LPS^hi^ NF (n = 46)* and *LPS^lo^ NF (n = 425)*) identified proteins that were differentially abundant in the *LPS^hi^ F* sub-group. All analyses of LPS sub-groups were performed against identical community children (*n = 251*). Heatmap visualizes **A** membrane-derived or **B** soluble plasma proteins, where directionality of Log2 fold change (vs. community children) was opposite in the *LPS^hi^ F* sub-group compared to the other 3 sub-groups.

**A, B** Differentially abundant proteins had significant differences in expression (two-tailed *p* < 0.05) in at least one LPS sub-group. **C** Dot plot of the genes that encode membrane-derived and soluble plasma proteins with differential directional expression in the *LPS^hi^ F* group (vs. community children) compared to expression in the other 3 groups (*LPS^hi^ NF, LPS^lo^ F and LPS^lo^ NF*) (vs. community children). All *n* values refer to independent children; no technical replicates were used. F fatal, NF non-fatal.

often cohort-specific, relying on the internal percentiles of each biomarker. There is a need for more appropriate EED biomarker thresholds, particularly for milder levels of intestinal inflammation and/or permeability, to better reflect conditions in EED settings. Standardizing summary statistics and adjusting for these variables in future research would improve comparability across studies and enhance our understanding of EED biomarkers in community settings. Pathway analysis of sub-groups indicated a distinct upregulation of the VEGF pathway in the *LPS^hi^ F* sub-group, suggesting a potential direct mechanism[29] for leaky gut in this context (see Fig. 8, panel A).

In the multifactorial settings of this study, plasma LPS acts in synergy with acute comorbidities to indicate the increased risk of mortality (see Fig. 8, panel E). This implies that inflammation due to elevated plasma LPS levels is typically regulated within community children due to chronic exposure[30], but the equilibrium might be disrupted, leading to immune dysregulation when comorbidities, such as acute intestinal infections, are present. Short-term LPS–TLR4 activation results in a complex systemic signaling response, but low dose or chronic LPS can induce long-term immunomodulatory effects[31,32]. This can be mediated either via epigenetic imprinting in various immune cells or via the induction of endogenous TLR4 ligands, such as S100A8 and S100A9. Our proteomics data indicate *LPS^hi^ F* sub-group has lower soluble TLR4 level than the others. Whether sTLR4 acts as a molecular sink for the excess plasma LPS or the lower sTLR4 are inherent to these subjects, thus leading to insufficient neutralization of plasma LPS in this group, is not clear. These mechanisms ensure host protective immunity while preventing chronic detrimental inflammation. In fact, the elevation of plasma AAT in the admissions cohort (vs. community children) may represent LPS immunomodulation since AAT has been shown to inhibit IL-8 production and LPS-induced NF-κB activation. This trend needs to be confirmed in a setting where diarrhea is not prevalent to account for dilution effects on the measured

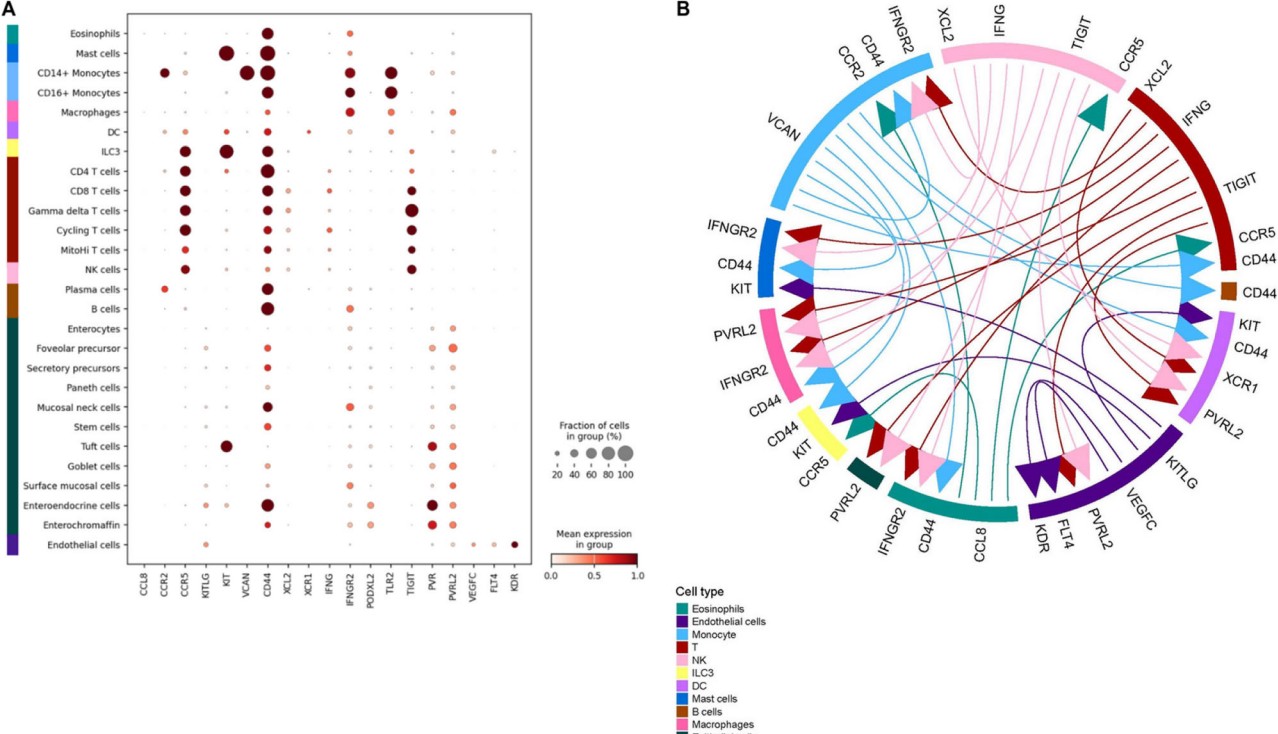

**Fig. 6 | Differentially abundant proteins in *LPS^hi Fatal* cohort indicate a cross-talk between immune and intestinal cell types.** The expression of selected differentially abundant proteins in *LPS^hi F* cohort (*n* = 39) was deconvoluted using the publicly available single-cell dataset. **A** Dot plot indicating transcript-level expression of immunologically relevant differentially abundant proteins in *the LPShi F* cohort and their binding partners, in a public single-cell transcriptomic dataset. **B** Possible ligand–receptor interactions in the *LPS^hi F* cohort that might lead to the increased mortality in the subgroup. F fatal.

stool biomarker content. Other candidates for LPS immunomodulation include S100A6 and SFTPD since they have been shown to regulate LPS-induced inflammation[33–35] and are correlated with plasma LPS and LPS signaling proteins in the *LPS^hi F* sub-group in our cohort (Supplementary Fig. 5G). Apart from TLR4, TRPA1 and TRPV1 are transmembrane ion channels that are activated by LPS and have been linked with inflammation in nerve ending nociceptors and iPSC-derived cardiomyocytes respectively[36,37]. Whether similar non-canonical LPS receptors or LPS-activated channels exist in the intestinal epithelium requires further investigation.

Alongside VEGF upregulation and reduced growth hormone signaling (Fig. 8, panels A and B), comparative pathway analysis suggested that the *LPS^hi F* cohort exhibited cytokine dysregulation, including increased IL-10 signaling, which could be correlated to reduced IL-12 and IL-15 signaling and increased anti-inflammatory M2 macrophage polarization (Fig. 8, panel C). This was accompanied by increased STAT3, PI3K/AKT and inhibited PTEN pathways (Fig. 8, panel D). These together suggest immune dysregulation, which we hypothesize may contribute to increased mortality in the study settings (Fig. 8, panel E). The IL-10 and VEGF pathways are known to activate PI3K/AKT signaling, which was also differentially upregulated in *LPS^hi F* cohort. Coupled with inhibition of PTEN, which inhibits the PI3K/AKT pathway, this axis may play a crucial role in the mortality mechanism of some of the children studied in this cohort (Fig. 8, panel D). Although these findings are preliminary, they point to the PI3K/Akt pathway as a potential therapeutic target to be explored. Since this study did not assess treatment courses, it cannot determine how treatment may have influenced mortality outcomes. Randomized clinical trials are needed to validate these findings.

Our data suggest that plasma LPS is an independent indicator of mortality. Using the Maxstat method, we identified a novel, unweighted cut-off of 11.58 E.U./ml for plasma LPS, which stratified 90-day survival in the admission population. While this finding provides an initial reference point for LPS-associated pathophysiology, the plasma LPS threshold identified is likely specific to this population, influenced by its unique comorbidity profiles. Notably, bootstrapping analysis (Supplementary Fig. 1B) revealed four distinct clusters of LPS cut-points, suggesting the existence of sub-populations with varying mortality risk profiles linked to LPS levels.

Further standardization and clinical validation in diverse cohorts are needed to assess its broader applicability. To translate this finding into clinical practice, it is essential to develop accurate clinical diagnostic assays for detecting plasma LPS and implement protective clinical interventions aimed at reducing its impact. From a therapeutic perspective, it needs to be shown whether blocking of an LPS-associated immune response can attenuate adverse immune effects and reduce mortality during acute illness or whether the observed immune dysregulation is a (insufficient) host protective immune response.

## Methods
### Ethics
The CHAIN protocol was approved by participating sites' ethics committees and the Oxford Tropical Research Ethics Committee (OxTREC), University of Oxford[38]. Recruitment involved assessing criteria, approaching eligible caregivers, discussing the study, and obtaining written informed consent, with compensation provided in line with institutional or national guidelines.

### CHAIN cohort
The Childhood Acute Illness and Nutrition (CHAIN) Network aims to improve child survival and growth by understanding and identifying intervenable mechanisms leading to death and poor growth among acutely ill children in sub-Saharan Africa and South Asia. The CHAIN

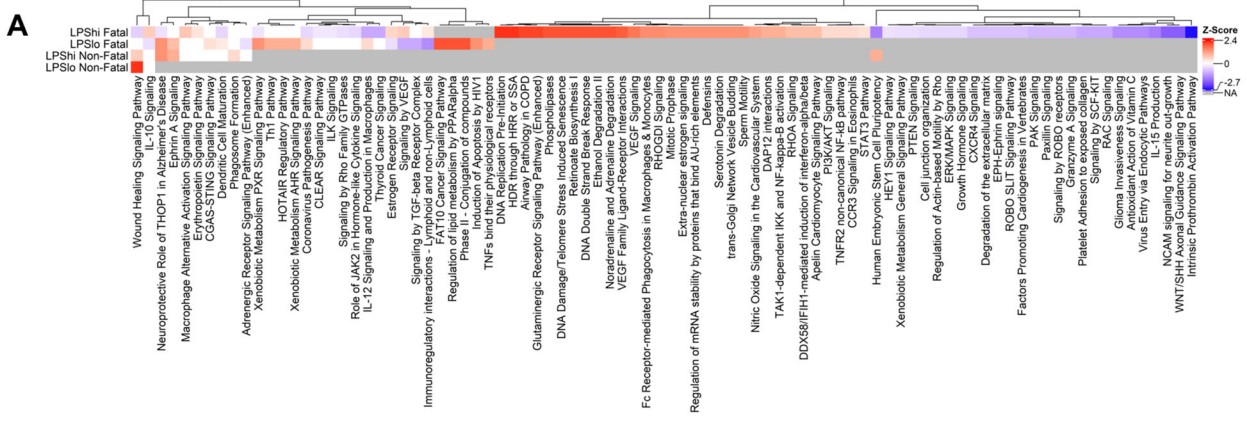

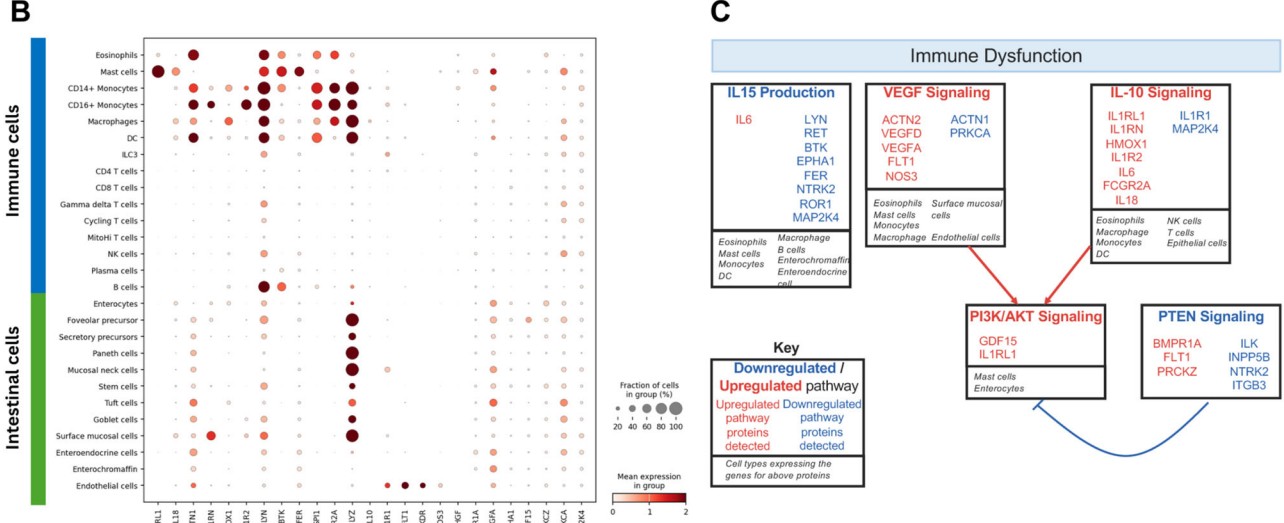

**Fig. 7 | Differentially regulated pathways in *LPS^hi Fatal* cohort and proposed mechanism of LPS-associated mortality.** Comparative IPA pathway analysis of LPS sub-groups (*LPS^hi F, LPS^lo F, LPS^hi NF and LPS^lo NF*) identified pathways that were differentially activated or inhibited. All analyses of LPS sub-groups were performed against an identical group of community children (n = 251). **A** Heatmap visualizes z-scores of canonical pathways that were significantly activated (red) or inhibited (blue) in the *LPS^hi F* analysis but directionally differentially regulated in the analyses of the other 3 LPS groups. **B** Dot plot of genes of proteins involved in the regulation of select immunological pathways differentially regulated in (**A**) and their binding partners determined using CellphoneDB, deconvoluted in a public single-cell transcriptomic dataset. **C** Proposed mechanism of immune dysregulation in *LPS^hi F* cohort that may lead to increased mortality. Red font: upregulated pathway/protein level, blue font: downregulated pathway/protein level. Red arrows: activation; blue blunt-head arrows: inhibition of the pathway. F Fatal, NF non-fatal.

cohort study (ClinicalTrials.gov ID: NCT03208725) was conducted between November 2016 and January 2019 at nine hospitals in Africa and South Asia: Dhaka and Matlab Hospitals (Bangladesh), Banfora Referral Hospital (Burkina Faso), Kilifi County, Mbagathi County and Migori County Hospitals (Kenya), Queen Elizabeth Hospital (Malawi), Civil Hospital (Pakistan), and Mulago National Referral Hospital (Uganda)[39]. The results from the CHAIN cohort mortality and post-discharge growth have recently been described[38,40]. The CHAIN Network conducted a Nested Case-Cohort (NCC) that utilized systematically collected samples and data from the CHAIN cohort to investigate biological mechanisms leading to inpatient and post-discharge mortality through an integrated multi-omics approach[21]. CHAIN NCC includes a random sub-cohort of 24% of the CHAIN cohort, plus all additional deaths for a total of 1008 patients. Additionally, 30 community children from each site, for a total of 270 community participants, were included. CHAIN NCC-generated datasets used in this analysis include plasma LPS, SomaScan plasma proteomics, TaqMan Array Card (TAC) pathogens from fecal

swabs, and stool levels for myeloperoxidase (MPO), alpha-1-antitrypsin (AAT), and calprotectin (CAL: S100A8/S100A9 heterodimer). Plasma samples and LPS levels were available from 638 cohort children at admission, including 199 children that died during the study period, 439 survivors and 251 community children (Supplementary Methods 6). Sex and age distribution of admission, community children and LPS-mortality sub-groups are presented in Tables 1 and 2. Sex was considered in relevant survival analyses as an adjustment covariate and was additionally included as a covariate in sensitivity analyses of biomarker expression.

## Patient samples
Blood, stool, and fecal swabs were collected at admission, discharge, and scheduled follow-up time periods from subjects or at a single community participant visit. They were processed and frozen at −80 °C on-site, then sent to a central biorepository in Kenya, then to consortium labs (e.g., University of Oxford) and industry partners (e.g., SomaLogic Inc.) for analysis. Further handling details are covered in

**Fig. 8 | Proposed Mechanism of the interplay of chronic intestinal inflammation and acute infection.** Intestinal inflammation and increased barrier permeability were prevalent in the community, as measured by fecal EED biomarkers. Comparative pathway analysis identified differentially regulated pathways supporting **A** exacerbated barrier dysfunction and **B** impaired growth (stunting) among hospitalized children. **C** Inflammatory and immunoregulatory protein expression was associated with pathways suggesting a shift toward cytokine dysregulation, and **D** broader disruption of signal transduction pathways. **E** Together, these findings suggest that LPS-associated pathophysiology in this setting likely involves immune dysfunction driven by chronic intestinal inflammation and its interaction with acute infections. Created in BioRender. Allen, C. (2025) https://BioRender.com/w25l747.

the CHAIN nested case-cohort study protocol[21]. This study included data for the time of admission and discharge for cohort participants and from community children.

## Plasma protein, LPS and EED markers analysis

Details regarding sample acquisition and analysis protocols are provided in the CHAIN nested case cohort study protocol[21] and Supplementary Methods 1. Briefly, Plasma LPS levels were measured using an optimized chromogenic Limulus amebocyte assay (ThermoFisher, UK). The aptamer-based SomaScan assay with a 7596 SOMAmers panel was used to quantify 7288 protein analytes in plasma (SomaLogic, USA). Stool myeloperoxidase (MPO), alpha-1-antitrypsin (AAT), and calprotectin (CAL) levels were quantified using ELISA (Immundiagnostik AG, Germany). The absolute concentration of plasma LPS and stool EED biomarkers was calculated using dose-response curves. Fecal CAL concentrations were reported for children older than 12 months, as levels may fluctuate significantly in younger infants due to factors such as breastfeeding. Fecal AAT concentrations were reported for children without acute diarrhea, since 61% of CHAIN admissions children presented with acute diarrhea (Table 1) and diarrheal illness may obscure quantification through dilution effects and protein loss.

**Enteropathogen quantification.** The Taqman Array Card (TAC, Life Technologies, USA) assay was used for the detection and quantification of 36 enteropathogens, 2 antimicrobial resistance genes and non-specific bacterial load. The TAC panel included the detection of:

**Bacteria:** *Aeromonas* spp., *Campylobacter* pan, *Campylobacter jejuni*, Enteroaggregative *Escherichia coli* (EAEC), atypical Enteropathogenic *E. coli* (aEPEC), Enteropathogenic *Escherichia coli* (EPEC), Enterotoxigenic *E. coli* producing heat-labile toxin (LT) (LT-ETEC), Enterotoxigenic *E. coli* producing heat-stable toxin (ST) (ST-ETEC), typical Enteropathogenic *E. coli* (tEPEC), Enterotoxigenic *Escherichia coli* (ETEC), *Mycobacterium tuberculosis*, Shiga toxin-producing *Escherichia coli* (STEC), *Helicobacter pylori*, *Plesiomonas shigelloides*, *Salmonella* spp., *Shigella* spp., *Vibrio cholerae*, *Clostridium difficile*.

**Total bacterial load:** 16S rRNA gene (X16s) of predefined bacterial pathogen

**Antimicrobial resistance (AMR) genes:** Beta-lactam (ESBL) resistance: *blaCTX-M*, Macrolide (Azithromycin) resistance: *mphA*

**Viruses:** Adenovirus, Astrovirus, Norovirus GI, Norovirus GII, Rotavirus, Sapovirus

**Parasites:** *Ancylostoma, Ascaris lumbricoides, Cyclospora cayetanensis, Cryptosporidium* spp., *Entamoeba histolytica, Enterocytozoon bieneusi, Enterocytozoon intestinalis, Giardia lamblia, Isospora belli, Necator americanus, Strongyloides stercoralis, Trichuris trichiura*.

## Cohort-specific analyses

CHAIN LPS study participants were categorized into cohorts and subgroups based on their demographic variables, health behaviors, comorbidities or survival outcomes at hospitalization:

**Demographic variables.** Binary sex (parent-reported: male, female) and continent (Africa, Asia) variables were recorded at hospital admission, while age (continuous) and categorical age group (<6 months, 6–11 months, ≥12 months) were captured at each time point.

**Breastfeeding status.** Given its association with EED[41], caregiver-reported breastfeeding status was captured at hospital admission.

**Anthropometric classification.** In the CHAIN cohort, anthropometric classification at each time point was assessed using mid-upper arm circumference (MUAC) and age-specific cutoffs. MUAC is a simple and reliable measure of acute malnutrition[42], is useful for screening in therapeutic feeding programs[43], and aligns with WHO guidelines for the management of acute malnutrition[44].

Anthropometric classification at hospitalization:

Severe acute malnutrition (SAM): MUAC < 11.5 cm for children aged ≥6 months, MUAC < 11 cm for children aged <6 months, or the presence of bilateral pitting edema.

Moderate acute malnutrition (MAM): MUAC 11.5 to <12.5 cm for children aged ≥6 months, or MUAC 11 to <12 cm for children aged <6 months.

Normal: MUAC ≥ 12.5 cm for children aged ≥6 months, or MUAC ≥ 12 cm for children aged <6 months.

Non-SAM: MUAC > 11.5 cm was used where age-specific cutoffs were not employed.

**Comorbidities.** Children hospitalized at admission were categorized into clinically relevant comorbidities based on common childhood illnesses outlined in the "Pocket Book of Hospital Care for Children"[45]. Comorbidities were defined as follows:

- Acute diarrhea or gastroenteritis (yes, no): clinical examination or caregiver report of acute diarrhea (<14 days prior to hospitalization) or clinician-diagnosed gastroenteritis. Children who presented with bloody diarrhea were excluded from this subgroup categorization.
- Lower respiratory tract infection (LRTI) (yes, no): clinician-diagnosed LRTI or pneumonia.
- Upper respiratory tract infection (URTI) (yes, no): clinician-diagnosed URTI.
- Malaria (yes, no): clinician-diagnosed malaria or a positive diagnostic test.
- Sepsis (yes, no): clinician-diagnosed sepsis or evidence of acidotic breathing.
- HIV status (positive, negative):

Positive: positive PCR test, positive antibody test, or known HIV-exposed and tested.
Negative: negative test results or untested status.

## Statistical analysis

Statistical and bioinformatic analysis for this study, including descriptive analysis, regression, correlation, survival analysis and adjustment for multiple comparison, was primarily performed using the R statistical software (Supplementary Methods 3). All statistical tests were two-tailed unless otherwise specified. For analyses involving TAC data, a threshold cycle (Ct) cutoff of 35 was used to denote positivity for all the targets, and Ct was used for regression and correlation models.

## Survival analysis

Survival analyses assessed inpatient, acute (30-day), short-term (90-day) and overall mortality, with a primary focus on 90-day mortality (Supplementary Methods 4).

**Kaplan–Meier survival curves.** Unweighted Kaplan–Meier estimators were used to create survival curves for evaluating 90-day and overall mortality differences. Survival curves employed the log-rank test to assess survival differences among strata at all-time points (null hypothesis) and incorporated risk tables documenting at-risk participants and cumulative censored cases.

**Cox proportional hazard (CoxPH) models.** Variable selection for IPW CoxPH models included demographic (e.g., age) and comorbidities (e.g., anthropometric classification) and biomarkers. Plasma LPS concentration was included in models as a key predictor to examine associations with mortality. Predictor variables were standardized to ensure comparability of scales. Mortality risks were evaluated through estimations of hazard ratios (HR) while accounting for censored data. A significance level of 0.05 was employed for all tests. Inverse proportional weighting was incorporated in the analysis. The proportional hazard assumption for all model variables was assessed. The estimated survival probabilities and time to death were visualized, and CoxPH statistics were also provided.

**Elastic net regression.** The CHAIN cohort collected 200+ clinical features per patient at admission. To identify relevant features for 90-day mortality, an elastic net regression was employed. Training (70%) and testing (30%) datasets were generated with reproducible splitting using randomly assigned 'set.seed(88)'. Both models used the "cox" family. K-fold cross-validation utilized the 'cv.glmnet' function, adopting the training's best lambda ($\lambda = 0.052$) for testing. Inverse proportional weights were incorporated in the analysis.

**Inverse proportional weighting.** Where appropriate, inverse proportional weighting (IPW) was incorporated into analyses to account for selection bias and allow for the generalization of findings to the entire CHAIN cohort. IPW was utilized for summary statistics of sub-groups,

CoxPH models, Elastic Net Regression and correlation matrices of proteomics data. Generally, weights were calculated to account for sub-sampling by taking the inverse of the proportion of participants included in the LPS analysis relative to the full CHAIN cohort:

$$W = 1/(n_{\text{LPS analysis}}/n_{\text{CHAIN cohort}}).$$ See Supplementary Methods 5 for details of weight calculations.

**Covariate adjustments.** Survival analyses were adjusted for demographic factors and clinically relevant comorbidities to account for potential confounding. Specifically, CoxPH models included adjustments for age, sex, and site (demographic variables), as well as HIV status, anthropometric classification, malaria, gastroenteritis, LRTI, and URTI (clinical comorbidities), based on their known associations with both exposures and outcomes. For summary statistics, correlation and pathway analysis of proteomics data, clinical comorbidities were not adjusted for, as their roles in the causal framework are likely complex—potentially acting as mediators, confounders, or colliders. Given this ambiguity, adjustment could introduce bias or obscure meaningful biological associations. However, sensitivity analyses using linear models of principal components (PC1 and PC2) demonstrated that plasma LPS remained independently associated with significant variance in the proteomic data, even when key comorbidities were mutually adjusted (Supplementary Fig. 8, Supplementary Table 24). Generalized models were also used in most cases to demonstrate that trends persisted after adjustment.

**Differential protein expression**
To ensure accurate assessment of biological pathways without potential bias in enrichment results, enrichment analyses and single-cell transcriptomic data were analyzed using unweighted data. Mean comparisons of plasma protein abundance between sub-groups were performed using the Wilcoxon test with Holm-adjusted $p$-values.

**Pathway enrichment analysis**
Detected plasma proteins (log2 fold change of ≥1 and adjusted p-values < 0.05 compared to community children) were used as input for pathway enrichment analysis in the Ingenuity pathway analysis (IPA, Qiagen). Identical control populations were used in all IPA analyses, allowing for comparison among sub-groups.

**Signature deconvolution in single-cell transcriptomics data**
To unravel the cellular and molecular correlates of EED, differentially abundant membrane or extracellular space proteins were deconvoluted using a published single-cell transcriptomics dataset of small intestinal biopsies from adults with and without EED from Zambia and the United States, respectively[26]. This allowed deciphering of the potential cell types expressing these proteins. The interacting partners of differentially expressed proteins were enlisted using CellphoneDB (https://www.cellphonedb.org) and similarly deconvoluted using the single-cell dataset.

**Reporting summary**
Further information on research design is available in the Nature Portfolio Reporting Summary linked to this article.

## Data availability
The data that support the findings of this study are archived on the Harvard Dataverse (https://doi.org/10.7910/DVN/EJA4F6). The data contain sensitive information about study participants and may include identifiers that could compromise confidentiality or lead to ethnic stigmatization. Access to these data requires submission of a formal request for consideration by our Data Governance Committee.

The data request form is available on the Harvard Dataverse link provided. Email completed data request form to the Data Governance Committee at dgc@kemri-wellcome.org. The requester provides investigators details, variables requested, intended use of the dataset, potential risks of the study including risks to confidentiality of individuals or communities, potential benefits of the study including to participant communities, scientific capacity building or health policy and planned outputs (if analysis on dataset will result in publication or reports or presentations). The requester also needs to formally agree to the conditions and limitations for data sharing to avoid misuse of shared data.

## Code availability
The analytics files and codes that support the findings of this study are archived on the Harvard Dataverse (https://doi.org/10.7910/DVN/EJA4F6).

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

## Acknowledgements

We thank the CHAIN study, including the participants and their families, the study hospitals, and the communities within participating sites. The study was supported by the Bill and Melinda Gates Foundation grant OPP1131320/INV-003225 (The CHAIN Network), Wellcome Trust Intermediate Fellowship grant 222967/B/21/Z (JMN) and NIHR Biomedical Research Centre, Oxford (HHU). We thank Alex Adams and Charles Keown-Stoneman for advice, and William Patrick Porter for technical support. For the purpose of open access, the CHAIN Network has applied a CC BY public copyright license to any author-accepted manuscript version arising from this submission.

## Author contributions

J.M.N., L.G., K.J., K.D.T., J.D.W., J.A.B. and H.H.U. conceptualized the study. C.A.D.A., A.G., J.M.N., L.G., A.H.D., C.L., E.M., A.G., C.T., B.O.S., W.G., R.H.J.B., W.P.V., M.J.C., T.A., A.S.M.S.B.S., D.A., A.S., Z.K. and K.J. established the cohort and performed the experiments. C.A.D.A., A.G. and L.G. did data management. J.M.N., K.D.T., J.L.W. and J.A.B. acquired the funding. C.A.D.A. and A.G. performed data analysis and wrote the original draft. C.A.D.A., A.G., L.G., C.L., R.H.J.B., W.P.V., A.S.M.S.B.S., K.J., K.D.T., J.L.W., J.M.N., J.A.B. and H.H.U. reviewed and edited the paper.

## Competing interests

The authors declare no competing interests.

## Additional information

Chris A. D. Allen ®[1,26], Arya Ghate ®[1,26], James M. Njunge ®[2,3,26], Lisa Gartner ®[1], Abdoulaye Hama Diallo[2,4,5], Christina Lancioni ®[2,6], Ezekiel Mupere ®[2,7], Agnes Gwela[2,3], Caroline Tigoi[2,3,8], Benson O. Singa[2,9], Wilson Gumbi[2,3], Robert H. J. Bandsma ®[2,10,11], Wieger P. Voskuijl[2,11,12,13], Mohammod Jobayer Chisti ®[2,14], Tahmeed Ahmed[2,14], Abu Sadat Mohammad Sayeem Bin Shahid[2,14], Dilruba Ahmed[2,15], Ali Saleem[2,16], Zaubina Kazi[16], Kelsey Jones[17,18], Kirkby D. Tickell[2,19], Judd L. Walson ®[2,19,20,21,22], James A. Berkley ®[2,3,8,27] & Holm H. Uhlig ®[1,23,24,25,27] ✉

[1]Translational Gastroenterology Unit, University of Oxford, Oxford, UK. [2]The Childhood Acute Illness and Nutrition Network, Nairobi, Kenya. [3]Department of Biosciences, KEMRI-Wellcome Trust Research Programme, Kilifi, Kenya. [4]Department of Public Health, University Joseph Ki-Zerbo, Ouagadougou, Burkina Faso. [5]Department of Public Health, Centre Muraz Research Institute, Bobo-Dioulasso, Burkina Faso. [6]Department of Pediatrics, Oregon Health and Science University, Portland, OR, USA. [7]Department of Paediatrics and Child Health, Makerere University College of Health Sciences, Kampala, Uganda. [8]Center for Tropical Medicine and Global Health, University of Oxford, Oxford, UK. [9]Center for Clinical Research, Kenya Medical Research Institute, Nairobi, Kenya. [10]Centre for Global Child Health, Hospital for Sick Children, Toronto, ON, Canada. [11]Department of Paediatrics and Child Health, Kamuzu University of Health Sciences, Blantyre, Malawi. [12]Amsterdam UMC, location University of Amsterdam, Amsterdam Institute for Global Child Health, Emma Children's hospital, Meibergdreef 9, Amsterdam, the Netherlands. [13]Amsterdam UMC, location University of Amsterdam, Department of Global Health, Amsterdam Institute for Global Health and Development, Meibergdreef 9, Amsterdam, the Netherlands. [14]Nutrition Research Division, International Centre for Diarrhoeal Disease Research, Bangladesh (icddr, b), Dhaka 1212, Bangladesh. [15]Clinical & Diagnostic Services, International Centre for Diarrhoeal Disease Research, Bangladesh (icddr, b), Dhaka 1212, Bangladesh. [16]Department of Pediatrics and Child Health, Aga Khan University, Karachi, Pakistan. [17]UKPaediatric Gastroenterology, Great Ormond Street Hospital, London, UK. [18]Kennedy Institute, University of Oxford, Oxford, UK. [19]Department of Global Health, University of Washington, Seattle, USA. [20]Department of Epidemiology, University of Washington, Seattle, USA. [21]Department of International Health and Medicine, Johns Hopkins University, Baltimore, USA. [22]Department of Paediatrics, Johns Hopkins University, Baltimore, USA. [23]Department of Paediatrics, University of Oxford, Oxford, UK. [24]Biomedical Research Centre, University of Oxford, Oxford, UK. [25]Centre for Human Genetics, University of Oxford, Oxford, UK. [26]These authors contributed equally: Chris A. D. Allen, Arya Ghate, James M. Njunge. [27]These authors jointly supervised this work: James A Berkley, Holm H. Uhlig. ✉e-mail: holm.uhlig@well.ox.ac.uk

