## [Peer Review file · Nature Communications]

Plasma lipopolysaccharide levels predict mortality in acutely ill children in Low- and Middle-Income Countries

Corresponding Author: Professor Holm Uhlig

Version 0:

Reviewer comments:

Reviewer #1

(Remarks to the Author)

In the manuscript "Plasma lipopolysaccharide levels, microbiota and biomarkers of enteric dysfunction predict mortality in acutely ill children in sub-Saharan Africa and South Asia" by Allen and colleagues assessed the associations of plasma LPS and mortality in a cohort of acutely ill hospitalized children where environmental enteric dysfunction (EED), enteric and systemic infections and malnutrition are prevalent. They showed an association of LPS to mortality and chronic inflammation and immunology pathways as well as metabolism and tissue remodelling. The authors suggest that LPS signalling, nutritional status, inflammation and tissue integrity as are important factors linked to mortality in acutely ill children. Overall, the authors provide a thoroughly analysis on the association of plasma LPS and different factors including sex, age, nutritional, microbiota etc and show that LPS, inflammation, metabolism and remodelling appear to be involved in the mortality rate seen in acutely hospitalised children. The authors have identified some potential therapeutic venues targeting inflammation and metabolism which will need to be validated in other cohorts for future clinical studies. Below are a list of comments identified in the paper including the lack of data or information on the analyses used, that need to be addressed.

Major comments

Study of the nested case-cohort

When describing the results, it is necessary to provide some range values to consider the description of each feature. For example, the nutritional status (severe, moderate and normal) does not specify in the text what are the values and their range to classify each term (Supplementary Methods 5; and line 202 of the manuscript). Please include these values in the text and in the supplementary methods table. Please include the calculation of "CHAIN NCC weights" in Supplementary Methods 5 or in the Methods section. This could be convenient and easier to find for future readers of this study.

Comparing the selected cohort and community children

The study included the differences between the hospitalised children and the healthy community children. The authors did not find differences in gut inflammation markers such as myeloperoxidase and calprotectin, but faecal AAT seems to be higher in community children. This last result is not shown in the results section ("High proportion of EED in the CHAIN cohort"), nor in the discussion. Can the authors discuss what the reason for these differences might be?. Is it possible that the "health community" group has some characteristics associated with chronic EED? Please include in the discussion or results section how their data on the community group compare to other healthy groups in similar studies. Also, include the values of the "healthy community children" in the supplementary table 7. This could help and differentiate between the healthy group and the hospitalised group, and also consider the 'baseline' of these values in the community.

Lines 405 to 409: Please show these results. Supplementary Table 1 shows that there are significant differences between admission/discharge/community. But are these differences also between admission and community? If not, it might be interesting to show the 16s rRNA results to compare both communities and find a common genus of bacteria. Furthermore, studying the a-diversity or b-diversity after hospital intervention will enrich this study.

If there are no significant differences, this should be mentioned in the discussion and explain that community may be a chronic feature in EED. As previously stated, it is relevant to compare to other healthy controls from other studies to draw on how "healthy" the community group used in this study actually is.

A cellular model of LPS associated immunopathology

The authors have mapped the plasma proteomic analysis to unravel the interaction between the immune system and the epithelium. However, the results section lacks information on these results (lines 367 to 370), showing only some of the differentially abundant proteins without explaining if they were up- or down-regulated and from which immune cell type. Furthermore, it is necessary to show the highlighted proteins and cell type interaction from this study section to explain the LPS and immune-immune and immune-epithelial interaction. These comments are also relevant to Figure 7.

Discussion

In lanes 404-13, the authors offer some explanations to the source of LPS, as it was not identified in this study. The authors indicate that "leaky gut" might be one possible factor based on fecal marker calprotectin. Although calprotectin is a marker of inflammation (neutrophils/macrophages), other markers could have been examined in the plasma e.g. zonulin, lipopolysaccharide-binding protein (LBP) and intestinal fatty acid-binding protein (I-FAB). The authors could also mention that reduction in pathways associated with cell junction organisation and RAC signalling (identified in the IPA analysis in Fig 8A) as supportive data to their speculation on leaky gut.

Another question is whether the significance is underestimated when comparing the diseased children against the healthy community population, as these also had high levels of LPS. This could have been addressed more in the discussion. How would other studies agree with the findings in this study? A short discussion on potential association or not with other studies should be included.

Minor comments

- Line 96: Considering the reference mentioned by the authors, the change is a shift from a pro-inflammatory effect and stimulation to a resolving and anti-inflammatory response. Given this, what is the significance or relevance to this study?
- Line 134-141: In general, the number of children included in the different parts of the study are difficult to follow, especially when looking at the supplementary tables. For example, in Suppl Table 4, there seem to be 1894 samples tested for LPS, while in the text it is stated in lines 139-141: "Plasma samples and LPS levels were available from 889 children including 199 children that died during the study period, 439 survivors and 251 community children"
- Line 202: Please add a range of values for nutritional status or more examples of relevant comorbidities in the study.
- Lines 279 and 280. Please add other comorbidities related to mortality if relevant. These results may be of interest for clinical intervention and focus on hospital therapies.
- Lines 294 and 295: Does this mean that gut inflammation is an intrinsic factor in healthy community individuals and patients? Is it to be expected that there are gut inflammation factors in the cohort, but some of them are not considered? Or are there some markers that are not considered in reaching this conclusion?
- Lines 304-305: What would be the reason for these differences between faeces and plasma?
- Line 328: Where are the results? Figure 4 only shows the values between LPS hi/low and F/NF.
- Line 330: But considering the data presented in the supplementary figure 9, only some of the IGFs present differences with the other groups, so it is not persisted for all IGF markers (IGF2, IGFBP3, IGF1...).
- Line 345 - 346 Please include in Table 1 the results of the healthy community children. Since the latter group showed some inflammatory features and LPS, it might be interesting to show the "baseline" of the healthy group in Table 1.
- DISCUSSION: One of the findings of the study was the cut-off point of LPS levels in the 90-day survival. Please include this information in the discussion as a relevant value of the study. This should be clearly defined for future studies or as an early point in diagnosis and follow-up of hospitalised patients.
- Line 417: Here it is mentioned that community children have elevated LPS levels but their inflammation is regulated. If this is a feature of chronic exposure, please mention this and support it with references from other studies.
- Line 418: "if comorbidities are present". Please list some of these comorbidities, or at least the most relevant ones.
- Lines 419 and 420: Add some evidence to support this statement. If possible, show the characteristics under short-term/acute LPS-TLR4 activation for comparison with long-term activation.
- Figure 2C: Why was 10.515 EU/ml LPS considered the cut-off in the non-wasted group? In the text, only 11.58 EU/ml was given as the cut-off between high and low LPS concentrations. Please clarify these values.
- Figure 3: The figure legend is a bit complicated. What does the size of the circles mean? Please add key for these shapes. Do you mean that despite the correlation next to 0, you have obtained a $p < 0.05$?
- Figure 6: These proteins are from plasma? Please add "plasma protein".
- Figure 8: differentially up- or down-regulated? If both types of regulation are shown, please specify which proteins are up-regulated and which are down-regulated. Figure 8C: Please explain the font colours in the diagram.
- Figure 9: Add a short summary to each sub-figure.

(Remarks on code availability)

Reviewer #2

(Remarks to the Author)

(Remarks on code availability)

Reviewer #3

(Remarks to the Author)

This manuscript investigates the association between plasma LPS levels, enteric biomarkers, and 90-day mortality in acutely ill children. Using a multicenter cohort, the study explores LPS as a marker of microbial translocation and systemic inflammation, with a focus on EED in low-resource settings. The findings suggest that elevated plasma LPS and inflammatory biomarkers correlate with increased mortality, independent of wasting status.

The study addresses a highly relevant and important topic. It focuses on childhood mortality in resource-limited settings. The investigation of LPS as a marker of microbial translocation is of interest given its potential as a predictor of mortality in acutely ill children. Additionally, the study benefits from its large multicenter cohort design. Nevertheless there are several concerns to be addressed:

1. The manuscript offers little in terms of novel insights. The relationship between LPS, systemic inflammation, and mortality is well-established in the literature, particularly in conditions such as sepsis, malnutrition, and enteric dysfunction. The findings of this study are primarily descriptive and confirm existing knowledge rather than advancing it. There is minimal exploration of new biology or mechanisms, which limits the impact of the study

2. The cohort comprises a highly heterogeneous group of acutely ill children with varying diseases, trajectories, and comorbidities, yet the manuscript lacks detailed stratification or tabulation of these conditions. There were little evaluation regarding the HIV, RTI, sepsis, malaria, or gastroenteritis. Some of the analyses were adjusted for these conditions, there were no detail on which which analyses were adjusted or how. A table summarizing the disease distribution, their status (acute or chronic, primary or secondary or complication diagnoses) and stratification is necessary to understand the heterogeneity of the cohort and the validity of the findings. This lack of clarity raises concerns about potential bias and confounding in the analyses.

3. Although the manuscript categorizes children into groups based on MUAC (SAM / MAM / normal), the assessment appears overly simplistic. The focus on MUAC and IGF markers does not provide a comprehensive evaluation of nutritional status. Were there systematic assessment of key nutritional biomarkers, such as serum albumin, iron, zinc, calcium, and vitamins (A, B9, B12, and D)? These would have been critical to understanding the role of nutrition in mortality, especially that it was presented as a key finding

4. There needs to be more explanation on how some thresholds were selected. The threshold for LPS (11.58) is not adequately explained. The manuscript does not explain why this specific cutoff was chosen or whether it is clinically meaningful. Furthermore, were there sensitivity analyses being performed - e.g. to evaluate the association between LPS and mortality at different time points, such as 30 or 60 days? This lack of justification and robustness undermines the reliability of the findings.

5. The fecal analysis appears narrow in scope, focusing on E. coli (EAEC3 strains) and azithromycin-resistant bacteria. The rationale for focusing on these specific strains is unclear, and the manuscript does not explain why other enteric pathogens or broader microbiome diversity were not assessed. Additionally, there is no mention of whether AMR strains beyond azithromycin resistant bacteria were analyzed. A more comprehensive approach to microbiota and AMR analysis would have provided deeper insights into microbial translocation and its impact on mortality.

6. The authors mentioned about intestinal permeability and translocation. Were biomarkers related to intestinal barrier, e.g. zonulin, claudin being evaluated? Were there other assessment of the GI physiology or function - e.g. villous atrophy, intestinal inflammation - beyond faecal calprotectin?

7. The pathway analysis relies heavily on public datasets and does not provide meaningful mechanistic insights. The lack of independent experimental validation significantly weakens the credibility of the proposed pathways. Furthermore, the discussion around therapeutic targets, such as bevacizumab and tocilizumab, is highly speculative. There is also no exploration of the directional effects of these pathways beyond up- or down-regulation.

(Remarks on code availability)

Version 1:

Reviewer comments:

Reviewer #1

(Remarks to the Author)

the authors have satisfactory responded my queries.

(Remarks on code availability)

This reviewer doesn't have the knowledge to decide whether the code is usable for the community.

Reviewer #3

(Remarks to the Author)

The authors have provided well-structured, and data-backed responses to most reviewer critiques. Their clarifications are mostly satisfactory. However, several issues require a more cautious tone or additional qualification.

1. The authors emphasized their use of proteomic analysis, pathway mapping (via IPA), and re-analysis of public single-cell transcriptomic data to derive cell-type specific pathways (e.g., PI3K/AKT signaling). These remain inferences rather than direct mechanistic evidence. No functional validation (e.g., ex vivo assays, causal modeling) was added. While the systems biology analysis is useful, the claims should remain hypothesis-generating, and this framing is not fully emphasized in the revised response. The mechanistic claims (e.g., PI3K/AKT signaling) should be explicitly qualified as preliminary and exploratory

2. While I understand the rationale for treating comorbidities like malaria or LRTI as potential mediators in the pathway from enteric dysfunction to mortality, this assumption is not definitively supported by data and analysis. Many of these conditions could also act as confounders, independently influencing both systemic inflammation and mortality risk. I would recommend the authors to consider adjust for them in sensitivity analyses or a mediation analysis

(Remarks on code availability)

Version 2:

Reviewer comments:

Reviewer #3

(Remarks to the Author)

The authors have responded satisfactorily to my questions.

(Remarks on code availability)

RESPONSE TO REVIEWER COMMENTS

Reviewer #1 (Remarks to the Author):

In the manuscript "Plasma lipopolysaccharide levels, microbiota and biomarkers of enteric dysfunction predict mortality in acutely ill children in sub-Saharan Africa and South Asia" by Allen and colleagues assessed the associations of plasma LPS and mortality in a cohort of acutely ill hospitalized children where environmental enteric dysfunction (EED), enteric and systemic infections and malnutrition are prevalent. They showed an association of LPS to mortality and chronic inflammation and immunology pathways as well as metabolism and tissue remodelling. The authors suggest that LPS signalling, nutritional status, inflammation and tissue integrity as are important factors linked to mortality in acutely ill children. Overall, the authors provide a thoroughly analysis on the association of plasma LPS and different factors including sex, age, nutritional, microbiota etc and show that LPS, inflammation, metabolism and remodelling appear to be involved in the mortality rate seen in acutely hospitalised children. The authors have identified some potential therapeutic venues targeting inflammation and metabolism which will need to be validated in other cohorts for future clinical studies. Below are a list of comments identified in the paper including the lack of data or information on the analyses used, that need to be addressed.

Major comments

Major comment 1: Study of the nested case-cohort

When describing the results, it is necessary to provide some range values to consider the description of each feature. For example, the nutritional status (severe, moderate and normal) does not specify in the text what are the values and their range to classify each term (Supplementary Methods 5; and line 202 of the manuscript). Please include these values in the text and in the supplementary methods table. Please include the calculation of "CHAIN NCC weights" in Supplementary Methods 5 or in the Methods section. This could be convenient and easier to find for future readers of this study.

Response #1: Thank you for highlighting this issue. We agree with the reviewer's comment and have revised the *Methods* and *Supplementary Methods* sections to clearly define key demographic variables and comorbidities used for cohort and sub-group stratification, as well as those incorporated as adjustments in the analyses. Details for calculating "CHAIN NCC Weights" were also added to *Supplementary Methods 5*.

Changes made:

- We have now added the following definitions to the *Methods* section (lines 198-234):
Demographic variables. Binary *sex* (male, female) and *continent* (Africa, Asia) variables were recorded at hospital admission, while *age* (continuous) and categorical *age group* (<6 months, 6-11 months, ≥12 months) were captured at each time point.

Breastfeeding status. Given its association with environmental enteropathy¹, caregiver-reported *breastfeeding status* was captured at hospital admission.

Anthropometric classification (“nutritional status” in previous version). In the CHAIN cohort, anthropometric classification at each time point was assessed using mid-upper arm circumference (MUAC) and age-specific cutoffs. MUAC is a simple and reliable measure of acute malnutrition², is useful for screening in therapeutic feeding programs³, and aligns with WHO guidelines for the management of acute malnutrition⁴.

Anthropometric classification at hospitalization:

- *Severe Acute Malnutrition (SAM):* MUAC <11.5 cm for children aged ≥ 6 months, MUAC <11 cm for children aged <6 months, or the presence of bilateral pitting oedema.
- *Moderate Acute Malnutrition (MAM):* MUAC 11.5 to <12.5 cm for children aged ≥ 6 months, or MUAC 11 to <12 cm for children aged <6 months.
- *Normal:* MUAC ≥ 12.5 cm for children aged ≥ 6 months, or MUAC ≥ 12 cm for children aged <6 months.
- *Non-wasted:* MUAC >11.5 cm was used where age-specific cutoffs were not employed.

Comorbidities

Children hospitalized at admission were categorized into clinically relevant comorbidities based on common childhood illnesses outlined in the “Pocket Book of Hospital Care for Children”⁵.

Comorbidities were defined as follows:

- *Acute Diarrhoea or Gastroenteritis (yes, no):* clinical examination or caregiver report of acute diarrhoea (<14 days prior to hospitalization) or clinician-diagnosed gastroenteritis. Children who presented with bloody diarrhoea were excluded from this sub-group categorization.
- *Lower Respiratory Tract Infection (LRTI) (yes, no):* clinician-diagnosed LRTI or pneumonia.
- *Upper Respiratory Tract Infection (URTI) (yes, no):* clinician-diagnosed URTI.
- *Malaria (yes, no):* clinician-diagnosed malaria or a positive diagnostic test.
- *Sepsis (yes, no):* clinician-diagnosed sepsis or evidence of acidotic breathing.
- *HIV status (positive, negative):*
 - *Positive:* positive PCR test, positive antibody test, or known HIV-exposed and tested.
 - *Negative:* negative test results or untested status.

Refer to *Supplementary Methods 2* for additional definitions.

- We have now added the calculation of “CHAIN NCC Weights” to *Supplementary Methods 5*:

“CHAIN NCC Weights” were calculated based on the proportion of children sampled into the nested case-control (NCC) from each stratum defined by survival status (survived or died) and anthropometric classification (Normal, MAM, or SAM) in the original CHAIN cohort (n = 3,101). The weights were computed using the formula: CHAIN NCC Weight = 1 / (n_NCC / n_COHORT).

Major comment 2: Comparing the selected cohort and community children

The study included the differences between the hospitalised children and the healthy community children. The authors did not find differences in gut inflammation markers such as myeloperoxidase and calprotectin, but faecal AAT seems to be higher in community children. This last result is not shown in the results section ("High proportion of EED in the CHAIN cohort"), nor in the discussion. Can the authors discuss what the reason for these differences might be?.

Response #2: Thank you for raising this important point. In our initial submission, we briefly mentioned the reduction of fecal AAT in the *admission* cohort (lines 303–304, *Results- initial submission*) and provided possible explanations (lines 423–427, *Discussion- initial submission*). However, we agree that further elaboration would enhance clarity. 61% of the *admission* cohort presented with diarrhoea. Fecal AAT, as a plasma-derived protein, may be more susceptible to dilution effects and protein loss in watery stool compared to fecal calprotectin and myeloperoxidase. We have also added *Supplementary Figure 4I* which shows no difference ($p = 0.25$) in fecal AAT levels between the non-acute diarrhoea *admission* and *community children*.

Supplementary Figure 4G-I: There is no difference in intestinal inflammation and permeability between children without acute diarrhoea at admission and community children

Expression of fecal biomarkers of intestinal inflammation, (G) myeloperoxidase (MPO), (H) calprotectin and (I) Alpha-1-antitrypsin (AAT), are not significantly different between children without acute diarrhoea at admission and community children. Comparisons were assessed by unpaired Wilcoxon signed-rank test and were inverse proportionally weighted to account for selection bias. The x-axis denotes hospitalization or community status while the y-axis represents biomarker expression in units of Log₂ transformed μg/ml, ηg/ml.

Changes made:

- Addition of *Supplementary Figure 4G-I* to Supplementary Materials.
- Update to the *Discussion* section to remove mention of putative role of fecal AAT in line 535:
In fact, the elevation of plasma AAT ~~and subsequent reduction of fecal AAT~~ in the admissions cohort (vs. community children) may represent LPS immunomodulation

since AAT has been shown to inhibit IL-8 production and LPS-induced NF- κ B activation.

Major comment 3: *Is it possible that the "health community" group has some characteristics associated with chronic EED? Please include in the discussion or results section how their data on the community group compare to other healthy groups in similar studies. Also, include the values of the "healthy community children" in the supplementary table 7. This could help and differentiate between the healthy group and the hospitalised group, and also consider the 'baseline' of these values in the community.*

Response #3: Thank you for these valuable suggestions. Our results indicate that community children exhibit elevated fecal biomarkers and reduced HAZ compared to reference values from high-income settings and are comparable with levels observed in other EED cohorts. Within the CHAIN study, community children were included if they had no hospital admission in the 14 days prior to contact with the study team and were not ill and therefore are considered to be well and typical for the community rather than 'healthy'. These community children were from the same neighbourhoods as acutely ill children and therefore, may have had anthropometric deficits, micronutrient deficiencies, helminth infections among others and therefore would not be regarded as 'healthy'. Therefore, the observed levels of intestinal inflammation and stunting—hallmarks of EED—are present among community children.

To address this, we have updated *Supplementary Table 9* stratifying the CHAIN *admission* and *community children*, which shows no difference ($p \geq 0.05$) in stunting frequency or fecal biomarkers levels. In CHAIN community children, mean fecal biomarkers (Calprotectin (CAL) = 341 μ g/ml, myeloperoxidase (MPO) = 3709 ng/ml, alpha-1-antitrypsin (AAT) = 503 μ g/ml) were higher than suggested high-income cutoffs (CAL > 122 μ g/ml, MPO > 2000 ng/ml, AAT > 270 μ g/ml). Additionally, *Supplementary Table 10* provides comparative community baselines for fecal biomarker expression and stunting in children from other independent community cohorts from LMICs where EED is prevalent. The *Results* section has been revised to further elaborate on these trends.

	Admission ^a Mean (SD)	Admission ^a Median (Q1, Q3)	Community ^a Mean (SD)	Community ^a Median (Q1, Q3)	p-value ^e	High-income reference for elevated levels
Fecal Biomarkers						
Alpha-1-antitrypsin (µg/ml) (non-acute diarrhoea) ^b	449 (557)	246 (101, 470)	341 (373)	203 (100, 412)	0.30	270 µg/ml ⁶
Myeloperoxidase (ng/ml) ^c	4106 (4585)	2228 (868, 6263)	3709 (3553)	2255 (1053, 5411)	0.70	<2000 ng/ml ⁷
Calprotectin (µg/ml) ^d (≥12 months)	699 (1,086)	251 (91, 912)	503 (1,193)	188 (105, 514)	0.13	122 µg/ml ⁸
Stunting						
HAZ ^c	-1.62 (1.68)	-1.53 (-2.55, -0.70)	-1.37 (1.36)	-1.41 (-2.23, -0.75)	0.08	0 ^f
Percent HAZ < -2 ^c	38%		31%		0.05	

^a N = inverse proportionally weighted total within CHAIN cohort (N =3101)

^b Unweighted n = 271 (admission), 129 (community children); Weighted N = 844 (admission), 129 (community children)

^c Unweighted n = 638 (admission), 251 (community children); Weighted N = 1894 (admission), 251 (community children)

^d Unweighted n = 233 (admission), 251 (community children); Weighted N = 731 (admission), 251 (community children)

^e Design-based Kruskal-Wallis test; Pearson's X²: Rao & Scott adjustment based on comparison of CHAIN Admission and Community children

^f World Health Organization reference

Supplementary Table 9: Fecal EED biomarker expression and growth stunting in CHAIN population

Fecal EED biomarkers and stunting are elevated in CHAIN admissions and community children compared to references from high-income cohorts. CHAIN admission and community children have similar ($p \geq 0.05$) expressions of fecal EED biomarkers and frequency of stunting. Summary statistics are inverse proportionally weighted to account for selection biases. HAZ = Length-for-age Z-score or Height-for-age Z-score.

Country (cohort, n) (Reference)	Age range	Cohort Description	Alpha-1-antitrypsin Mean (SD) µg/ml or µg/g	Myeloperoxidase Mean (SD) ng/ml	Calprotectin Mean (SD) µg/ml or µg/g	Length-for-age Z-score (HAZ) Mean (SD)	% Stunting LAZ/HAZ <-2SD
CHAIN community (n=251)	Under 5 years	Community-based comparison group from the CHAIN cohort. Children were asymptomatic to serious diseases and/or acute infections.	341 (373)	3,709 (3,553)	576 (1,017)	-1.37 (1.36)	31% ^a
Peru MAL-ED cohort (n=303) (Colston et al. , 2017)	Birth - 5 years	Observation study of 303 infants from a semi-urban community situated in the forested interior of the country. Study exclusion: 1) children with serious illnesses, 2) children whose mother < 16 years of age at birth, 3) children from non-singleton pregnancy. Stool samples were collected in the 2 days before a diarrheal episode. Summary statistics were generated from longitudinal data.	598.3 (651.0)	12,482.8 (12,910.7)		-1.6 (0.9)	
Multiple countries MAL-ED cohort Bangladesh (n=186), Brazil (n=99), India (n=207), Nepal (n=122), Peru (n=145), South Africa (n=132), Tanzania (n=126) (Richard et al. , 2019)	Birth - 5 years	The MAL-ED observational cohort of 1017 children were conducted at multiple sites. Inclusion criteria included birth weight >1500 g, no serious illnesses, mother at least 16 years of age at infant's birth. Samples collected during diarrhoea episodes were excluded from the analysis.	Bangladesh: 400 (100) India: 400 (100) Nepal: 400 (100) Brazil: 300 (100) Peru: 400 (200) South Africa: 200 (100) Tanzania: 300 (200)	Bangladesh: 4380 (1834) India: 7866 (3959) Nepal: 4166 (1880) Brazil: 3008 (2394) Peru: 7950 (3878) South Africa: 4614 (1750) Tanzania: 5511 (2520) (Measured in: ng/mol)		At Enrollment: Bangladesh: -1.0 (1.0) India: -1.0 (1.1) Nepal: -0.7 (1.0) Brazil: -0.8 (1.2) Peru: -1.0 (0.9) South Africa: -0.8 (1.0) Tanzania: -1.0 (1.2) At 5 years old: Bangladesh: -1.6 (0.9) India: -1.5 (0.9) Nepal: -1.3 (0.9) Brazil: -0.2 (1.0) Peru: -1.3 (0.8) South Africa: -0.9 (1.0) Tanzania: -1.9 (0.9)	
Multiple countries Afrobiota project Madagascar (n=417), Central African Republic (CAR) (n=387) (Vonaesch et al. , 2022)	2-5 years old	Cross-sectional study on stunting of 804 children aged 2-5 years from Madagascar and Central African Republic. Study did not include children experiencing severe diseases.	Madagascar: 590.7 (551.0) CAR: 396.3 (318.6)		Madagascar: 764.5 (1024.4) CAR: 549.2 (647.4)	Madagascar: -2.0 (1.1) CAR: -1.8 (1.4)	Madagascar: 48% CAR: 44%
Country (cohort, n) (Reference)	Age range	Cohort Description	Alpha-1-antitrypsin Median [Q1, Q3] µg/ml or µg/g	Myeloperoxidase Median [Q1, Q3] ng/ml	Calprotectin Median [Q1, Q3] µg/ml or µg/g	Length-for-age Z-score (HAZ) Median [Q1, Q3]	% Stunting LAZ/HAZ <-2SD
CHAIN community (n=251)	Under 5 years	Community-based comparison group from the CHAIN cohort. Children were asymptomatic to serious diseases and/or acute infections.	203 [100, 412]	2,255 [1,053, 5,411]	254 [124, 692]	-1.41 [-2.23, -0.75]	31% ^a

Bangladesh MAL-ED cohort (n=246) (Arndt et al. , 2016)	3-24 months	Observational study of 246 children from the Mirpur urban slum. Study exclusion included: 1) children with serious illnesses, 2) children whose mother < 16 years of age at birth, 3) children from non-singleton pregnancy. Participants were followed until 2 years old.	3-21 months: 380 [190, 717.5] (1,194 samples)	3-21 months: 3354.9 [1,594.9, 7430.1] (1,185 samples)		Birth: -0.99 [-1.68, -0.40] 3 months: -1.17[-1.80, -0.47] 24 months: -1.99[-2.60, -1.30]	Birth: 16.7% 3 months: 15.6% 24 months: 49.5
Bangladesh GEMS DSS (n=216) (George et al. , 2015)	6-30 months	Nested cohort study of 216 randomly selected children from rural Bangladeshi households with live chickens in their compound. Participants were followed for 9 months.	260 [160, 510]	3,576.75 [1,969.50, 5,998.00]	402.67 [193.37, 822.30]		9-months follow-up: 34 (n=205)
Zimbabwe SHINE trial (n=1169) (Mutasa et al. , 2021)	0-18 months	Observational study of a subgroup of 1169 infants in rural Zimbabwe. Mothers were uninfected with HIV at enrollment. Children were followed longitudinally.	1 month:350 [170,980] 3 months:360 [180,780] 6 months:340 [190,640] 12 months:300 [160,580] 18 months:230 [110,420]	1 month: 5861 [2777,11647] 3 months: 8221 [3962,18074] 6 months: 7233 [4166,12413] 12 months: 3959 [2107,7044] 18 months: 2063 [967,4109]		1 month: -0.79 [-1.62, -0.03] 3 months: -0.84 [-1.58, -0.07] 6 months: -0.87 [-1.61, -0.07] 12 months: -1.12 [-1.87, -0.46] 18 months: -1.41 [-2.80, -0.74]	
Multiple countries Afrubiota project Madagascar (n=417), Central African Republic (CAR) (n=387) (Vonaesch et al. , 2022)	2-5 years old	Cross-sectional study on stunting of 804 children aged 2–5 years from Madagascar and Central African Republic. Study did not include children experiencing severe diseases.	Madagascar: 500 [220, 795] CAR: 330 [130, 573]		Madagascar: 448 [249, 853] CAR: 308 [172, 617]	Madagascar: -1.97 [-2.83, -1.31] CAR: -1.75 [-2.88, -0.77]	Madagascar: 48% CAR: 44%
India MANTRA program (n=221) (Sinharoy et al. , 2021)	Under 5 years	Nested sub-study of 221 community control children within the larger MANTRA program. Matched cohort study of a household-level water and sanitation intervention in rural Odisha, India.	355.2 [207.6,620.8]	812.8 [495.4,1477.2]		-1.73 (1.32) (Mean)	
Bangladesh MAL-ED (n=265) (Fahim et al. , 2018)	Birth - 2 years	Observational study of 265 healthy newborns from an urban community with low socioeconomic status and inadequate sanitation. Study exclusion included: 1) children with serious illnesses, 2) children whose mother < 16 years of age, 3) children from non-singleton pregnancy.	330 [180,620] (627 samples)	3895.42 [1,563.76, 8432.82] (625 samples)		7 months: -1.22 [-1.87, -0.58] 15 months: -1.74 [-2.42, -1.15] 24 months: -1.98 [-2.60, -1.30]	7 months: 20.3% 15 months: 41.7% 24 months: 47.9%

Supplementary Table 10: Comparison of fecal EED biomarkers and stunting in the CHAIN community children and other EED studies

Fecal biomarkers of environmental enteric dysfunction (EED) and stunting were similarly elevated in the community-based comparison group from the CHAIN cohort compared to other published EED cohorts. Summary statistics for fecal EED biomarkers and Length-for-Age Z-score (HAZ) are reported Mean (SD) (upper panel) or as Median (Q1, Q3) (lower panel), while stunting prevalence is reported as a percentage (%).

Relevant studies were identified through a PubMed search using the following terms on March 14, 2025: (“EED” OR “Environmental Enteropathy” OR “Environmental Enteric Dysfunction” OR “Environmental Enteric Disease” OR “Tropical Enteropathy” OR “Tropical Enteric Dysfunction”) AND (“Myeloperoxidase” OR “MPO”) AND (“alpha-1 antitrypsin” OR “AAT”) OR (“Calprotectin” AND (“alpha-1 antitrypsin” OR “AAT”)) OR (“Myeloperoxidase” OR “MPO”) AND “Calprotectin”) AND (“HAZ” OR “LAZ” OR “height-for-age” OR “length-for-age” OR “stunting”). Inclusion criteria: Studies were included if they assessed at least **two** of the three key fecal biomarkers (α -1-antitrypsin, myeloperoxidase, calprotectin) alongside stunting. When multiple studies were available from the same cohort and sample population, the study most like the CHAIN cohort in age and population characteristics was selected.

Major comment 4: *Lines 405 to 409: Please show these results. Supplementary Table 1 shows that there are significant differences between admission/discharge/community. But are these differences also between admission and community? If not, it might be interesting to show the 16s rRNA results to compare both communities and find a common genus of bacteria. Furthermore, studying the α -diversity or β -diversity after hospital intervention will enrich this study.*

If there are no significant differences, this should be mentioned in the discussion and explain that community may be a chronic feature in EED. As previously stated, it is relevant to compare to other healthy controls from other studies to draw on how "healthy" the community group used in this study actually is.

Response #4: We agree that further characterization of the community children would strengthen our manuscript. To address this, we have added a new table (*Table 1*) presenting comparisons of demographics, comorbidity frequencies, levels of fecal biomarkers, plasma barrier function proteins, plasma LPS, LPS signal transduction markers, as well as entero-pathogen presence and abundance. These results reveal variations in barrier function proteins and entero-pathogen presence and expression in the admission cohort compared to community children. 16S rRNA was detected at significantly elevated levels ($p < 0.001$) in the admission group compared to community children. More specifically, quantification of *Enteroaggregative E. coli*, *Enterotoxigenic E. coli*, *Shigella toxin-producing Enterotoxigenic E. coli*, *V. cholerae* and *CTX-M-producing* bacteria were significantly elevated ($p < 0.01$) in the admission cohort.

Our findings highlight that while EED may be prevalent in the community, barrier dysfunction and entero-pathogen load are exacerbated during acute infection. While we agree that α - and β -diversity assessment could provide insights into the mechanisms of LPS-associated mortality, our current analyses are constrained by the predefined, targeted detection of specific pathogens and 16S rRNA. The real-time PCR TaqMan Array Card (TAC, Life Technologies, USA) platform used in this study detects 18 bacterial targets, 2 antimicrobial resistance gene, and quantifies 16S rRNA. Robust diversity analyses ideally require comprehensive microbial community profiling through sequencing of 16S rRNA amplicons or through metagenomics which was not done in this study, but which would have allowed for a more informative assessment of richness and evenness.

Characteristics	Admission Weighted N = 1,894 ^a Unweighted N = 638	Community children Weighted N = 251 ^a Unweighted N = 251	p-value ^b
Plasma LPS (E.U./ml) (Mean (SD))	4.40 (5.25)	4.47 (5.34)	>0.9
Demographics			
Age group			0.2
<6 months	361 / 1,894 (19%)	41 / 251 (16%)	
6-11 months	690 / 1,894 (36%)	81 / 251 (32%)	
12 months & above	844 / 1,894 (45%)	129 / 251 (51%)	
Admission age (Mean (SD))	11.47 (5.57)	12.48 (6.11)	0.049
Sex			0.047
Female	742 / 1,894 (39%)	118 / 251 (47%)	
Male	1,152 / 1,894 (61%)	133 / 251 (53%)	
Continent			0.032
Africa	1,085 / 1,894 (57%)	165 / 251 (66%)	
Asia	809 / 1,894 (43%)	86 / 251 (34%)	
Comorbidity frequencies			
Malaria History	337 / 1,894 (18%)	11 / 251 (4.4%)	<0.001
Sepsis	295 / 1,894 (16%)	0 / 251 (0%)	<0.001
Gastroenteritis	1,142 / 1,872 (61%)	0 / 251 (0%)	<0.001
URTI	98 / 1,894 (5.2%)	0 / 251 (0%)	0.060
LRTI	774 / 1,894 (41%)	0 / 251 (0%)	<0.001
HIV status			0.2
Negative	1,844 / 1,894 (97%)	248 / 251 (99%)	
Positive	50 / 1,894 (2.6%)	3 / 251 (1.2%)	
TB	3 / 1,701 (0.2%)	0 / 221 (0%)	0.6
Breastfeeding status	1,428 / 1,894 (75%)	202 / 251 (80%)	0.12
Presence of at least 1 Entero-pathogen			
Viruses	776 / 1,702 (46%)	53 / 222 (24%)	<0.001
Parasites	517 / 1,770 (29%)	56 / 231 (24%)	0.2
Gram-negative Bacteria	1,579 / 1,820 (87%)	206 / 241 (85%)	0.6
Entero-pathogen Quantification (Mean (SD))^c			
Enterococci	28.85 (6.31)	30.31 (5.32)	0.007
Enterotoxigenic E. coli	32.79 (4.69)	33.86 (2.91)	0.008
Shigella toxin-producing Enterotoxigenic E. coli	33.51 (4.06)	34.63 (1.71)	<0.001
V. cholerae	34.79 (1.53)	35.00 (0.02)	0.006
CTX-M-producing bacteria	27.24 (6.79)	30.62 (4.99)	<0.001
X16s	13.90 (3.28)	15.25 (3.71)	<0.001
Cryptosporidium	33.90 (3.46)	34.35 (2.58)	0.015
Adenovirus 40/41	34.22 (3.19)	34.71 (1.59)	0.005
Rotavirus	33.01 (4.26)	34.92 (0.82)	<0.001
Sapovirus	34.64 (1.65)	34.89 (0.77)	0.029
Fecal Biomarkers of EED (Mean (SD))			
Fecal MPO ng/ml	4,105.51 (4,584.89)	3,708.52 (3,552.86)	0.7
Fecal Calprotectin (> 12 months) (µg/ml)	698.85 (1,086.33)	502.96 (1,192.81)	0.13
Fecal AAT (non-acute diarrhoea) (µg/ml)	448.57 (557.01)	341.36 (372.63)	0.30
Barrier Function Biomarker (Mean (SD))^d			
Plasma Zonulin	46,479.64 (29,948.42)	33,384.53 (19,464.19)	<0.001
Plasma Diamine Oxidase	4,776.11 (4,226.63)	6,809.30 (3,594.82)	<0.001
Plasma FABP2	3,208.20 (2,915.06)	2,746.65 (1,951.33)	0.3
Plasma CDH1.1	1,468.48 (2,669.83)	1,602.89 (2,216.58)	0.005
Plasma OCLN	960.71 (291.05)	908.77 (166.60)	0.005
Plasma ZO-1	719.87 (1,718.78)	788.77 (1,570.68)	<0.001
Systemic Immune Activation and Inflammation Biomarkers (Mean (SD))^d			
Plasma Heparin-Binding Protein	24,326.27 (11,449.57)	19,629.43 (8,129.72)	<0.001
Plasma TREM1	2,168.66 (1,457.04)	1,691.83 (515.93)	<0.001
Plasma SOCS-3	1,480.93 (2,312.10)	1,305.70 (440.17)	0.074
Systemic Inflammatory Biomarkers (Mean (SD))^d			
Plasma CRP	69,067.92 (35,630.07)	32,055.93 (28,576.09)	<0.001
Plasma SAA1	72,275.72 (72,226.17)	17,974.93 (35,545.88)	<0.001
Plasma MPO	27,618.30 (14,306.43)	16,296.82 (5,131.00)	<0.001
Plasma Calprotectin	1,773.65 (1,065.01)	1,749.45 (3,856.34)	<0.001
Plasma AAT	50,645.94 (15,708.32)	33,395.31 (9,471.53)	<0.001

^aN = inverse proportionally weighted total within CHAIN cohort (n=3101); ^bDesign-based Kruskal-Wallis test, Pearson's X²; Rao & Scott adjustment; ^cThreshold Cycle (Ct) from quantitative PCR (qPCR) assay; E.U./ml = Endotoxin Unit per ml; ^dRFU = Relative Fluorescence Unit

Table 1: Demographic, Entero-pathogen, Fecal Biomarkers and Plasma Barrier and Systemic Inflammation Profiles of Hospitalized vs. Community Children

Demographics (age, sex, continent), fecal biomarkers (MPO, calprotectin, AAT), and plasma LPS concentrations are comparable between Admission and Community children. In contrast, the presence of comorbidities (malaria, sepsis, gastroenteritis, LRTI), barrier protein expression (zonulin, diamine oxidase) and plasma systemic inflammatory (CRP, SAA1, MPO, calprotectin, AAT) differ significantly between the two cohorts. Both unweighted and inverse proportionally weighted totals (N) are presented. Only entero-pathogen with significant difference in expression between the cohorts are presented. See Supplementary Table 7C for all pathogen levels. LPS = lipopolysaccharide, MPO = myeloperoxidase, AAT = Alpha-1-antitrypsin, FABP2 = Intestinal-type fatty acid-binding protein, TLR4 = Toll-like Receptor 4.

Major comment #5: A cellular model of LPS associated immunopathology

The authors have mapped the plasma proteomic analysis to unravel the interaction between the immune system and the epithelium. However, the results section lacks information on these results (lines 367 to 370), showing only some of the differentially abundant proteins without explaining if they were up- or down-regulated and from which immune cell type. Furthermore, it is necessary to show the highlighted proteins and cell type interaction from this study section to explain the LPS and immune-immune and immune-epithelial interaction. These comments are also relevant to Figure 7.

Response #5: Thank you for raising this valid point. To clear this confusion, we have now edited the manuscript to mention the directionality of the protein expression. We did try to plot the expression of all differentially abundant proteins showcased in Figure 6A and B, but the transcript level expression of some of these proteins in the publicly available dataset used was either not detected, or very low to be visualised on the dot plot. Thus, we had to exclude these proteins from the dot plot. We have now elaborated on the cell types expressing some example proteins to highlight the proposed immune-immune and immune-epithelial cross-talks.

Changes made:

- Updates in results section (lines 445-464):

A cellular model of LPS associated immunopathology

Plasma proteomics analysis showed that immunologically relevant soluble membrane proteins (like TLR4, IFNGR2) (Figure 6A) or extracellular proteins (like CCL8, IFNA7) (Figure 6B), were downregulated in *LPS^{hi} F* compared to all the other sub-groups, indicating an immune dysregulation. To understand the mechanisms of intestinal and systemic inflammation and LPS signalling, we leveraged our SomaScan proteomics data and a publicly available single cell transcriptomics dataset from EED patients⁹. The expression of genes encoding differentially abundant proteins in *LPS^{hi} F* sub-group by cell types in single cell datasets was analysed (Supplementary Figure 7). This deconvolution method helped us identify which intestinal cell types are potentially involved in immune signalling in *LPS^{hi} F* sub-group. The transcript level expression of the differentially expressed proteins in the single cell dataset indicated that these are expressed by various immune (i.e. TLR4, VCAN) and epithelial cell

types (i.e. DPP4, VTN) in the small intestine (Figure 6C). This similarly allowed to visualise cellular interaction mediated by genes for relevant differentially abundant proteins, like CCL8, IFNG, VEGF, VCAN, etc., and their binding partners. The possible interactions like - CCL8 (expressed by monocytes) with CCR2/CCR5 (Plasma cells and T cells), XCL1 (T cell) with XCR1 (dendritic cells), VCAN (monocytes) with CD44 (immune and epithelial cells) and IFNG (T cells) with IFNGR2 (immune and epithelial cells) highlight immune-immune and immune-epithelial cell cross-talk (Figure 7A-B) illustrating immunopathology in LPS-mediated mortality in the *LPS^{hi} F* sub-group.

Major comment #6: Discussion

In lanes 404-13, the authors offer some explanations to the source of LPS, as it was not identified in this study. The authors indicate that “leaky gut” might be one possible factor based on fecal marker calprotectin. Although calprotectin is a marker of inflammation (neutrophils/macrophages), other markers could have been examined in the plasma e.g. zonulin, lipopolysaccharide-binding protein (LBP) and intestinal fatty acid-binding protein (I-FAB). The authors could also mention that reduction in pathways associated with cell junction organisation and RAC signalling (identified in the IPA analysis in Fig 8A) as supportive data to their speculation on leaky gut.

Response #6: Thank you for this valuable suggestion. We have added comparisons of plasma barrier function markers (zonulin, FABP2, diamine oxidase, cadherin-1, occludin, and ZO-1) between admission and community children in *Table 1*. These results indicate significant barrier dysfunction in the *admission* cohort.

While plasma FABP2 levels were similar between groups ($p = 0.3$), suggesting comparable enterocyte damage, plasma zonulin was significantly elevated ($p < 0.001$) in the *admission* cohort, indicating increased intestinal permeability. Additionally, intestinal mucosal integrity appeared more compromised in the *admission* group ($p < 0.001$), as evidenced by lower plasma diamine oxidase levels. Reduced plasma cadherin-1 and ZO-1 ($p < 0.01$) suggest weakened adhesion and greater tight junction breakdown.

Although plasma occludin was unexpectedly elevated in the *admission* cohort, this may reflect increased shedding due to epithelial barrier disruption rather than enhanced tight junction function.

Taken together, these unadjusted comparisons suggest significant barrier dysfunction in the *admission* cohort, likely driven by acute infection and increased entero-pathogen load.

Changes made (lines 505-511):

Discussion:

However, similar plasma LPS levels between the cohorts indicate that hospitalization-related enteric bacterial infection is not the sole source of elevated plasma LPS. Further studies are needed to evaluate the impact of diet, dysbiosis, liver impairment, and antibiotic use on plasma LPS concentrations. But a role for leaky gut is supported by expression of fecal biomarkers like calprotectin, plasma barrier function markers like zonulin, FABP2, ZO-1

(Table 1) as well as inhibition of RAC signalling and cell junction organization pathways (Figure 8A) in *LPS^{hi} F* group.

Major comment #7: *Another question is whether the significance is underestimated when comparing the diseased children against the healthy community population, as these also had high levels of LPS. This could have been addressed more in the discussion. How would other studies agree with the findings in this study? A short discussion on potential association or not with other studies should be included.*

Response #7: Thank you for this suggestion. We agree with the reviewer, as *community children* were from the same neighbourhoods as acutely ill children and may have had anthropometric deficits, micronutrient deficiencies, and helminth infections, among other factors. *Community children* in the CHAIN study were included if they had no hospital admission in the 14 days prior to contact with the study team and should thus be characterized as ‘not acutely ill’ rather than ‘healthy’. In fact, *Supplementary Table 9* suggest that intestinal inflammation and stunting are similar in the CHAIN *admission* cohort and *community children*.

We have also added *Supplementary Table 10* which shows fecal biomarkers and stunting data for 5 other independent cohorts from LMICs with high prevalence of EED. Our findings are comparable to other studies in our settings. For example, in the CHAIN *community children*, the medians and ranges detected for fecal AAT (203 [100, 412]) and MPO (2,255 [1,053, 5,411]) are comparable to children at the 18-month timepoint of the SHINE trial from Zimbabwe (AAT: 230 [110, 420]; MPO: 2063 [967, 4109]) (Mutasa *et al.*, 2021). The CHAIN *community children* median and range detected for fecal calprotectin (254 ([124, 692]) are comparable to the Central African Republic site of the *Afrobiota* project (308 [172, 617]) (Vonaesch *et al.*, 2022). Stunting frequency in CHAIN *community children* (31%) is comparable to the 9-months follow-up levels (34%) of the Bangladesh site of the GEMS DSS (George *et al.*, 2015), while median length-for-age Z-score in CHAIN *community children* (-1.41 [-2.23, -0.75]) is similar to children at the 18-month timepoint (-1.41 [-2.80, -0.74]) of the SHINE trial from Zimbabwe (Mutasa *et al.*, 2021).

Changes made:

- Addition of the following to the *Discussion* section (lines 511-519):
Thresholds of EED biomarkers in this setting would benefit from systematic reviews and meta-analyses that adjust for key confounders such as age, HIV status, and diarrhoea. Previous studies have employed composite EED scores to standardize EED assessment^{10,11}, and while valuable, these scores are often cohort-specific, relying on the internal percentiles of each biomarker. There is a need for more appropriate EED biomarker thresholds, particularly for milder levels of intestinal inflammation and/or permeability, to better reflect conditions in EED settings. Standardizing summary statistics and adjusting for these variables in future research would improve comparability across studies and enhance our understanding of EED biomarkers in community settings.
- *Supplementary Tables 9 and 10* have been shown in response to comment 3 above.

Minor comments

• *Line 96: Considering the reference mentioned by the authors, the change is a shift from a pro-inflammatory effect and stimulation to a resolving and anti-inflammatory response. Given this, what is the significance or relevance to this study?*

Response: We agree and thank the reviewer for this observation and have now removed the text and the associated reference from line 96 in the introduction whose focus was on tolerance to LPS which is not the focus of the manuscript.

Change made:

- Update to the *Introduction* section (lines 98-100):
LPS engages with TLR4 and accessory proteins (MD-2 and LBP) to induce an inflammatory response with tumor necrosis factor (TNF) and interleukin (IL)-1 β release¹². ~~Chronic exposure to low concentration LPS can lead to a LPS tolerance response¹³~~. LPS-TLR4 interaction plays an indispensable role in the gut homeostasis.

• *Line 134-141: In general, the number of children included in the different parts of the study are difficult to follow, especially when looking at the supplementary tables. For example, in Suppl Table 4, there seem to be 1894 samples tested for LPS, while in the text is stated in lines 139-141: “Plasma samples and LPS levels were available from 889 children including 199 children that died during the study period, 439 survivors and 251 community children”*

Response: Thank you for highlighting this. We have now updated the manuscript (including *Results* and table headers) to indicate unweighted totals (‘n’) and inversely proportionally weighted totals (‘N’).

• *Line 202: Please add a range of values for nutritional status or more examples of relevant comorbidities in the study.*

Response: We agree that variable definition would enhance clarity and have updated the manuscript accordingly. This includes addition of definitions for demographics, breastfeeding status and comorbidities to the *Methods* section.

• *Lines 279 and 280. Please add other comorbidities related to mortality if relevant. These results may be of interest for clinical intervention and focus on hospital therapies.*

Response: Thank you for pointing this out. We have added blood glucose, dehydration, capillary refill, and oedema to this section (Supplementary Table 8).

Change made:

- Update to the *Results* section:
Plasma LPS associations with mortality were reinforced by elastic net regression (Supplementary Table 8), which also identified other mortality-related comorbidities, such as anthropometric classification, blood glucose, dehydration, capillary refill, oedema, fecal bacterial populations and fecal calprotectin. These data suggest that high plasma LPS levels are a strong indicator of mortality, independent of nutritional status.

• *Lines 294 and 295: Does this mean that gut inflammation is an intrinsic factor in healthy community individuals and patients? Is it to be expected that there are gut inflammation factors in the cohort, but some of them are not considered? Or are there some markers that are not considered in reaching this conclusion?*

Response: Our study and others in these settings demonstrate *community children* frequently have enteric infections and high levels of both intestinal and systemic inflammation¹². *Community children* within this study were recruited to help establish socio and biological norms but may have anthropometric deficits, micronutrient deficiencies, helminth infections among other factors and therefore would not be regarded as ‘*healthy*’. The high levels of these biomarkers are likely driven by the burden of enteric pathogens which are clinically silent. For example, Kosek *et al.* 2017 demonstrated that higher burdens of either entero-invasive and/or mucosal-disrupting entero-pathogens were associated with elevated biomarker of gut, systemic inflammation and reduced growth¹².

• *Lines 304-305: What would be the reason for these differences between faeces and plasma?*

Response: Biomarkers may differ between stool and plasma due to their direct interaction with the gut environment including the inciting pathogens, the complex matrix and the changes that may occur in fecal material as they transit the gut. Plasma biomarkers may reflect broadly systemic changes throughout the body and may vary depending, for example, on the severity of EED and general pro- and anti-inflammatory immune balances to ensure systemic homeostasis.

• *Line 328: Where are the results? Figure 4 only shows the values between LPS^{hi/low} and F/NF.*

Response: Thank you for bringing this to our attention. We have now added 2 new tables (*Supplementary Table 16* and *17*) to capture the summary statistics (mean and standard deviation of log₂ transformed value) of the top and bottom expressed proteins in children from the *LPS^{hi} Fatal* sub-group. The Wilcoxon p-value was also provided for protein comparisons of *LPS^{hi} Fatal* sub-group vs. the other 3 *LPS-mortality* sub-groups.

	LPSHI F (SD)	LPSLO F (SD)	LPSHI NF (SD)	LPSLO NF (SD)
CA6	9.578 (1.189)	9.759 (1.16)	10.445 (1.197)	10.568 (1.214)
CA6.1	8.361 (1.724)	8.776 (1.685)	9.979 (1.73)	9.987 (1.67)
CDON	11.963 (0.921)	12.519 (1.012)	12.61 (0.928)	12.983 (0.968)
COL10A1	11.585 (1.601)	11.839 (1.746)	12.951 (1.547)	13.324 (1.555)
COL1A1	12.694 (1.71)	13.086 (1.687)	14.135 (1.411)	14.345 (1.521)
DEFA3	15.123 (1.404)	14.262 (1.69)	13.538 (1.466)	13.331 (1.33)
FTH1.FTL	13.687 (1.739)	13.664 (1.693)	12.511 (1.963)	12.764 (1.711)
FTL	13.83 (1.717)	13.792 (1.677)	12.69 (1.912)	12.891 (1.699)
HAMP	12.59 (2.191)	11.812 (2.083)	11.36 (2.305)	11.545 (1.828)
IGF1	10.876 (0.856)	11.204 (0.925)	11.859 (0.894)	12.127 (1.0)
IGF2	11.146 (1.193)	11.679 (1.158)	12.602 (0.92)	12.824 (1.112)
IGFBP3	12.007 (0.844)	12.23 (0.886)	12.922 (0.914)	13.206 (0.994)
IGLL1	11.546 (1.895)	11.554 (1.739)	12.836 (1.634)	13.247 (1.678)
IL1RL1	15.057 (1.267)	14.695 (1.661)	13.716 (1.404)	13.519 (1.402)
IL1RN	13.588 (1.358)	12.742 (1.242)	12.311 (1.111)	12.275 (1.106)
LCN2	11.506 (1.612)	10.773 (1.535)	10.549 (1.258)	9.844 (1.104)
LTA.LTB	7.383 (0.595)	7.961 (0.807)	8.236 (0.78)	8.688 (0.838)
PADI1	7.608 (1.479)	7.556 (0.95)	7.531 (0.593)	7.67 (0.864)
PLA2G2A	13.128 (1.914)	12.21 (2.306)	12.628 (1.994)	12.166 (2.002)
PRSS2	13.702 (1.815)	12.944 (1.596)	12.762 (1.823)	12.049 (1.496)

Supplementary Table 16: Sub-group summary statistics of top differentially expressed proteins

Summary statistics (Mean and standard deviation) of the top10 or bottom 10 expressed proteins. Proteins were selected after assessment of log2 fold change of expression between LPS^{hi} F sub-group (n=37) and community children (n=251). Statistics are presented for each LPS-mortality sub-group (LPS^{lo} F, LPS^{hi} NF, LPS^{lo} NF and LPS^{hi} F). SD = standard deviation, F = Fatal, NF = Non-fatal.

	GENE	COMPARISON	P_VALUE
1	CA6.1	LPSHi F vs LPSlo F	0.122
2	CA6.1	LPSHi F vs LPSHi NF	5.65E-05
3	CA6.1	LPSHi F vs LPSlo NF	1.43E-07
4	IGF1	LPSHi F vs LPSlo F	0.0381
5	IGF1	LPSHi F vs LPSHi NF	1.99E-06
6	IGF1	LPSHi F vs LPSlo NF	<1e-10
7	IGF2	LPSHi F vs LPSlo F	0.0166
8	IGF2	LPSHi F vs LPSHi NF	6.20E-08
9	IGF2	LPSHi F vs LPSlo NF	<1e-10
10	LTA.LTB	LPSHi F vs LPSlo F	2.66E-05
11	LTA.LTB	LPSHi F vs LPSHi NF	1.56E-07
12	LTA.LTB	LPSHi F vs LPSlo NF	<1e-10
13	CA6	LPSHi F vs LPSlo F	0.214
14	CA6	LPSHi F vs LPSHi NF	0.000493
15	CA6	LPSHi F vs LPSlo NF	9.70E-07
16	COL10A1	LPSHi F vs LPSlo F	0.502
17	COL10A1	LPSHi F vs LPSHi NF	0.000131
18	COL10A1	LPSHi F vs LPSlo NF	5.15E-09
19	IGLL1	LPSHi F vs LPSlo F	0.808
20	IGLL1	LPSHi F vs LPSHi NF	0.000826
21	IGLL1	LPSHi F vs LPSlo NF	1.32E-07
22	CDON	LPSHi F vs LPSlo F	0.00135
23	CDON	LPSHi F vs LPSHi NF	0.00131
24	CDON	LPSHi F vs LPSlo NF	5.56E-09
25	IGFBP3	LPSHi F vs LPSlo F	0.14
26	IGFBP3	LPSHi F vs LPSHi NF	5.49E-06
27	IGFBP3	LPSHi F vs LPSlo NF	<1e-10
28	COL1A1	LPSHi F vs LPSlo F	0.233
29	COL1A1	LPSHi F vs LPSHi NF	0.000107
30	COL1A1	LPSHi F vs LPSlo NF	5.47E-08
31	IL1RL1	LPSHi F vs LPSlo F	0.299
32	IL1RL1	LPSHi F vs LPSHi NF	2.77E-05
33	IL1RL1	LPSHi F vs LPSlo NF	2.17E-09
34	PRSS2	LPSHi F vs LPSlo F	0.0189
35	PRSS2	LPSHi F vs LPSHi NF	0.0191
36	PRSS2	LPSHi F vs LPSlo NF	7.28E-08
37	PADI1	LPSHi F vs LPSlo F	0.565
38	PADI1	LPSHi F vs LPSHi NF	0.42
39	PADI1	LPSHi F vs LPSlo NF	0.0174
40	DEFA3	LPSHi F vs LPSlo F	0.00548
41	DEFA3	LPSHi F vs LPSHi NF	7.98E-06
42	DEFA3	LPSHi F vs LPSlo NF	<1e-10
43	HAMP	LPSHi F vs LPSlo F	0.0353
44	HAMP	LPSHi F vs LPSHi NF	0.0157
45	HAMP	LPSHi F vs LPSlo NF	0.000775
46	PLA2G2A	LPSHi F vs LPSlo F	0.0221
47	PLA2G2A	LPSHi F vs LPSHi NF	0.227

48	PLA2G2A	LPS ^{hi} F vs LPS ^{lo} NF	0.0039
49	FTL	LPS ^{hi} F vs LPS ^{lo} F	0.892
50	FTL	LPS ^{hi} F vs LPS ^{hi} NF	0.00529
51	FTL	LPS ^{hi} F vs LPS ^{lo} NF	0.000826
52	FTH1.FTL	LPS ^{hi} F vs LPS ^{lo} F	0.951
53	FTH1.FTL	LPS ^{hi} F vs LPS ^{hi} NF	0.00499
54	FTH1.FTL	LPS ^{hi} F vs LPS ^{lo} NF	0.00116
55	IL1RN	LPS ^{hi} F vs LPS ^{lo} F	0.00108
56	IL1RN	LPS ^{hi} F vs LPS ^{hi} NF	2.54E-05
57	IL1RN	LPS ^{hi} F vs LPS ^{lo} NF	2.27E-08
58	LCN2	LPS ^{hi} F vs LPS ^{lo} F	0.0153
59	LCN2	LPS ^{hi} F vs LPS ^{hi} NF	0.00458
60	LCN2	LPS ^{hi} F vs LPS ^{lo} NF	5.77E-10

Supplementary Table 17: Wilcoxon sub-group comparison of top differentially expressed proteins
Wilcoxon comparison of LPS-mortality sub-groups (LPS^{lo} F, LPS^{hi} NF, LPS^{lo} NF vs. LPS^{hi} F) of the top10 or bottom 10 expressed proteins in the LPS^{hi} F sub-group. The top 10 and bottom 10 proteins were selected after assessment of log₂ fold change of expression between LPS^{hi} F sub-group (n=37) and community children (n=251). F = Fatal, NF = Non-fatal.

• *Line 330: But considering the data presented in the supplementary figure 9, only some of the IGFs present differences with the other groups, so it is not persisted for all IGF markers (IGF2, IGFBP3, IGF1...).*

Response: Thank you for highlighting this point. *Supplementary Table 10 (above)* demonstrates that IGF axis biomarkers have significantly lower expression in the LPS^{hi} F sub-group compared to the non-fatal sub-groups (LPS^{hi} NF or LPS^{lo} NF). These results suggest that LPS-associated mortality mechanisms may be linked to nutritional factors. Now *Supplementary Figure 6* further indicates that differences in IGF biomarker expression are most pronounced in more severely malnourished children. We acknowledge that the smaller differences observed in IGF biomarkers between the LPS^{hi} F and LPS^{lo} F sub-groups likely reflects the involvement of IGF suppression in other, non-LPS-associated mortality mechanisms.

• *Line 345 - 346 Please include in Table 1 the results of the healthy community children. Since the latter group showed some inflammatory features and LPS, it might be interesting to show the "baseline" of the healthy group in Table 1.*

Response: We agree and have added a new tables (*Table 1, Supplementary Table 11 and Supplementary Table 13*) to show comparison of entero-pathogen presence and quantification.

• *DISCUSSION: One of the findings of the study was the cut-off point of LPS levels in the 90-day survival. Please include this information in the discussion as a relevant value of the study. This should be clearly defined for future studies or as an early point in diagnosis and follow-up of hospitalised patients.*

Response: Thank you for suggesting this. We have added this as a finding.

Change made:

- Addition to the *Discussion* section (lines 561-567):
Our data suggest that plasma LPS is an independent indicator of mortality. Using the Maxstat method, we identified a novel, unweighted cut-off of 11.58 E.U./ml for plasma LPS in the admission population, which stratified 90-day survival. While this finding provides an initial reference point for LPS-associated pathophysiology, the plasma LPS threshold identified is likely specific to this population, influenced by its unique comorbidity profiles. Notably, bootstrapping analysis (Supplementary Figure 1B) revealed four distinct clusters of LPS cutpoints, suggesting the existence of sub-populations with varying mortality risk profiles linked to LPS levels.

• *Line 417: Here it is mentioned that community children have elevated LPS levels but their inflammation is regulated. If this is a feature of chronic exposure, please mention this and support it with references from other studies.*

Response: We have moved a previous reference¹³ (Burgueño, J. F. & Abreu, M. T, 2020) from the *Introduction* to this section of the *Discussion* to emphasize the point that chronic or low-dose LPS inflammation may have immunoregulatory effects.

• *Line 418: "if comorbidities are present". Please list some of these comorbidities, or at least the most relevant ones.*

Response: We have updated this sentence to indicate that acute intestinal infections may disrupt the equilibrium of LPS-associated regulation.

Change made:

- Update to the *Discussion* section (lines 524-527):
This implies that inflammation due to elevated plasma LPS levels are typically regulated within community children due to chronic exposure³⁷, but the equilibrium might be disrupted, leading to immune dysregulation when comorbidities, such as acute intestinal infections, are present.

• *Lines 419 and 420: Add some evidence to support this statement. If possible, show the characteristics under short-term/acute LPS-TLR4 activation for comparison with long-term activation.*

Response: We have now added two references as evidence pertaining to tolerance mechanisms in LPS-TLR4 axis, specifically in gut milieu.

Changes made:

- Update to Discussion section (lines 527-529):
Short term LPS-TLR4 activation results in a complex systemic signaling response, but low dose or chronic LPS can induce long term immunomodulatory effects^{13,14}.

• *Figure 2C: Why was 10.515 EU/ml LPS considered the cut-off in the non-wasted group? In the text, only 11.58 EU/ml was given as the cut-off between high and low LPS concentrations. Please clarify these values.*

Response: Thank you for pointing this out. While we acknowledge that the LPS cutoffs will likely vary based on the complexity of comorbidities of the patient, we have updated this figure with a 11.58 E.U/ml cutoff to assist with simplicity of interpretation. The result of this analysis remains unchanged – with plasma LPS still significantly associated with mortality ($p < 0.001$) in the non-wasted group.

Figure 2C: Levels of plasma LPS in non-wasted children at admission is associated with mortality

(C) Unweighted Kaplan Meier curve show association between plasma LPS and mortality in the non-wasted ($MUAC > 11.5\text{cm}$) hospitalized children. Plasma LPS levels were stratified according to maximally selected rank statistics for the admission cohort (Supplementary Figure 1A) and survival curves were compared by Log-rank tests. Colored shadings represent confidence interval for each stratum. A table of the number of at-risk participants and the cumulative number of censored cases is included.

• *Figure 3: The figure legend is a bit complicated. What does the size of the circles mean? Please add key for these shapes. Do you mean that despite the correlation next to 0, you have obtained a $p < 0.05$?*

Response: We have moved the original *Figure 3* to *Supplementary Figure 5* and replaced it with a new figure that provides a more informative, systems biology-based, and unbiased approach to identifying proteins associated with plasma LPS and 90-day survival.

In this updated analysis, correlations were assessed across more than 7,000 proteins with plasma LPS and 90-day survival. P-values for the full correlation dataset were adjusted for false discovery rate using the Benjamini-Hochberg method. To enhance interpretability, only associations with an adjusted p-value < 0.05 and a correlation coefficient (R) between 0.4 and -0.4 were included in the correlogram, highlighting proteins most strongly correlated with LPS or mortality. The figure now displays correlation coefficients for each comparison, with the legend clarifying that only statistically significant associations ($p < 0.05$) are shown. Non-significant correlations are represented as blank, white cells.

A**B**
Figure 3: Correlation of proteins and their associations with plasma LPS and 90-day survival in *LPS^{hi} Fatal* sub-group.

Correlograms depicting the associations of (A) plasma LPS and (B) 90-day survival with the strongest of correlations with SomaScan proteins detected in the *LPS^{hi} Fatal* sub-group. Correlation matrices were inverse-proportionally weighted to account for selection biases. Correlograms were hierarchically clustered using the ‘ward.D2’ method. Each circle represents the correlation coefficient (r), with color intensity indicating the strength of the correlation. Positive correlations are shown in red, and negative correlations are shown in blue. Only correlations with an absolute value of $r \geq 0.4$ are displayed, highlighting moderate to high associations. Additionally, only coefficients with a p-value < 0.05 are shown; coefficients with p-value ≥ 0.05 are represented as blank, white circles. All p-values were adjusted for false discovery rates using the Benjamini-Hochberg (BH) method, based on the full SomaScan dataset of over 7300 proteins.

- *Figure 6: These proteins are from plasma? Please add "plasma protein".*

Response: Thank you for pointing this out. Detection of all SomaScan proteins were done using plasma samples. We have now updated our descriptions to differentiate between “*membrane-derived*” and “*soluble*” plasma proteins.

- *Figure 8: differentially up- or down-regulated? If both types of regulation are shown, please specify which proteins are up-regulated and which are down-regulated. Figure 8C: Please explain the font colours in the diagram.*

Response: Thank you for pointing this out. The pathways shown in Figure 8A are both up- and down-regulated in the *LPS^{hi} F* group. The upregulated pathways are depicted in red colour in the heatmap and downregulated pathways are depicted in blue. The proteins important in immunological pathways detected in IPA analysis (Figure 8A) like IL1RL1, IL10, GDF15, VEGFA and their binding partners in signaling (assessed using CellphoneDB) were deconvoluted using publicly available single cell dataset. The main take-away of this panel is that the proteins involved in pathways are expressed by immune and epithelial cell types indicating an immune cross-talk. We have now edited this part in results to clarify this better. The regulation directionality of pathways was mentioned in the “Key” box in figure 8C. to avoid any confusion, the legend of Figure 8C has been modified to include a color guide.

Changes made:

- Updated results section (lines 472-477):
To decipher the cellular cross-talks in these settings, the important immunological proteins involved in select pathways from IPA analysis, and their binding partners determined using CellphoneDB were deconvoluted on single-cell transcriptomics data. The proteins regulating these pathways are expressed by myeloid cells, intestinal epithelial and endothelial cells (Figure 8B) indicating an immune cross-talk amongst these cell types in small intestine.
- Updated legend for Figure 8 (lines 698-706):
Figure 8: Differentially regulated pathways in *LPS^{hi} Fatal* cohort and proposed mechanism of LPS associated mortality. Comparative IPA pathway analysis of LPS sub-groups (*LPS^{hi} F*, *LPS^{lo} F*, *LPS^{hi} NF* and *LPS^{lo} NF*) identified pathways that were differentially activated or inhibited. All analyses of LPS sub-groups were performed against an identical group of community children (n=251). (A) Heatmap visualizes z-scores of canonical pathways that were significantly activated (red) or inhibited (blue) in the *LPS^{hi} F* analysis but directionally differentially regulated in the analyses of the other 3 LPS groups. (B) Dot plot of genes of proteins involved in regulation of select immunological pathways differentially regulated in (A) and their binding partners determined using CellphoneDB deconvoluted in public single cell transcriptomic dataset. (C) Proposed mechanism of immune dysregulation in *LPS^{hi} F* cohort that may lead to increased mortality. Red font: upregulated pathway/protein level, green font: down-regulated

pathway/protein level. Red arrows: activation; green blunt-head arrows: inhibition of pathway. F = Fatal, NF = Non-fatal.

- Figure 9: Add a short summary to each sub-figure.

Response: Thank you for suggesting this. We have added the following figure legend to Figure 9 (lines 710-717):

Intestinal inflammation and increased barrier permeability were prevalent in community, as measured by fecal EED biomarkers. Comparative pathway analysis identified differentially regulated pathways supporting (A) exacerbated barrier dysfunction and (B) impaired growth (stunting) among hospitalized children. (C) Inflammatory and immunoregulatory protein expression was associated with pathways suggesting a shift toward cytokine dysregulation, and (D) broader disruption of signal transduction pathways. (E) Together, these findings suggest that LPS-associated pathophysiology in this setting likely involves immune dysfunction driven by chronic intestinal inflammation and its interaction with acute infections.

Reviewer #2 (Remarks to the Author):

Reviewer #3 (Remarks to the Author):

This manuscript investigates the association between plasma LPS levels, enteric biomarkers, and 90-day mortality in acutely ill children. Using a multicenter cohort, the study explores LPS as a marker of microbial translocation and systemic inflammation, with a focus on EED in low-resource settings. The findings suggest that elevated plasma LPS and inflammatory biomarkers correlate with increased mortality, independent of wasting status.

The study addresses a highly relevant and important topic. It focuses on childhood mortality in resource-limited settings. The investigation of LPS as a marker of microbial translocation is of interest given its potential as a predictor of mortality in acutely ill children. Additionally, the study benefits from its large multicenter cohort design. Nevertheless there are several concerns to be addressed:

Response: Thank you for acknowledging that the study addresses an important topic.

***Comment #1:** The manuscript offers little in terms of novel insights. The relationship between LPS, systemic inflammation, and mortality is well-established in the literature, particularly in conditions such as sepsis, malnutrition, and enteric dysfunction. The findings of this study are primarily descriptive and confirm existing knowledge rather than advancing it. There is minimal exploration of new biology or mechanisms, which limits the impact of the study*

Response # 1: We sincerely appreciate the reviewer's thoughtful comments and acknowledge that the relationship between LPS, systemic inflammation, and mortality has been previously explored in conditions such as sepsis, malnutrition enteropathy, and HIV. However, our study provides novel insights by examining this relationship in a population where environmental enteropathy (EED) is prevalent. In CHAIN, 59% of children were hospitalized with at least two common comorbidities (gastroenteritis, malaria, sepsis, LRTI, URTI, HIV, or severe acute malnutrition), and 21% had at least three. This underscores the limitations of studies conducted in more homogeneous clinical settings. The reviewer has rightly pointed out the heterogeneity of our study population, and we believe this heterogeneity strengthens our findings. This reflects the real-world complexity of EED, which often presents subclinically. These complex data require complex analysis and uni- and multivariate normalisation.

We agree with the reviewer that our novel findings should be articulated more clearly. To address this, we have added *Supplementary Methods 7* (below), which summarizes LPS-associated findings from previous studies in EED settings. Notably, only three studies in the context of EED have reported LPS-related outcomes. These studies primarily used plasma LPS as a marker of bacterial translocation in the context of acute illness and did not explore LPS-associated immunological pathways as a driver of mortality. Our finding that plasma LPS is associated with mortality in EED settings represents a novel contribution to the field. Additionally, while site-specific differences may independently influence mortality, we demonstrate that the association between plasma LPS and mortality persists even after adjusting for site differences (*Figure 2D*).

In a setting where malnutrition is a major driver of mortality, and where many interventions focus on nutritional outcomes, we show that plasma LPS is associated with mortality even when adjusting for anthropometric classification. This finding is important because it highlights a need to account for immunological pathway in interventions aimed at improving childhood mortality in this setting. We also validate previous findings showing that HIV and other biomarker for enteric dysfunction, such as fecal calprotectin and myeloperoxidase, are independently associated with mortality.

Our methods integrated systems biology approaches to investigate immunological pathways and supplemented our findings with single-cell transcriptomic deconvolution, thus offering a deeper understanding of these mechanisms in the context of EED – where access to intestinal samples for research is often limited. Our findings demonstrate that elevated plasma LPS and intestinal inflammation are also prevalent among community children and provide mechanistic insights into immunological pathways linked to mortality. We identified the 19 top proteins associated with plasma LPS in the most at-risk sub-group (*LPS^{hi} Fatal*) (*Supplementary Table 12*) and identify immunologically relevant pathways such as, VEGF signalling, IL-15 production, IL-12 signalling in macrophages, IL-10 signalling, PI3K/AKT and ERK/MAPK signalling, that are differentially regulated in sub-group comparisons. These are all important first steps to unravelling the immunological mechanisms at play in a complex, heterogenous setting.

Taken together, our key finding is that plasma LPS is associated with mortality even after adjusting for demographic characteristics, site differences and common comorbidities, including anthropometric classification, and that immune dysregulation emerges as an underlying mechanism in a heterogeneous population. We hope that these revisions clearly demonstrate the novel and highly clinically relevant aspect of our work were LPS is a biomarker of mortality.

Reference	Study Objective	Cohort Description	LPS-associated finding(s)	LPS and mortality finding
Mwape et al., 2017¹⁵ “Immunogenicity of Rotavirus Vaccine (Rotarix™) in Infants with Environmental Enteric Dysfunction.”	Assessed the relationships between EED biomarkers and rotavirus seroconversion after immunization with Rotarix. Study assessed LPS signaling biomarkers soluble CD14(sCD14) and Endotoxin Core IgG (EndoCab) ELISA assessment. Authors suggested sCD14 and EndoCab are biomarkers of gut leakage.	Retrospective cohort of 142 Zambian infants.	 No evidence that levels of sCD14 or EndoCab are associated with seroconversion status. 	No findings specific to LPS-associated mortality.
Amadi et al., 2017¹⁶ “Impaired Barrier Function and Autoantibody Generation in Malnutrition Enteropathy in Zambia.”	Assessed gut structure and function in children with Severe Acute Malnutrition (SAM) and persistent diarrhoea and adult comparison group. Study directly assessed plasma LPS and LPS signaling molecules, sCD14 and LPS Binding protein (LBP). Authors suggest that LPS is a biomarker of bacterial translocation.	Cross-sectional analysis of 34 children with SAM and persistent diarrhoea in a population from poor communities in Zambia, where stunting is prevalent. 61 adult controls were also investigated. Potential impact of HIV status was also assessed.	 Plasma LPS levels were higher in samples from participants with bacterial DNA detected by 16S RNA compared to those with no bacterial DNA detected. Plasma LPS was higher in adults and malnourished children compared to healthy children. Serum LBP was elevated in malnourished children compared to comparison groups. IGF-1 was inversely correlated with plasma LPS. Plasma LPS was inversely correlated with GLP-2 and this association was stronger in HIV-negative adults. TF3 immunoreactivity in duodenal aspirates was associated with reduced plasma LPS. HIV status was not associated with increased plasma LPS. 	No findings specific to LPS-associated mortality.
Kelly et al., 2010¹⁷ “Gastric and Intestinal Barrier Impairment in Tropical Enteropathy and HIV.”	Assessed impact of micronutrient supplementation on gut barrier function. Study assessed serum LPS, anti-LPS IgG and anti-LPS IgM. Authors used serum LPS, and other biomarkers, as markers of intestinal permeability.	Observational sub-studies nested in a RCT of daily supplementation in children in Zambia. 87 participants were enrolled in the sub-study of mucosal permeability.	 Serum LPS levels were not significantly different among supplementation or HIV status groups. Anti-LPS IgM was reduced in micronutrient recipients. No correlation between serum LPS and anti-LPS antibodies levels. Log-transformed Xylose recovery was negatively associated with log-transformed anti-LPS IgG Log-transformed TNFRp55 was associated with anti-LPS IgG and anti-LPS IgM. Authors suggested that background EED may facilitate translocation by other mechanisms than paracellular permeability. 	No findings specific to LPS-associated mortality.

Supplementary Methods 7: Findings from studies on LPS, Environmental Enteric Dysfunction and Mortality

This table summarizes the LPS-associated and LPS-mortality-associated findings from previous studies in Environmental Enteric Dysfunction (EED) settings. Three relevant studies were identified through a PubMed search on March 14, 2025, using the following terms: ("EED" OR "Environmental Enteropathy" OR "Environmental Enteric Dysfunction" OR "Environmental Enteric Disease" OR "Tropical Enteropathy" OR "Tropical Enteric Dysfunction") AND ("lipopolysaccharide" OR "LPS" OR "Endotoxin") AND ("mortality" OR "survival" OR "death") AND ("human" OR "children" OR "infant").

	CORRELATION COEFFICIENT	ADJUSTED P-VALUE
PLASMA LPS	1	0
GOLM2	-0.403041241	0.003034158
SELPLG	-0.457752658	0.000358588
ADH5	0.402791084	0.003060736
DNAJB2	0.401941138	0.003152971
TRAF4	-0.409055307	0.002452456
CD7	-0.423650484	0.00143083
NRBP1	0.413019831	0.002125059
KCNG4	0.408410818	0.002509719
CD247	0.420418345	0.001616229
S100A16	0.410918297	0.002293413
SMS	0.424430778	0.001389172
DCTN6	0.421705189	0.001539849
C1QL3	-0.411435732	0.002250778
ATXN10	0.439076953	0.000782596
ICOSLG	-0.474097188	0.000172833
SIRPB1	-0.409268394	0.002433795
CLEC6A	-0.430607531	0.00109503
SELPLG.1	0.410799986	0.002303327
LILRA5.1	-0.408252243	0.002523977

Supplementary Table 12: Correlation coefficients and p-values of proteins most associated with plasma in the *LPS^{hi} Fatal* sub-group.

Correlation coefficients and p-values of the 19 proteins most associated with plasma LPS among over 7300 SomaScan proteins in the *LPS^{hi} Fatal* sub-group. Correlation coefficients were inverse-proportionally weighted to account for selection biases. Only correlations with an absolute value of $r \geq 0.4$ and p-value < 0.05 are displayed. All p-values were adjusted for false discovery rates using the Benjamini-Hochberg (BH) method, based on the full SomaScan dataset of over 7300 proteins.

Comment #2: *The cohort comprises a highly heterogeneous group of acutely ill children with varying diseases, trajectories, and comorbidities, yet the manuscript lacks detailed stratification or tabulation of these conditions. There were little evaluation regarding the HIV, RTI, sepsis, malaria, or gastroenteritis. Some of the analyses were adjusted for these conditions, there were no detail on which which analyses were adjusted or how. A table summarizing the disease distribution, their status (acute or chronic, primary or secondary or complication diagnoses) and stratification is necessary to understand the heterogeneity of the cohort and the validity of the findings. This lack of clarity raises concerns about potential bias and confounding in the analyses.*

Response #2: Thank you for raising this important point. We have revised the *Methods* section to more adequately indicate where adjustments were made in analyses. Additionally, we have added a new table (*Table 1*) which provides comparisons of comorbidity frequencies, fecal biomarkers, plasma barrier function proteins, entero-pathogen levels and frequencies and LPS signal transduction expression between *admission* and *community children*.

Characteristics	Admission Weighted N = 1,894 ^a Unweighted N = 638	Community children Weighted N = 251 ^a Unweighted N = 251	p-value ^b
Plasma LPS (E.U./ml) (Mean (SD))	4.40 (5.25)	4.47 (5.34)	>0.9
Demographics			
Age group			0.2
<6 months	361 / 1,894 (19%)	41 / 251 (16%)	
6-11 months	690 / 1,894 (36%)	81 / 251 (32%)	
12 months & above	844 / 1,894 (45%)	129 / 251 (51%)	
Admission age (Mean (SD))	11.47 (5.57)	12.48 (6.11)	0.049
Sex			0.047
Female	742 / 1,894 (39%)	118 / 251 (47%)	
Male	1,152 / 1,894 (61%)	133 / 251 (53%)	
Continent			0.032
Africa	1,085 / 1,894 (57%)	165 / 251 (66%)	
Asia	809 / 1,894 (43%)	86 / 251 (34%)	
Comorbidity frequencies			
Malaria History	337 / 1,894 (18%)	11 / 251 (4.4%)	<0.001
Sepsis	295 / 1,894 (16%)	0 / 251 (0%)	<0.001
Gastroenteritis	1,142 / 1,872 (61%)	0 / 251 (0%)	<0.001
URTI	98 / 1,894 (5.2%)	0 / 251 (0%)	0.060
LRTI	774 / 1,894 (41%)	0 / 251 (0%)	<0.001
HIV status			0.2
Negative	1,844 / 1,894 (97%)	248 / 251 (99%)	
Positive	50 / 1,894 (2.6%)	3 / 251 (1.2%)	
TB	3 / 1,701 (0.2%)	0 / 221 (0%)	0.6
Breastfeeding status	1,428 / 1,894 (75%)	202 / 251 (80%)	0.12
Presence of at least 1 Entero-pathogen			
Viruses	776 / 1,702 (46%)	53 / 222 (24%)	<0.001
Parasites	517 / 1,770 (29%)	56 / 231 (24%)	0.2
Gram-negative Bacteria	1,579 / 1,820 (87%)	206 / 241 (85%)	0.6
Entero-pathogen Quantification (Mean (SD))^c			
Enterococcal E. coli	28.85 (6.31)	30.31 (5.32)	0.007
Enterotoxigenic E. coli	32.79 (4.69)	33.86 (2.91)	0.008
Shigella toxin-producing Enterotoxigenic E. coli	33.51 (4.06)	34.63 (1.71)	<0.001
V. cholerae	34.79 (1.53)	35.00 (0.02)	0.006
CTX-M-producing bacteria	27.24 (6.79)	30.62 (4.99)	<0.001
mphA Azithromycin Resistance	23.08 (6.47)	27.36 (5.69)	<0.001
X16s	13.90 (3.28)	15.25 (3.71)	<0.001
Cryptosporidium	33.90 (3.46)	34.35 (2.58)	0.015
Adenovirus 40/41	34.22 (3.19)	34.71 (1.59)	0.005
Rotavirus	33.01 (4.26)	34.92 (0.82)	<0.001
Sapovirus	34.64 (1.65)	34.89 (0.77)	0.029
Fecal Biomarkers of EED (Mean (SD))			
Fecal MPO ng/ml	4,105.51 (4,584.89)	3,708.52 (3,552.86)	0.7
Fecal Calprotectin (> 12 months) (µg/ml)	698.85 (1,086.33)	502.96 (1,192.81)	0.13
Fecal AAT (non-acute diarrhoea) (µg/ml)	448.57 (557.01)	341.36 (372.63)	0.30
Barrier Function Biomarker (Mean (SD))^d			
Plasma Zonulin	46,479.64 (29,948.42)	33,384.53 (19,464.19)	<0.001
Plasma Diamine Oxidase	4,776.11 (4,226.63)	6,809.30 (3,594.82)	<0.001
Plasma FABP2	3,208.20 (2,915.06)	2,746.65 (1,951.33)	0.3
Plasma CDH1.1	1,468.48 (2,669.83)	1,602.89 (2,216.58)	0.005
Plasma OCLN	960.71 (291.05)	908.77 (166.60)	0.005
Plasma ZO-1	719.87 (1,718.78)	788.77 (1,570.68)	<0.001
Systemic Immune Activation and Inflammation Biomarkers (Mean (SD))^d			
Plasma Heparin-Binding Protein	24,326.27 (11,449.57)	19,629.43 (8,129.72)	<0.001
Plasma TREM1	2,168.66 (1,457.04)	1,691.83 (515.93)	<0.001
Plasma SOCS-3	1,480.93 (2,312.10)	1,305.70 (440.17)	0.074
Systemic Inflammatory Biomarkers (Mean (SD))^d			
Plasma CRP	69,067.92 (35,630.07)	32,055.93 (28,576.09)	<0.001
Plasma SAA1	72,275.72 (72,226.17)	17,974.93 (35,545.88)	<0.001
Plasma MPO	27,618.30 (14,306.43)	16,296.82 (5,131.00)	<0.001
Plasma Calprotectin	1,773.65 (1,065.01)	1,749.45 (3,856.34)	<0.001
Plasma AAT	50,645.94 (15,708.32)	33,395.31 (9,471.53)	<0.001

^aN = inverse proportionally weighted total within CHAIN cohort (n=3101); ^bDesign-based Kruskal-Wallis test, Pearson's X²; Rao & Scott adjustment; ^cThreshold Cycle (Ct) from quantitative PCR (qPCR) assay; E.U./ml = Endotoxin Unit per ml; ^dRFU = Relative Fluorescence Unit

Table 1: Demographic, Entero-pathogen, Fecal Biomarkers and Plasma Barrier and Systemic Inflammation Profiles of Hospitalized vs. Community Children

Demographics (age, sex, continent), fecal biomarkers (MPO, calprotectin, AAT), and plasma LPS concentrations are comparable between Admission and Community children. In contrast, the presence of comorbidities (malaria, sepsis, gastroenteritis, LRTI), barrier protein expression (zonulin, diamine oxidase) and plasma systemic inflammatory (CRP, SAA1, MPO, calprotectin, AAT) and LPS signal transduction biomarkers (CD14, soluble CD14, LPS binding protein) differ significantly between the two cohorts. Both unweighted and inverse proportionally weighted totals (N) are presented. Only entero-pathogen with significant difference in expression between the cohorts are presented. See Supplementary Table 7C for all pathogen levels. LPS = lipopolysaccharide, MPO = myeloperoxidase, AAT = Alpha-1-antitrypsin, FABP2 = Intestinal-type fatty acid-binding protein, TLR4 = Toll-like Receptor 4.

We have updated the *Methods* section to explicitly specify where our models have been adjusted for covariates. However, it is our belief that many comorbidities likely serve as mediator (signal 2) of LPS-associated mortality. This assumption is based on several findings: 1) levels of plasma LPS are similar between the *admission cohort* and *community children* and 2) *common comorbidities (such as gastroenteritis, LRTI, URTI, malaria and sepsis) are not independently associated with mortality in this setting when adjusted for each other.* This suggests that common comorbidities likely mediate, rather than confounds, mortality outcomes and for this reason we have not adjusted proteomics and pathway models for comorbidities, for fear that this may mask their role in LPS-driven pathophysiology. For future studies, mediation analyses would be beneficial to further elucidate the role of each comorbidity in mortality mechanisms.

We have added a new figure (*Figure 2D*) to highlight that plasma LPS is significantly associated with mortality ($p = 0.002$) even when adjustments are made for other common comorbidities (*Anthropometric classification, Malaria, Gastroenteritis, Sepsis, LRTI, URTI, HIV status*) and demographic factors (*sex, age, site*). This suggests that while anthropometric classification, HIV status, and study site may independently influence mortality, the relationship between plasma LPS and mortality may not be strictly organ-specific. A systems biology approach could therefore provide valuable insights into these associations.

Figure 2D: Plasma LPS is associated with mortality

Elevated plasma LPS, more severe anthropometric classification, HIV status and site differences are independently associated with increased risk of mortality in admissions children as assessed by an inverse proportionally weighted cox proportional hazard model. Forest plot shows log hazard ratios with 95% confidence intervals and p-values. Analysis was weighted to account for selection biases. CoxPH model was adjusted for demographics and common comorbidities that affect this population. The model satisfied hazards and linearity assumptions.

Comment #3: Although the manuscript categorizes children into groups based on MUAC (SAM / MAM / normal), the assessment appears overly simplistic. The focus on MUAC and IGF markers does not provide a comprehensive evaluation of nutritional status. Were

Response #3: Thank you for highlighting a need to more clearly define how nutritional status is defined within our cohort. We have updated any references to ‘nutritional status’ with ‘anthropometric classification’ and have added clarifying information in the *Methods* section to denote that ‘anthropometric classification’ is defined by MUAC and age range and this is consistent with provision of care for children assessed as severely malnourished. While we acknowledge that anthropometric classification is an important variable in mortality outcomes in this population, we offer that this is not a central objective of this study. In fact, one of our most important findings is that plasma LPS is associated with mortality, regardless of anthropometric classification.

Comment # 4: There needs to be more explanation on how some thresholds were selected. The threshold for LPS (11.58) is not adequately explained. The manuscript does not explain why this specific cutoff was chosen or whether it is clinically meaningful. Furthermore, were

these sensitivity analyses being performed -

Response # 4: Thank you for your valuable feedback. We appreciate your comment on the threshold for plasma LPS (11.58 E.U./ml) to clarify its clinical relevance. As far as we are aware, there are no established cut-off for clinical utility in this cohorts. We hypothesized that children with higher LPS at admission (possibly due to microbial translocation, bacterial overgrowth, impaired clearance or intestinal architectural disruption) are at higher risk of poor outcomes post-discharge and thus provides a sound basis for categorizing our population. We have expanded the *Methods* section with additional explanation of the cut-point selection process and its clinical implications. This assessment utilizes the *Maxstat* method which identifies the optimal cut-off candidate through maximal log-rank statistics and uses robust bootstrapping technique to validate this result. We recognize the importance of validating this threshold in different cohorts.

Change made:

- Update to the *Supplementary Methods 4:*

Cut point of plasma LPS. The 'surv_cutpoint' function from the 'survminer' R package was utilized to derive optimal unweighted cut-off points for continuous variables. This function uses maximally selected log-rank statistics (*Maxstat* method), which identifies the most statistically significant cut-off point by maximizing the difference in survival between two groups based on plasma LPS concentration. The cut-point is clinically useful for identifying thresholds that stratify patients into high and low plasma LPS groups, based on the greatest separation of survival outcomes. Briefly, the *Maxstat* method evaluates numerous candidate cut-off points for plasma LPS and selects the optimal threshold (11.58 E.U./ml in this case) by assessing the maximum value of the log-rank statistic for each candidate cut-off. The selected cut-point was validated through Kaplan-Meier survival curve visualization, which demonstrated significant differences in survival between patients with high vs. low plasma LPS concentrations (*Supplementary Figure 1A*). Additionally, the *Maxstat* method employs bootstrapping to assess the robustness of the selected cut-point and to provide confidence intervals for the cut-off. This resampling approach ensures the reliability of the identified threshold in predicting survival outcomes.

Supplementary Figure 1A: LPS cutoff analysis of Log-Rank Statistic for 90-day survival

The optimal cut point of LPS based on maximally selected rank statistics of 90-day survival outcome is 11.58 E.U./ml for the CHAIN LPS study. The maximally selected rank statistics stratifies the cohort into two groups and tests all possible cutpoints. The standardized statistics of the cut point that most significantly separates the groups is selected and this is then used to define a high and low expression within the cohort.

With regards to mortality at different time points: we have provided the justification for 90-day mortality in *Supplementary Methods 4*.

Change made:

- Update to the *Supplementary Methods 4*:

Justification of 90-day mortality. The impact of plasma LPS on the immune system varies between acute and chronic exposure. Assessing 90-day mortality captures both acute and prolonged LPS effects, enhancing our understanding of its role in survival. Since the CHAIN NCC includes inpatient deaths beyond 30 days, 90-day mortality was useful for evaluating LPS-associated chronic inflammation, immune dysregulation, and immune modulation over time. Studies of cardiac fibrosis in animal models have shown elevated mortality following moderate LPS exposure for 60 to 90 days¹⁸, and there were no differences in sensitivity or specificity in predictive value of plasma LPS in survival outcomes at 30-day, 60-day or 90-day time points (*Supplementary Methods 8*).

Additionally, with a subsequent post-discharge mortality peak within the CHAIN NCC around 150 days, this timeframe facilitates the analysis of complex mortality trends without the confounding effects of events beyond 90 days.

Supplementary Methods 8: Time-dependent ROC curves for plasma LPS concentration in predicting survival outcomes.

The ROC curves illustrate the predictive ability of unadjusted plasma LPS concentration for survival outcomes at 30, 60, and 90 days. The Area Under the Curve (AUC) values with 95% confidence intervals are displayed in the legend for each time point. The diagonal dashed line represents the reference line for no discrimination (AUC = 0.5). Statistical comparisons of AUC values between time points are provided, with p-values calculated using a Z-test to assess differences between AUCs at 30 or 60 days versus 90 days.

Comment #5: The fecal analysis appears narrow in scope, focusing on E. coli (EAEC3 strains) and azithromycin-resistant bacteria. The rationale for focusing on these specific strains is unclear, and the manuscript does not explain why other enteric pathogens or broader microbiome diversity were not assessed. Additionally, there is no mention of whether

AMR strains beyond azithromycin resistant bacteria were analyzed deeper insights into microbial translocation and its impact on mortality.

Response #5: We thank the reviewer for this insightful comment and agree that providing additional details on the assessment of entero-pathogens would enhance the understanding of pathogen expression among the groups. The TaqMan Array Card (TAC) (Life Technologies) was employed to detect 36 entero-pathogens, including 18 bacteria, 6 viruses, and 12 parasites, along with 2 antimicrobial resistance genes and bacterial load. We have revised the *Methods* section to elaborate specifics on the detection of entero-pathogens.

Change made:

- Update to the *Method* section (lines 173-191):

Entero-pathogen Quantification. The Taqman Array Card (TAC, Life technologies, USA) assay was used for detection and quantification of 36 entero-pathogens, 2 antimicrobial resistance genes and non-specific bacterial load. A threshold cycle (Ct) cutoff of 35 was used to denote positivity for all the targets and Ct was used for regression and correlation models involving TAC data. The TAC panel included the detection of:

- **Bacteria:** *Aeromonas* spp., *Campylobacter pan*, *Campylobacter jejuni*, Enteroaggregative *Escherichia coli* (EAEC), atypical *Enteropathogenic E. coli* (aEPEC), Enteropathogenic *Escherichia coli* (EPEC), Enterotoxigenic *E. coli* producing heat-labile toxin (LT) (LT-EPEC), Enterotoxigenic *E. coli* producing heat-stable toxin (ST) (ST-EPEC), typical Enteropathogenic *E. coli* (tEPEC), Enterotoxigenic *Escherichia coli* (EPEC), *Mycobacterium tuberculosis*, Shiga toxin-producing *Escherichia coli* (STEC), *Helicobacter pylori*, *Plesiomonas shigelloides*, *Salmonella* spp., *Shigella* spp., *Vibrio cholerae*, *Clostridium difficile*.
- **Total bacterial load:** 16S rRNA gene (*XI6s*) of predefined bacterial pathogen
- **Antimicrobial resistance (AMR) genes:** Beta-lactam (ESBL) resistance: *blaCTX-M*, Macrolide (Azithromycin) resistance: *mphA*
- **Viruses:** *Adenovirus*, *Astrovirus*, *Norovirus GI*, *Norovirus GII*, *Rotavirus*, *Sapovirus*
- **Parasites:** *Ancylostoma*, *Ascaris lumbricoides*, *Cyclospora cayetanensis*, *Cryptosporidium* spp., *Entamoeba histolytica*, *Enterocytozoon bieneusi*, *Enterocytozoon intestinalis*, *Giardia lamblia*, *Isospora belli*, *Necator americanus*, *Strongyloides stercoralis*, *Trichuris trichiura*.

The complete entero-pathogen results for both *admission* and *community children* are presented in *Supplementary Table 11*. *Table 1* highlights only those pathogens with significant differences in detection between the two groups (green rows in *Supplementary Table 11*). Taken together, our findings show that the presence and quantity of entero-pathogens are higher in the *admission* cohort compared to the *community children*. *We found that quantification of Enteroaggregative E. coli, Enterotoxigenic E. coli, Shigella toxin-producing Enterotoxigenic E. coli, V. cholerae, CTX-M-producing and mphA Azithromycin Resistance bacteria were significantly elevated (p < 0.01) in the admission cohort.*

Characteristics	Admission Weighted N = 1,894 ^a Unweighted n = 638	Community children Weighted N = 251 ^a Unweighted n = 251	p-value ^b
Enteropathogen frequencies			
Viruses	776 / 1,702 (46%)	53 / 222 (24%)	<0.001
Parasites	517 / 1,770 (29%)	56 / 231 (24%)	0.2
Gram-negative Bacteria	1,579 / 1,820 (87%)	206 / 241 (85%)	0.6
Enteropathogens Quantification (Mean (SD))^c			
Aeromonas	34.95 (0.50)	34.88 (0.90)	0.4
Campylobacter_pan	32.08 (4.87)	32.72 (3.75)	>0.9
Campylobacter_jejuni_coli	33.19 (4.02)	33.83 (3.06)	0.11
EAEC	28.85 (6.31)	30.31 (5.32)	0.007
tEPEC	33.62 (4.03)	34.24 (2.50)	0.069
aEPEC	33.77 (3.02)	33.92 (2.90)	0.6
ETEC	32.79 (4.69)	33.86 (2.91)	0.008
ST_ETEC	33.51 (4.06)	34.63 (1.71)	<0.001
LT_ETEC	34.44 (2.33)	34.29 (2.34)	0.2
H_pylori	34.96 (0.38)	35.00 (0.02)	0.054
Plesiomonas	34.91 (0.75)	34.94 (0.55)	0.6
Salmonella	34.86 (1.01)	34.97 (0.37)	0.068
Shigella_EIEC	33.66 (4.04)	34.13 (2.67)	0.6
STEC	34.96 (0.36)	34.91 (0.71)	0.4
V_cholerae	34.79 (1.53)	35.00 (0.02)	0.006
CTX_M	27.24 (6.79)	30.62 (4.99)	<0.001
mphA Azithromycin Resistance	23.08 (6.47)	27.36 (5.69)	<0.001
X16s	13.90 (3.28)	15.25 (3.71)	<0.001
C_difficile	34.95 (0.67)	34.94 (0.49)	0.3
Ascaris	35.00 (0.06)	35.00 (0.00)	0.2
Cyclospora	35.00 (0.00)	35.00 (0.02)	0.3
E_bieneusi	34.83 (1.05)	34.85 (0.93)	0.7
E_intestinalis	34.96 (0.52)	34.99 (0.10)	>0.9
Giardia	34.07 (2.81)	34.04 (2.76)	0.8
Iso spora	35.00 (0.08)	35.00 (0.00)	0.084
Necator	34.99 (0.17)	35.00 (0.01)	0.9
Cryptosporidium	33.90 (3.46)	34.35 (2.58)	0.015
Adenovirus_40_41	34.22 (3.19)	34.71 (1.59)	0.005
Astrovirus	34.69 (1.73)	34.87 (1.28)	0.11
Norovirus_GI	34.92 (0.74)	34.82 (0.98)	0.2
Norovirus_GII	34.28 (2.54)	34.60 (1.70)	0.4
Rotavirus	33.01 (4.26)	34.92 (0.82)	<0.001
Sapovirus	34.64 (1.65)	34.89 (0.77)	0.029
M_tuberculosis	34.99 (0.12)	35.00 (0.00)	0.2

^aN = inverse proportionally weighted total within CHAIN cohort (n=3101); ^bDesign-based Kruskal-Wallis test; Pearson's X²: Rao & Scott adjustment; ^cThreshold Cycle (Ct) from quantitative PCR (qPCR) assay; E.U./ml = Endotoxin Unit per ml

Supplementary Table 11: Entero-pathogen profiles of Admission cohort vs. Community children

Entero-pathogen presence and quantification are significantly different between Admission and Community children. Comparisons with significant differences are highlighted in green. The following entero-pathogens were excluded from analysis because all tests in Admission or Community children were negative or missing: *Ancylostoma*, *E.histolytica*, *Strongyloides* and *Trichuris*. Both unweighted (*n*) and inverse proportionally weighted (*N*) totals are presented.

Comment #6: The authors mentioned about intestinal permeability and translocation. Were biomarkers related to intestinal barrier, e.g. zonulin, villous atrophy, intestinal inflammation - beyond faecal calprotectin?

Response #6: Thank you for raising this point. We have added data to show that the admission cohort had poor barrier function compared to the community children based on expression of plasma zonulin, FABP2, diamine oxidase, oxidase, cadherin-1, occludin, and ZO-1 (Table 1). These results suggest that the admission cohort presented with comparable enterocyte damage (FABP2) but increased intestinal permeability (elevated zonulin), more compromised intestinal mucosal integrity (reduced diamine oxidase), and weakened adhesion and greater tight junction breakdown (reduced cadherin-1 and ZO-1).

Besides fecal calprotectin, fecal myeloperoxidase is also a marker of intestinal inflammation, and we show that levels of similar between the admission and community controls (Table 1) but are elevated in the *LPS^{hi} Fatal* sub-group compared to the *LPS^{hi} Non-Fatal* sub-group (Table 2).

Characteristic	LPS ^{hi} Fatal Weighted N = 34 ^a Unweighted n = 39	LPS ^{hi} Non-Fatal Weighted N = 145 ^a Unweighted n = 46	LPS ^{lo} Fatal Weighted N = 98 ^a Unweighted n = 125	LPS ^{lo} Non-Fatal Weighted N = 1,609 ^a Unweighted n = 425	p-value ^b
Plasma LPS (E.U./ml) (Mean (SD))	16.7 (4.6)	17.3 (4.6)	3.6 (3.2)	3.0 (3.1)	<0.001
Anthropometric classification					<0.001
Acute A (SAM)	18 / 34 (52%)	48 / 145 (33%)	64 / 98 (66%)	341 / 1,609 (21%)	
Acute B (MAM)	5 / 34 (15%)	29 / 145 (20%)	14 / 98 (14%)	258 / 1,609 (16%)	
Acute C (Normal)	11 / 34 (33%)	68 / 145 (47%)	20 / 98 (20%)	1,009 / 1,609 (63%)	
Admission age (Mean (SD))	10.2 (5.6)	10.8 (4.5)	11.0 (5.6)	11.6 (5.6)	0.5
Sex					0.008
Female	19 / 34 (56%)	83 / 145 (57%)	47 / 98 (48%)	591 / 1,609 (37%)	
Male	15 / 34 (44%)	62 / 145 (43%)	51 / 98 (52%)	1,017 / 1,609 (63%)	
HIV status					<0.001
negative	31 / 34 (90%)	139 / 145 (96%)	86 / 98 (88%)	1,579 / 1,609 (98%)	
positive	3 / 34 (9.9%)	5 / 145 (3.6%)	11 / 98 (12%)	30 / 1,609 (1.8%)	
Entero-pathogen Quantification^c					
Enteroaggregative E.coli	25.8 (5.8)	30.5 (5.9)	27.5 (5.9)	28.8 (6.4)	0.001
mphA Azithromycin Resistance	21.2 (5.2)	26.7 (7.1)	21.4 (5.9)	22.9 (6.4)	<0.001
X16s	12.2 (1.9)	14.0 (3.5)	13.0 (3.0)	14.0 (3.3)	<0.001
Rotavirus	31.9 (5.7)	33.2 (4.7)	34.5 (2.3)	32.9 (4.3)	<0.001
Fecal Biomarkers of EED (Mean (SD))					
Fecal Calprotectin	852.4 (1,210.2)	328.3 (462.0)	895.7 (1,387.5)	586.3 (951.8)	0.003
Barrier Function Biomarker (Mean (SD))^d					
Plasma Zonulin	33,736.2 (32,243.8)	38,441.4 (30,298.5)	42,406.5 (35,107.7)	47,717.2 (29,380.6)	0.021
Plasma Diamine Oxidase	3,377.3 (3,205.3)	4,433.3 (2,628.9)	4,808.8 (12,543.9)	4,834.4 (3,252.7)	<0.001
Plasma CDH1.1	955.5 (409.8)	1,130.3 (474.6)	1,278.8 (1,012.4)	1,521.2 (2,872.5)	0.007
Plasma CDH1.2	22,900.8 (8,445.9)	27,482.1 (7,827.2)	23,456.4 (8,073.5)	25,908.1 (6,163.6)	0.004
Plasma OCLN	1,053.7 (386.2)	1,012.9 (520.2)	1,029.2 (331.7)	949.9 (254.8)	0.008
Plasma ZO-1	446.2 (126.4)	621.5 (615.5)	518.0 (510.9)	746.7 (1,846.2)	<0.001
Plasma TNFAIP3	1,459.9 (451.2)	1,305.2 (306.8)	1,627.1 (1,337.0)	1,657.9 (1,059.6)	0.003
Systemic Immune Activation and Inflammation Biomarkers (Mean (SD))^d					
Plasma Heparin-Binding Protein	31,240.7 (10,668.0)	26,322.3 (13,020.1)	29,175.7 (19,256.3)	23,707.3 (10,530.0)	<0.001
Plasma TREM1	4,411.7 (4,014.6)	2,601.7 (1,686.2)	2,908.0 (2,404.2)	2,037.7 (1,177.0)	<0.001
Systemic Inflammation (Mean (SD))^d					
Plasma MPO	35,788.1 (20,177.6)	24,180.2 (12,891.4)	34,692.0 (22,355.8)	27,326.2 (13,461.8)	0.002
LPS Signal transduction (Mean (SD))^d					
Plasma CD14	2,397.6 (1,344.7)	1,607.0 (605.7)	2,364.8 (1,620.1)	1,673.0 (617.4)	<0.001
Plasma LPS Binding Protein	77,694.5 (40,981.5)	53,099.1 (31,694.3)	76,151.9 (38,968.7)	67,794.0 (33,009.2)	0.001
Plasma TLR4	379.6 (176.7)	438.6 (196.9)	673.4 (1,266.3)	511.7 (1,053.9)	0.019

^aN = inverse proportionally weighted total within CHAIN cohort (n=3101); ^bDesign-based Kruskal-Wallis test, Pearson's X²: Rao & Scott adjustment; ^cThreshold Cycle (Ct) from quantitative PCR (qPCR) assay; ^dRFU = Relative Fluorescence Units; E.U./ml = Endotoxin Unit per ml

Table 2: Key differences in pathogen quantification and inflammatory biomarker expression between *LPS-mortality* sub-groups

Quantification of entero-pathogens (X16s, EAEC, mphA azithromycin resistance genes, and rotavirus), fecal calprotectin, plasma MPO, biomarkers of barrier function (zonulin, diamine

oxidase, CDH1, OCLN, ZO-1, and TNFAIP3), immune activation (heparin-binding protein and TREM-1), and LPS signal transduction (CD14, LBP, and TLR4) revealed significant differences between LPS–mortality sub-groups in unadjusted comparisons.

Specifically, levels of XI6s, EAEC, mphA, plasma diamine oxidase, zonulin, MPO, CD14, heparin-binding protein, TREM-1, and CDH1 differed significantly between the LPS^{hi} F group and at least one other LPS–mortality sub-group in both unadjusted and adjusted (Supplementary Table 14-15) analyses. CTX-M–producing bacteria were also detected at significantly higher levels for the LPS^{hi} F vs. LPS^{lo} NF sub-group in adjusted comparisons. Adjusted analyses controlled for admission age, sex, study site, and HIV status. F = Fatal; NF = Non-fatal.

Comment #7: The pathway analysis relies heavily on public datasets and does not provide meaningful mechanistic insights. The lack of independent experimental validation significantly weakens the credibility of the proposed pathways. Furthermore, the discussion around therapeutic targets, such as bevacizumab and tocilizumab, is highly speculative. There is also no exploration of the directional effects of these pathways beyond up- or down-regulation.

Response #7:

We have used our SOMAlogics proteomics data from our cohort for pathway analysis and used the public dataset to deconvolute the cell types possibly expressing the proteins involved in these pathways. Given that the subjects in the study are children from low-resource settings, the availability of actual tissue samples is very limited. Under these constraints, use of a public dataset with EED tissue samples proves to be an efficient source of information to understand the possible mechanism of LPS mediated mortality in these settings. The main aim of the study is to correlate the mortality and plasma LPS levels in low-resource settings. The in house single cell analysis or experimental validation of molecular mechanism would constitute a project on its own. We agree that the hypothesized use of specific therapies is speculative. We have now refrained from mentioning specific drugs for potential treatment and restricted to mentioning that PI3K/AKT pathway can be a potential therapeutic target, pending clinical trial studies. The possible interactions of pathways detected by IPA has been shown in figure 9. We have elaborated on it in the discussion for clarity.

Change made:

- Update to Discussion section (lines 547-559):
Alongside VEGF upregulation and reduced growth hormone signaling (Figure 9, panels A and B), comparative pathway analysis indicated that the LPS^{hi} F cohort exhibited cytokine dysregulation including increased IL-10 signalling, which could be correlated to reduced IL-12 and IL-15 signalling and increased anti-inflammatory M2 macrophage polarization (Figure 9, panel C). This was accompanied with increased STAT3, PI3K/AKT and inhibited PTEN pathways (Figure 9, panel D). These together suggest that the dysregulation of immune cytokines could contribute to increased mortality in the study settings (Figure 9, panel E). The IL-10 and VEGF pathways are known to activate PI3K/AKT signalling, which was also differentially upregulated in LPS^{hi} F cohort. Coupled

with inhibition of PTEN, which inhibits PI3K/AKT pathway, this axis may play a crucial role in the mortality mechanism of some of the children studied in this cohort (*Figure 9, panel D*) and warrants further investigation as a potential therapeutic target. Since this study did not assess treatment courses, it cannot determine how treatment may have influenced mortality outcomes. Randomized clinical trials are needed to validate these findings.

References:

- 1 Crane, R. J. *et al.* Cessation of exclusive breastfeeding and seasonality, but not small intestinal bacterial overgrowth, are associated with environmental enteric dysfunction: A birth cohort study amongst infants in rural Kenya. *eClinicalMedicine* **47**, 101403 (2022). <https://doi.org:10.1016/j.eclinm.2022.101403>
- 2 Berkley, J. *et al.* Assessment of severe malnutrition among hospitalized children in rural Kenya: comparison of weight for height and mid upper arm circumference. *JAMA* **294**, 591-597 (2005). <https://doi.org:10.1001/jama.294.5.591>
- 3 Goossens, S. *et al.* Mid-upper arm circumference based nutrition programming: evidence for a new approach in regions with high burden of acute malnutrition. *PLOS ONE* **7**, e49320 (2012). <https://doi.org:10.1371/journal.pone.0049320>
- 4 in *Guideline: Updates on the Management of Severe Acute Malnutrition in Infants and Children WHO Guidelines Approved by the Guidelines Review Committee* (2013).
- 5 in *Pocket Book of Hospital Care for Children: Guidelines for the Management of Common Childhood Illnesses WHO Guidelines Approved by the Guidelines Review Committee* (2013).
- 6 Beckmann, G. T. & Ruffer, A. *Mikroökologie des Darmes: Grundlagen, Diagnostik, Therapie.* (Schlütersche, 2000).
- 7 Saiki, T. Myeloperoxidase concentrations in the stool as a new parameter of inflammatory bowel disease. *Kurume Med J* **45**, 69-73 (1998). <https://doi.org:10.2739/kurumemedj.45.69>
- 8 Kolho, K. L. & Alfthan, H. Concentration of fecal calprotectin in 11,255 children aged 0-18 years. *Scand J Gastroenterol* **55**, 1024-1027 (2020). <https://doi.org:10.1080/00365521.2020.1794026>
- 9 Kummerlowe, C. *et al.* Single-cell profiling of environmental enteropathy reveals signatures of epithelial remodeling and immune activation. *Science Translational Medicine* **14**, eabi8633 (2022). <https://doi.org:10.1126/scitranslmed.abi8633>
- 10 Kosek, M. *et al.* Fecal markers of intestinal inflammation and permeability associated with the subsequent acquisition of linear growth deficits in infants. *Am J Trop Med Hyg* **88**, 390-396 (2013). <https://doi.org:10.4269/ajtmh.2012.12-0549>
- 11 Hasan, M. M. *et al.* Gut biomolecules (I-FABP, TFF3 and lipocalin-2) are associated with linear growth and biomarkers of environmental enteric dysfunction (EED) in Bangladeshi children. *Sci Rep* **12**, 13905 (2022). <https://doi.org:10.1038/s41598-022-18141-8>
- 12 Kosek, M. N. Causal Pathways from Enteropathogens to Environmental Enteropathy: Findings from the MAL-ED Birth Cohort Study. *EBioMedicine* **18**, 109-117 (2017). <https://doi.org:10.1016/j.ebiom.2017.02.024>
- 13 Burgueno, J. F. & Abreu, M. T. Epithelial Toll-like receptors and their role in gut homeostasis and disease. *Nat Rev Gastroenterol Hepatol* **17**, 263-278 (2020). <https://doi.org:10.1038/s41575-019-0261-4>
- 14 Butcher, S. K., O'Carroll, C. E., Wells, C. A. & Carmody, R. J. Toll-Like Receptors Drive Specific Patterns of Tolerance and Training on Restimulation of Macrophages. *Frontiers in immunology* **9**, 933 (2018). <https://doi.org:10.3389/fimmu.2018.00933>
- 15 Mwape, I. *et al.* Immunogenicity of rotavirus vaccine (Rotarix™) in infants with environmental enteric dysfunction. *PLOS ONE* **12**, e0187761 (2017). <https://doi.org:10.1371/journal.pone.0187761>

- 16 Amadi, B. *et al.* Impaired Barrier Function and Autoantibody Generation in Malnutrition Enteropathy in Zambia. *EBioMedicine* **22**, 191-199 (2017). <https://doi.org:10.1016/j.ebiom.2017.07.017>
- 17 Kelly, P. *et al.* Gastric and intestinal barrier impairment in tropical enteropathy and HIV: limited impact of micronutrient supplementation during a randomised controlled trial. *BMC Gastroenterol* **10**, 72 (2010). <https://doi.org:10.1186/1471-230X-10-72>
- 18 Lew, W. Y. *et al.* Recurrent exposure to subclinical lipopolysaccharide increases mortality and induces cardiac fibrosis in mice. *PLOS ONE* **8**, e61057 (2013). <https://doi.org:10.1371/journal.pone.0061057>

RESPONSE TO REVIEWER COMMENTS

Reviewer #3 (Remarks to the Author):

The authors have provided well-structured, and data-backed responses to most reviewer critiques. Their clarifications are mostly satisfactory. However, several issues require a more cautious tone or additional qualification.

Comment 1: Clarify Exploratory Nature of Mechanistic Claims

1. The authors emphasized their use of proteomic analysis, pathway mapping (via IPA), and re-analysis of public single-cell transcriptomic data to derive cell-type specific pathways (e.g., PI3K/AKT signaling). These remain inferences rather than direct mechanistic evidence. No functional validation (e.g., ex vivo assays, causal modeling) was added. While the systems biology analysis is useful, the claims should remain hypothesis-generating, and this framing is not fully emphasized in the revised response. The mechanistic claims (e.g., PI3K/AKT signaling) should be explicitly qualified as preliminary and exploratory

Response: We thank the reviewer for their thoughtful suggestion. The following edits have been made to highlight the exploratory nature of the pathway analysis of proteomics data:

- Addition of “exploratory” to line 61 of the *Abstract*.
- Line 473: **In this exploratory systems approach**, comparative pathway analysis of the four sub-groups revealed that cytokine related pathways like IL-10 and VEGF signaling were upregulated
- Line 476: Thus, **we hypothesize that** immune-cytokine dysregulation may contribute to increased mortality in the cohort.
- Lines 482-485: **Given the** pattern of increased IL-10 and VEGF signaling, reduced IL-15 production, enhanced PI3K/Akt signaling and reduced regulation by PTEN, **we infer** an immune-dysregulation **is likely** associated with LPS-correlated mortality (Figure 8C).
- Discussion, line 558: These together suggest **immune** dysregulation, which **we hypothesize** may contribute to increased mortality in the study settings.
- Discussion, lines 563-566: **Although these findings are preliminary, they point to PI3K/Akt** pathway as a potential therapeutic target **to be explored**. Since this study did not assess treatment courses, it cannot determine how treatment may have influenced mortality outcomes. Randomized clinical trials are needed to validate these findings.

Comment 2: Clarify Role of Comorbidities as Mediators vs Confounders

2. While I understand the rationale for treating comorbidities like malaria or LRTI as potential mediators in the pathway from enteric dysfunction to mortality, this assumption is not definitively supported by data and analysis. Many of these conditions could also act as confounders, independently influencing both systemic inflammation and mortality risk. I would recommend the authors to consider adjust for them in sensitivity analyses or a mediation analysis.

Response: We appreciate the reviewer's thoughtful comment. We agree that comorbidities such as malaria, LRTI, and others may serve as confounders, mediators, or even colliders in the causal pathway linking enteric dysfunction and mortality. These roles are likely complex, context-specific, and multifactorial. We acknowledge that our current analyses do not definitively establish a mediating role for these comorbidities, and we have now clarified this limitation in the revised manuscript.

To address the concern regarding potential confounding, we performed multivariable linear regressions of principal components derived from the *SomaScan* proteomic data (PC1 and PC2) with mutual adjustment for seven key comorbidities: HIV, gastroenteritis, malaria, sepsis, URTI, LRTI, and anthropometric classification. These analyses demonstrate that plasma LPS remains significantly associated with PC2 (Estimate = -0.547 , $p < 0.001$), even when adjusting for these comorbidities. This suggests that plasma LPS has an independent relationship with proteomic variation.

Additionally, *Supplementary Table 1* illustrates that several comorbidities significantly contribute to the variance in PC1 and PC2. For instance, malaria, gastroenteritis, sepsis, and anthropometric classification were associated with PC1, while malaria, HIV, LRTI, and plasma LPS were significantly associated with PC2. We also show in *Supplementary Figure 2* that plasma LPS explains approximately 15% of the variance in PC2 after adjusting for all comorbidities.

Together, these findings suggest that while comorbidities likely influence both systemic inflammation and mortality—potentially acting as confounders or mediators—plasma LPS retains an independent association with proteomic patterns relevant to mortality. We agree that further causal modeling or mediation analysis is warranted to disentangle these relationships and will consider this in future work.

Outcome	Term	Estimate	Std. Error	T value	P-value
PC1	(Intercept)	-6.992	3.079	-2.27	0.0235
PC1	`Plasma LPS`	-0.166	0.165	-1.01	0.315
PC1	Malaria	-7.142	2.860	-2.50	0.0128
PC1	HIV	6.214	4.120	1.51	0.132
PC1	Gastroenteritis	6.693	2.067	3.24	0.00127
PC1	Sepsis	-6.125	2.536	-2.41	0.016
PC1	LRTI	-1.598	2.052	-0.78	0.436
PC1	URTI	4.029	4.444	0.91	0.365
PC1	`Anthropometric classification`	3.238	1.157	2.80	0.00528
PC2	(Intercept)	6.433	2.418	2.66	0.008
PC2	`Plasma LPS`	-0.547	0.130	-4.21	<0.001
PC2	Malaria	-18.995	2.246	-8.46	<0.001
PC2	HIV	-11.554	3.235	-3.57	<0.001
PC2	Gastroenteritis	-0.455	1.623	-0.28	0.779
PC2	Sepsis	-2.362	1.992	-1.19	0.236
PC2	LRTI	-5.074	1.611	-3.15	0.00172
PC2	URTI	3.476	3.490	1.00	0.32
PC2	`Anthropometric classification`	1.324	0.908	1.46	0.146

Supplementary Table 1: Adjusted linear regression results for associations between plasma LPS, comorbidities, and principal components PC1 and PC2.

Linear regression models were fitted separately for PC1 and PC2 as outcome variables. Each model included plasma LPS concentration and the following comorbidities as covariates: malaria, HIV, gastroenteritis, sepsis, lower respiratory tract infection (LRTI), upper respiratory tract infection (URTI), and anthropometric classification. The table presents regression coefficients (Estimates), standard errors, t-values, and p-values for each term in the model. Values are rounded to three decimal places, and p-values less than 0.001 are reported as "<0.001". Statistically significant associations are highlighted in green.

Adjusted Relationship: Plasma LPS vs PC2

Adjusted for all comorbidities | Adjusted $R^2 = 0.15$ | Plasma LPS $p < 0.001$

Supplementary Figure 2: Adjusted association between plasma LPS concentration and PC2.

A linear regression model was generated to examine the relationship between plasma LPS concentration and principal component 2 (PC2) of the *SomaScan* proteomics data, adjusting for comorbidities including malaria, HIV, gastroenteritis, sepsis, lower respiratory tract infection (LRTI), upper respiratory tract infection (URTI), and anthropometric classification. Each point represents an individual participant. The blue line shows the fitted regression line with 95% confidence interval. The model yielded an adjusted R^2 of 0.15 and a p-value for the plasma LPS coefficient of <0.001 .

Changes made:

- The following (highlighted in red) was added to the Methods section:

Lines 285-292: For summary statistics, correlation and pathway analysis of proteomics data, clinical comorbidities were not adjusted for, as their roles in the causal framework are likely complex—potentially acting as mediators, confounders, or colliders. Given this ambiguity, adjustment could introduce bias or obscure meaningful biological associations. However, sensitivity analyses using linear models of principal components (PC1 and PC2) demonstrated that plasma LPS remained independently associated with significant variance in the proteomic data, even when key comorbidities were mutually adjusted (*Supplementary Figure 2, Supplementary Table 1*).